# 🦙 TIME-LLM: TIME SERIES FORECASTING BY REPROGRAMMING LARGE LANGUAGE MODELS

**Ming Jin**[1]*, **Shiyu Wang**[2]*, **Lintao Ma**[2], **Zhixuan Chu**[2], **James Y. Zhang**[2], **Xiaoming Shi**[2],
**Pin-Yu Chen**[3], **Yuxuan Liang**[6], **Yuan-Fang Li**[1], **Shirui Pan**[4]†, **Qingsong Wen**[5]†

[1]Monash University [2]Ant Group [3]IBM Research [4]Griffith University [5]Alibaba Group
[6]The Hong Kong University of Science and Technology (Guangzhou)

```
{ming.jin, yuanfang.li}@monash.edu, pin-yu.chen@ibm.com
yuxliang@outlook.com, s.pan@griffith.edu.au, qingsongedu@gmail.com
{weiming.wsy,lintao.mlt,chuzhixuan.czx,james.z,peter.sxm}@antgroup.com
```

## ABSTRACT

Time series forecasting holds significant importance in many real-world dynamic systems and has been extensively studied. Unlike natural language process (NLP) and computer vision (CV), where a single large model can tackle multiple tasks, models for time series forecasting are often specialized, necessitating distinct designs for different tasks and applications. While pre-trained foundation models have made impressive strides in NLP and CV, their development in time series domains has been constrained by data sparsity. Recent studies have revealed that large language models (LLMs) possess robust pattern recognition and reasoning abilities over complex sequences of tokens. However, the challenge remains in effectively aligning the modalities of time series data and natural language to leverage these capabilities. In this work, we present TIME-LLM, a reprogramming framework to repurpose LLMs for general time series forecasting with the backbone language models kept intact. We begin by reprogramming the input time series with text prototypes before feeding it into the frozen LLM to align the two modalities. To augment the LLM's ability to reason with time series data, we propose Prompt-as-Prefix (PaP), which enriches the input context and directs the transformation of reprogrammed input patches. The transformed time series patches from the LLM are finally projected to obtain the forecasts. Our comprehensive evaluations demonstrate that TIME-LLM is a powerful time series learner that outperforms state-of-the-art, specialized forecasting models. Moreover, TIME-LLM excels in both few-shot and zero-shot learning scenarios. The code is made available at https://github.com/KimMeen/Time-LLM.

## 1 INTRODUCTION

Time series forecasting is a critical capability across many real-world dynamic systems (Jin et al., 2023a; Hu et al., 2024), with applications ranging from demand planning (Leonard, 2001) and inventory optimization (Li et al., 2022) to energy load forecasting (Liu et al., 2023a) and climate modeling (Schneider & Dickinson, 1974). Each time series forecasting task typically requires extensive domain expertise and task-specific model designs. This stands in stark contrast to foundation language models like GPT-3 (Brown et al., 2020), GPT-4 (OpenAI, 2023), Llama (Touvron et al., 2023), *inter alia*, which can perform well on a diverse range of NLP tasks in a few-shot or even zero-shot setting.

Pre-trained foundation models, such as large language models (LLMs), have driven rapid progress in computer vision (CV) and natural language processing (NLP). While time series modeling has not benefited from the same significant breakthroughs, LLMs' impressive capabilities have inspired their application to time series forecasting (Jin et al., 2023b; 2024). Several desiderata exist for leveraging LLMs to advance forecasting techniques: ***Generalizability.*** LLMs have demonstrated a remarkable capability for few-shot and zero-shot transfer learning (Brown et al., 2020). This suggests their

---

*Equal Contribution
†Corresponding Authors

potential for generalizable forecasting across domains without requiring per-task retraining from scratch. In contrast, current forecasting methods are often rigidly specialized by domain. ***Data efficiency.*** By leveraging pre-trained knowledge, LLMs have shown the ability to perform new tasks with only a few examples. This data efficiency could enable forecasting for settings where historical data is limited. In contrast, current methods typically require abundant in-domain data. ***Reasoning.*** LLMs exhibit sophisticated reasoning and pattern recognition capabilities (Mirchandani et al., 2023; Luo et al., 2023b;a). Harnessing these skills could allow making highly precise forecasts by leveraging learned higher-level concepts. Existing non-LLM methods are largely statistical without much innate reasoning. ***Multimodal knowledge.*** As LLM architectures and training techniques improve, they gain more diverse knowledge across modalities like vision, speech, and text (Ma et al., 2023). Tapping into this knowledge could enable synergistic forecasting that fuses different data types. Conventional tools lack ways to jointly leverage multiple knowledge bases. ***Easy optimization.*** LLMs are trained once on massive computing and then can be applied to forecasting tasks without learning from scratch. Optimizing existing forecasting models often requires significant architecture search and hyperparameter tuning (Zhou et al., 2023b). In summary, LLMs offer a promising path to make time series forecasting more general, efficient, synergistic, and accessible compared to current specialized modeling paradigms. Thus, adapting these powerful models for time series data can unlock significant untapped potential.

The realization of the above benefits hinges on the effective alignment of the modalities of time series data and natural language. However, this is a challenging task as LLMs operate on discrete tokens, while time series data is inherently continuous. Furthermore, the knowledge and reasoning capabilities to interpret time series patterns are not naturally present within LLMs' pre-training. Therefore, it remains an open challenge to unlock the knowledge within LLMs in activating their ability for general time series forecasting in a way that is accurate, data-efficient, and task-agnostic.

In this work, we propose TIME-LLM, a reprogramming framework to adapt large language models for time series forecasting while keeping the backbone model intact. The core idea is to *reprogram* the input time series into text prototype representations that are more naturally suited to language models' capabilities. To further augment the model's reasoning about time series concepts, we introduce *Prompt-as-Prefix* (PaP), a novel idea in enriching the input time series with additional context and providing task instructions in the modality of natural language. This provides declarative guidance about desired transformations to apply to the reprogrammed input. The output of the language model is then projected to generate time series forecasts. Our comprehensive evaluation demonstrates that large language models can act as effective few-shot and zero-shot time series learners when adopted through this reprogramming approach, outperforming specialized forecasting models. By leveraging LLMs' reasoning capability while keeping the models intact, our work points the way toward multimodal foundation models that can excel on both language and sequential data tasks. Our proposed reprogramming framework offers an extensible paradigm for imbuing large models with new capabilities beyond their original pre-training. Our main contributions in this work can be summarized as follows:

- We introduce a novel concept of *reprogramming* large language models for time series forecasting without altering the pre-trained backbone model. In doing so, we show that forecasting can be cast as yet another "language" task that can be effectively tackled by an off-the-shelf LLM.

- We propose a new framework, TIME-LLM, which encompasses reprogramming the input time series into text prototype representations that are more natural for the LLM, and augmenting the input context with declarative prompts (e.g., domain expert knowledge and task instructions) to guide LLM reasoning. Our technique points towards multimodal foundation models excelling in both language and time series.

- TIME-LLM consistently exceeds state-of-the-art performance in mainstream forecasting tasks, especially in few-shot and zero-shot scenarios. Moreover, this superior performance is achieved while maintaining excellent model reprogramming efficiency. Thus, our research is a concrete step in unleashing LLMs' untapped potential for time series and perhaps other sequential data.

## 2    RELATED WORK

**Task-specific Learning.** Most time series forecasting models are crafted for specific tasks and domains (e.g., traffic prediction), and trained end-to-end on small-scale data. An illustration is in

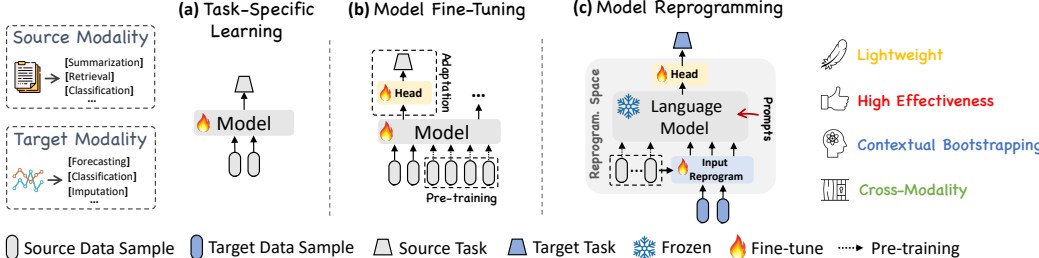

Figure 1: Schematic illustration of reprogramming large language models (LLMs) in comparison of **(a)** task-specific learning and **(b)** model fine-tuning. Our proposal investigates and demonstrates **(c)** how to effectively reprogram open-sourced LLMs as powerful time series learners where well-developed time series pre-trained models are not readily available.

Fig. 1(a). For example, ARIMA models are designed for univariate time series forecasting (Box et al., 2015), LSTM networks are tailored for sequence modeling (Hochreiter & Schmidhuber, 1997), and temporal convolutional networks (Bai et al., 2018) and transformers (Wen et al., 2023) are developed for handling longer temporal dependencies. While achieving good performance on narrow tasks, these models lack versatility and generalizability to diverse time series data.

**In-modality Adaptation.** Relevant research in CV and NLP has demonstrated the effectiveness of pre-trained models that can be fine-tuned for various downstream tasks without the need for costly training from scratch (Devlin et al., 2018; Brown et al., 2020; Touvron et al., 2023). Inspired by these successes, recent studies have focused on the development of time series pre-trained models (TSPTMs). The first step among them involves time series pre-training using different strategies like supervised (Fawaz et al., 2018) or self-supervised learning (Zhang et al., 2022b; Deldari et al., 2022; Zhang et al., 2023). This allows the model to learn representing various input time series. Once pre-trained, it can be fine-tuned on similar domains to learn how to perform specific tasks (Tang et al., 2022). An example is in Fig. 1(b). The development of TSPTMs leverages the success of pre-training and fine-tuning in NLP and CV but remains limited on smaller scales due to data sparsity.

**Cross-modality Adaptation.** Building on in-modality adaptation, recent work has further explored transferring knowledge from powerful pre-trained foundations models in NLP and CV to time series modeling, through techniques such as multimodal fine-tuning (Yin et al., 2023) and model reprogramming (Chen, 2022). Our approach aligns with this category; however, there is limited pertinent research available on time series. An example is Voice2Series (Yang et al., 2021), which adapts an acoustic model (AM) from speech recognition to time series classification by editing a time series into a format suitable for the AM. Recently, Chang et al. (2023) proposes LLM4TS for time series forecasting using LLMs. It designs a two-stage fine-tuning process on the LLM - first supervised pre-training on time series, then task-specific fine-tuning. Zhou et al. (2023a) leverages pre-trained language models without altering their self-attention and feedforward layers. This model is fine-tuned and evaluated on various time series analysis tasks and demonstrates comparable or state-of-the-art performance by transferring knowledge from natural language pre-training. Distinct from these approach, we neither edit the input time series directly nor fine-tune the backbone LLM. Instead, as illustrated in Fig. 1(c), we propose reprogramming time series with the source data modality along with prompting to unleash the potential of LLMs as effective time series machines.

## 3 METHODOLOGY

Our model architecture is depicted in Fig. 2. We focus on reprogramming an embedding-visible language foundation model, such as Llama (Touvron et al., 2023) and GPT-2 (Radford et al., 2019), for general time series forecasting ***without*** requiring any fine-tuning of the backbone model. Specifically, we consider the following problem: given a sequence of historical observations $\mathbf{X} \in \mathbb{R}^{N \times T}$ consisting of $N$ different 1-dimensional variables across $T$ time steps, we aim to reprogram a large language model $f(\cdot)$ to understand the input time series and accurately forecast the readings at $H$ future time steps, denoted by $\hat{\mathbf{Y}} \in \mathbb{R}^{N \times H}$, with the overall objective to minimize the mean square errors between the ground truths $\mathbf{Y}$ and predictions, i.e., $\frac{1}{H} \sum_{h=1}^{H} ||\hat{\mathbf{Y}}_h - \mathbf{Y}_h||_F^2$.

Our method encompasses three main components: (1) input transformation, (2) a pre-trained and frozen LLM, and (3) output projection. Initially, a multivariate time series is partitioned into $N$

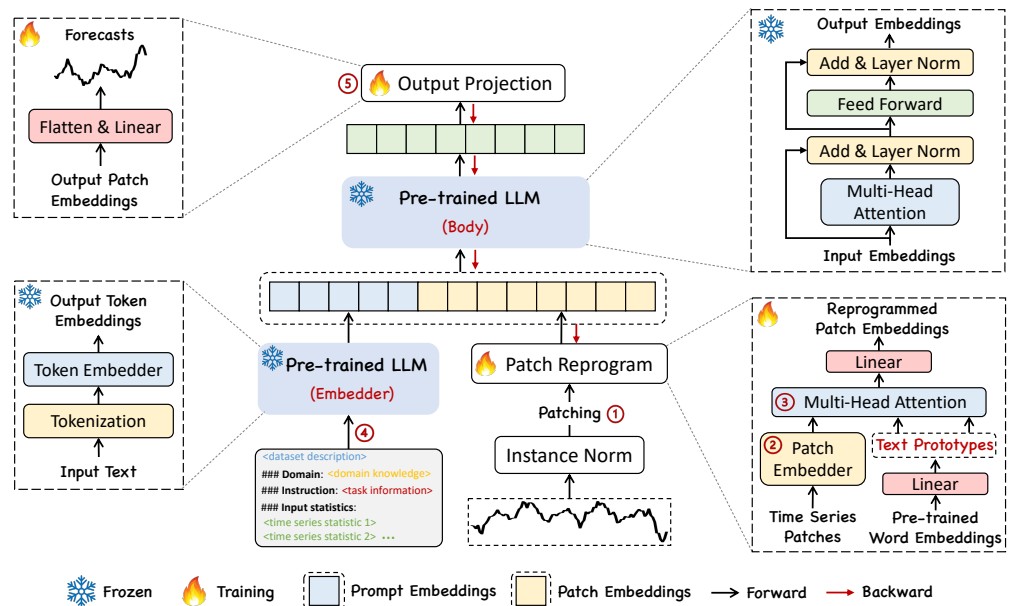

Figure 2: The model framework of TIME-LLM. Given an input time series, we first tokenize and embed it via ① patching along with a ② customized embedding layer. ③ These patch embeddings are then reprogrammed with condensed text prototypes to align two modalities. To augment the LLM's reasoning ability, ④ additional prompt prefixes are added to the input to direct the transformation of input patches. ⑤ The output patches from the LLM are projected to generate the forecasts.

univariate time series, which are subsequently processed independently (Nie et al., 2023). The $i$-th series is denoted as $\mathbf{X}^{(i)} \in \mathbb{R}^{1 \times T}$, which undergoes normalization, patching, and embedding prior to being reprogrammed with learned text prototypes to align the source and target modalities. Then, we augment the LLM's time series reasoning ability by prompting it together with reprogrammed patches to generate output representations, which are projected to the final forecasts $\hat{\mathbf{Y}}^{(i)} \in \mathbb{R}^{1 \times H}$.

We note that only the parameters of the lightweight input transformation and output projection are updated, while the backbone language model is frozen. In contrast to vision-language and other multimodal language models, which usually fine-tune with paired cross-modality data, TIME-LLM is directly optimized and becomes readily available with only a small set of time series and a few training epochs, maintaining high efficiency and imposing fewer resource constraints compared to building large domain-specific models from scratch or fine-tuning them. To further reduce memory footprints, various off-the-shelf techniques (e.g., quantization) can be seamlessly integrated for slimming TIME-LLM.

## 3.1 MODEL STRUCTURE

**Input Embedding.** Each input channel $\mathbf{X}^{(i)}$ is first individually normalized to have zero mean and unit standard deviation via reversible instance normalization (RevIN) in mitigating the time series distribution shift (Kim et al., 2021). Then, we divide $\mathbf{X}^{(i)}$ into several consecutive overlapped or non-overlapped patches (Nie et al., 2023) with length $L_p$; thus the total number of input patches is $P = \lfloor \frac{(T-L_p)}{S} \rfloor + 2$, where $S$ denotes the horizontal sliding stride. The underlying motivations are two-fold: (1) better preserving local semantic information by aggregating local information into each patch and (2) serving as tokenization to form a compact sequence of input tokens, reducing computational burdens. Given these patches $\mathbf{X}_P^{(i)} \in \mathbb{R}^{P \times L_p}$, we embed them as $\hat{\mathbf{X}}_P^{(i)} \in \mathbb{R}^{P \times d_m}$, adopting a simple linear layer as the patch embedder to create dimensions $d_m$.

**Patch Reprogramming.** Here we reprogram patch embeddings into the source data representation space to align the modalities of time series and natural language to activate the backbone's time series understanding and reasoning capabilities. A common practice is learning a form of "noise" that, when applied to target input samples, allows the pre-trained source model to produce the desired target outputs without requiring parameter updates. This is technically feasible for bridging data

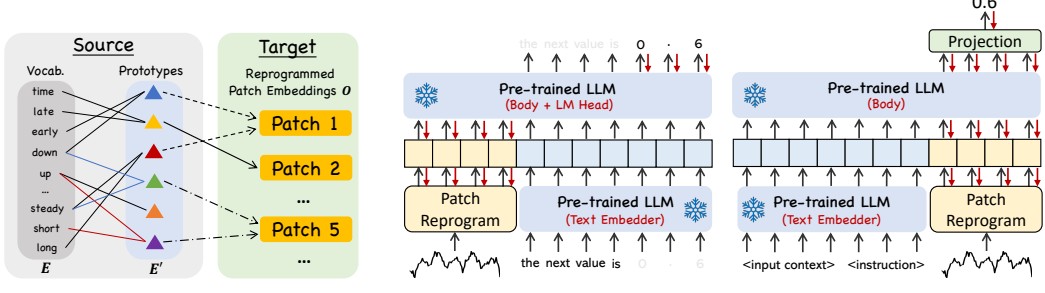

**(a)** Patch Reprogramming        **(b)** Patch-as-Prefix and Prompt-as-Prefix

Figure 3: Illustration of **(a)** patch reprogramming and **(b)** Patch-as-Prefix versus Prompt-as-Prefix.

modalities that are identical or similar. Examples include repurposing a vision model to work with cross-domain images (Misra et al., 2023) or reprogramming an acoustic model to handle time series data (Yang et al., 2021). In both cases, there are explicit, learnable transformations between the source and target data, allowing for the direct editing of input samples. However, time series can neither be directly edited nor described losslessly in natural language, posing significant challenges to directly bootstrap the LLM for understanding time series without resource-intensive fine-tuning.

To close this gap, we propose reprogramming $\hat{\mathbf{X}}_P^{(i)}$ using pre-trained word embeddings $\mathbf{E} \in \mathbb{R}^{V \times D}$ in the backbone, where $V$ is the vocabulary size. Nevertheless, there is no prior knowledge indicating which source tokens are directly relevant. Thus, simply leveraging $\mathbf{E}$ will result in large and potentially dense reprogramming space. A simple solution is to maintain a small collection of text prototypes by linearly probing $\mathbf{E}$, denoted as $\mathbf{E}' \in \mathbb{R}^{V' \times D}$, where $V' \ll V$. An illustration is in Fig. 3(a). Text prototypes learn connecting language cues, e.g., "short up" (red lines) and "steady down" (blue lines), which are then combined to represent the local patch information (e.g., "short up then down steadily" for characterizing patch 5) without leaving the space where the language model is pre-trained. This approach is efficient and allows for the adaptive selection of relevant source information. To realize this, we employ a multi-head cross-attention layer. Specifically, for each head $k = \{1, \cdots, K\}$, we define query matrices $\mathbf{Q}_k^{(i)} = \hat{\mathbf{X}}_P^{(i)} \mathbf{W}_k^Q$, key matrices $\mathbf{K}_k^{(i)} = \mathbf{E}' \mathbf{W}_k^K$, and value matrices $\mathbf{V}_k^{(i)} = \mathbf{E}' \mathbf{W}_k^V$, where $\mathbf{W}_k^Q \in \mathbb{R}^{d_m \times d}$ and $\mathbf{W}_k^K, \mathbf{W}_k^V \in \mathbb{R}^{D \times d}$. Specifically, $D$ is the hidden dimension of the backbone model, and $d = \lfloor \frac{d_m}{K} \rfloor$. Then, we have the operation to reprogram time series patches in each attention head defined as:

$$\mathbf{Z}_k^{(i)} = \text{ATTENTION}(\mathbf{Q}_k^{(i)}, \mathbf{K}_k^{(i)}, \mathbf{V}_k^{(i)}) = \text{SOFTMAX}\left(\frac{\mathbf{Q}_k^{(i)} \mathbf{K}_k^{(i)\top}}{\sqrt{d_k}}\right) \mathbf{V}_k^{(i)}. \quad (1)$$

By aggregating each $\mathbf{Z}_k^{(i)} \in \mathbb{R}^{P \times d}$ in every head, we obtain $\mathbf{Z}^{(i)} \in \mathbb{R}^{P \times d_m}$. This is then linearly projected to align the hidden dimensions with the backbone model, yielding $\mathbf{O}^{(i)} \in \mathbb{R}^{P \times D}$.

**Prompt-as-Prefix.** Prompting serves as a straightforward yet effective approach task-specific activation of LLMs (Yin et al., 2023). However, the direct translation of time series into natural language presents considerable challenges, hindering both the creation of instruction-following datasets and the effective utilization of on-the-fly prompting without performance compromise (Xue & Salim, 2022). Recent advancements indicate that other data modalities, such as images, can be seamlessly integrated as the prefixes of prompts, thereby facilitating effective reasoning based on these inputs (Tsimpoukelli et al., 2021). Motivated by these findings, and to render our approach directly applicable to real-world time series, we pose an alternative question: *can prompts act as prefixes to enrich the input context and guide the transformation of reprogrammed time series patches?* We term this concept as *Prompt-as-Prefix* (PaP) and observe that it significantly enhances the LLM's adaptability to downstream tasks while complementing patch reprogramming (See Sec. 4.5 later).

The Electricity Transformer Temperature (ETT) indicates the electric power long-term deployment. Each data point consists of the target oil temperature and 6 power load features ... Below is the information about the input time series:

**[BEGIN DATA]**
***
**[Domain]:** We usually observe that electricity consumption peaks at noon, with a significant increase in transformer load
***
**[Instruction]**: Predict the next \<H\> steps given the previous \<T\> steps information attached
***
**[Statistics]:** The input has a minimum of \<min_val\>, a maximum of \<max_val\>, and a median of \<median_val\>. The overall trend is \<upward or downward\>. The top five lags are \<lag_val\>.
**[END DATA]**

Figure 4: Prompt example. \<\> and \<\> are task-specific configurations and calculated input statistics.

An illustration of the two prompting approaches is in Fig. 3(b). In *Patch-as-Prefix*, a language model is prompted to predict subsequent values in a time series, articulated in natural language. This approach encounters certain constraints: (1) language models typically exhibit reduced sensitivity in processing high-precision numerals without the aid of external tools, thereby presenting substantial challenges in accurately addressing practical forecasting tasks over long horizons; (2) intricate, customized post-processing is required for different language models, given that they are pre-trained on diverse corpora and may employ different tokenization types in generating high-precision numerals with precision and efficiency. This results in forecasts being represented in disparate natural language formats, such as ['0', '.', '6', '1'] and ['0', '.', '61'], to denote the decimal 0.61.

*Prompt-as-Prefix*, on the other hand, tactfully avoids these constraints. In practice, we identify three pivotal components for constructing effective prompts: (1) dataset context, (2) task instruction, and (3) input statistics. A prompt example is in Fig. 4. The dataset context furnishes the LLM with essential background information concerning the input time series, which often exhibits distinct characteristics across various domains. Task instruction serves as a crucial guide for the LLM in the transformation of patch embeddings for specific tasks. We also enrich the input time series with additional crucial statistics, such as trends and lags, to facilitate pattern recognition and reasoning.

**Output Projection.** Upon packing and feedforwarding the prompt and patch embeddings $\mathbf{O}^{(i)}$ through the frozen LLM as shown in Fig. 2, we discard the prefixal part and obtain the output representations. Following this, we flatten and linear project them to derive the final forecasts $\hat{\mathbf{Y}}^{(i)}$.

## 4 MAIN RESULTS

TIME-LLM consistently outperforms state-of-the-art forecasting methods by large margins across multiple benchmarks and settings, especially in few-shot and zero-shot scenarios. We compared our approach against a broad collection of up-to-date models, including a recent study that fine-tunes language model for time series analysis (Zhou et al., 2023a). To ensure a fair comparison, we adhere to the experimental configurations in (Wu et al., 2023) across all baselines with a unified evaluation pipeline[1]. We use Llama-7B (Touvron et al., 2023) as the default backbone unless stated otherwise.

**Baselines.** We compare with the SOTA time series models, and we cite their performance from (Zhou et al., 2023a) if applicable. Our baselines include a series of Transformer-based methods: PatchTST (2023), ESTformer (2022), Non-Stationary Transformer (2022), FEDformer (2022), Autoformer (2021), Informer (2021), and Reformer (2020). We also select a set of recent competitive models, including GPT4TS (2023a), LLMTime (2023), DLinear (2023), TimesNet (2023), and LightTS (2022a). In short-term forecasting, we further compare our model with N-HiTS (2023b) and N-BEATS (2020). More details are in Appendix A.

### 4.1 LONG-TERM FORECASTING

**Setups.** We evaluate on ETTh1, ETTh2, ETTm1, ETTm2, Weather, Electricity (ECL), Traffic, and ILI, which have been extensively adopted for benchmarking long-term forecasting models (Wu et al., 2023). Details of the implementation and datasets can be found in Appendix B. The input time series length $T$ is set as 512, and we use four different prediction horizons $H \in \{96, 192, 336, 720\}$. The evaluation metrics include mean square error (MSE) and mean absolute error (MAE).

**Results.** Our brief results are shown in Tab. 1, where TIME-LLM outperforms all baselines in most cases and significantly so to the majority of them. The comparison with GPT4TS (Zhou et al., 2023a) is particularly noteworthy. GPT4TS is a very recent work that involves fine-tuning on the backbone language model. We note average performance gains of **12%** and **20%** over GPT4TS and TimesNet, respectively. When compared with the SOTA task-specific Transformer model PatchTST, by reprogramming the smallest Llama, TIME-LLM realizes an average MSE reduction of 1.4%. Relative to the other models, e.g., DLinear, our improvements are also pronounced, exceeding **12%**.

### 4.2 SHORT-TERM FORECASTING

**Setups.** We choose the M4 benchmark (Makridakis et al., 2018) as the testbed, which contains a collection of marketing data in different sampling frequencies. More details are provided in Appendix B. The prediction horizons in this case are relatively small and in [6, 48]. The input lengths

---

[1]https://github.com/thuml/Time-Series-Library

Table 1: Long-term forecasting results. All results are averaged from four different forecasting horizons: $H \in \{24, 36, 48, 60\}$ for ILI and $\{96, 192, 336, 720\}$ for the others. A lower value indicates better performance. **Red**: the best, Blue: the second best. Our full results are in Appendix D.

| Methods | TIME-LLM (**Ours**) | | GPT4TS (2023a) | | DLinear (2023) | | PatchTST (2023) | | TimesNet (2023) | | FEDformer (2022) | | Autoformer (2021) | | Stationary (2022) | | ETSformer (2022) | | LightTS (2022a) | | Informer (2021) | | Reformer (2020) | |
|---|---|---|---|---|---|---|---|---|---|---|---|---|---|---|---|---|---|---|---|---|---|---|---|---|
| Metric | MSE | MAE | MSE | MAE | MSE | MAE | MSE | MAE | MSE | MAE | MSE | MAE | MSE | MAE | MSE | MAE | MSE | MAE | MSE | MAE | MSE | MAE | MSE | MAE |
| $ETTh1$ | **0.408** | **0.423** | 0.465 | 0.455 | 0.422 | 0.437 | 0.413 | 0.430 | 0.458 | 0.450 | 0.440 | 0.460 | 0.496 | 0.487 | 0.570 | 0.537 | 0.542 | 0.510 | 0.491 | 0.479 | 1.040 | 0.795 | 1.029 | 0.805 |
| $ETTh2$ | 0.334 | 0.383 | 0.381 | 0.412 | 0.431 | 0.446 | **0.330** | **0.379** | 0.414 | 0.427 | 0.437 | 0.449 | 0.450 | 0.459 | 0.526 | 0.516 | 0.439 | 0.452 | 0.602 | 0.543 | 4.431 | 1.729 | 6.736 | 2.191 |
| $ETTm1$ | **0.329** | **0.372** | 0.388 | 0.403 | 0.357 | 0.378 | 0.351 | 0.380 | 0.400 | 0.406 | 0.448 | 0.452 | 0.588 | 0.517 | 0.481 | 0.456 | 0.429 | 0.425 | 0.435 | 0.437 | 0.961 | 0.734 | 0.799 | 0.671 |
| $ETTm2$ | **0.251** | **0.313** | 0.284 | 0.339 | 0.267 | 0.333 | 0.255 | 0.315 | 0.291 | 0.333 | 0.305 | 0.349 | 0.327 | 0.371 | 0.306 | 0.347 | 0.293 | 0.342 | 0.409 | 0.436 | 1.410 | 0.810 | 1.479 | 0.915 |
| $Weather$ | **0.225** | **0.257** | 0.237 | 0.270 | 0.248 | 0.300 | **0.225** | 0.264 | 0.259 | 0.287 | 0.309 | 0.360 | 0.338 | 0.382 | 0.288 | 0.314 | 0.271 | 0.334 | 0.261 | 0.312 | 0.634 | 0.548 | 0.803 | 0.656 |
| $ECL$ | **0.158** | **0.252** | 0.167 | 0.263 | 0.166 | 0.263 | 0.161 | **0.252** | 0.192 | 0.295 | 0.214 | 0.327 | 0.227 | 0.338 | 0.193 | 0.296 | 0.208 | 0.323 | 0.229 | 0.329 | 0.311 | 0.397 | 0.338 | 0.422 |
| $Traffic$ | **0.388** | 0.264 | 0.414 | 0.294 | 0.433 | 0.295 | 0.390 | **0.263** | 0.620 | 0.336 | 0.610 | 0.376 | 0.628 | 0.379 | 0.624 | 0.340 | 0.621 | 0.396 | 0.622 | 0.392 | 0.764 | 0.416 | 0.741 | 0.422 |
| $ILI$ | **1.435** | 0.801 | 1.925 | 0.903 | 2.169 | 1.041 | 1.443 | **0.797** | 2.139 | 0.931 | 2.847 | 1.144 | 3.006 | 1.161 | 2.077 | 0.914 | 2.497 | 1.004 | 7.382 | 2.003 | 5.137 | 1.544 | 4.724 | 1.445 |
| $1^{st}$Count | **7** | | 0 | | 0 | | 5 | | 0 | | 0 | | 0 | | 0 | | 0 | | 0 | | 0 | | 0 | |

Table 2: Short-term time series forecasting results on M4. The forecasting horizons are in [6, 48] and the three rows provided are weighted averaged from all datasets under different sampling intervals. A lower value indicates better performance. **Red**: the best, Blue: the second best. More results are in Appendix D.

| Methods | | TIME-LLM (**Ours**) | GPT4TS (2023a) | TimesNet (2023) | PatchTST (2023) | N-HiTS (2023b) | N-BEATS (2020) | ETSformer (2022) | LightTS (2022a) | DLinear (2023) | FEDformer (2022) | Stationary (2022) | Autoformer (2021) | Informer (2021) | Reformer (2020) |
|---|---|---|---|---|---|---|---|---|---|---|---|---|---|---|---|
| Average | SMAPE | **11.983** | 12.69 | 12.88 | 12.059 | 12.035 | 12.25 | 14.718 | 13.525 | 13.639 | 13.16 | 12.780 | 12.909 | 14.086 | 18.200 |
| | MASE | **1.595** | 1.808 | 1.836 | 1.623 | 1.625 | 1.698 | 2.408 | 2.111 | 2.095 | 1.775 | 1.756 | 1.771 | 2.718 | 4.223 |
| | OWA | **0.859** | 0.94 | 0.955 | 0.869 | 0.869 | 0.896 | 1.172 | 1.051 | 1.051 | 0.949 | 0.930 | 0.939 | 1.230 | 1.775 |

are twice as prediction horizons. The evaluation metrics are symmetric mean absolute percentage error (SMAPE), mean absolute scaled error (MSAE), and overall weighted average (OWA).

**Results.** Our brief results with unified seeds across all methods are in Tab. 2. TIME-LLM consistently surpasses all baselines, outperforming GPT4TS by **8.7%**. TIME-LLM remains competitive even when compared with the SOTA model, N-HiTS (Challu et al., 2023b) , w.r.t. MASE and OWA.

## 4.3 FEW-SHOT FORECASTING

**Setups.** LLMs have recently demonstrated remarkable few-shot learning capabilities (Liu et al., 2023b). In this section, we assess whether our reprogrammed LLM retains this ability in forecasting tasks. We adhere to the setups in (Zhou et al., 2023a) for fair comparisons, and we evaluate on scenarios with limited training data (i.e., $\leq$ first $10\%$ training time steps).

**Results.** Our brief 10% and 5% few-shot learning results are in Tab. 3 and Tab. 4 respectively. TIME-LLM remarkably excels over all baseline methods, and we attribute this to the successful knowledge activation in our reprogrammed LLM. Interestingly, both our approach and GPT4TS consistently surpass other competitive baselines, further underscoring the potential prowess of language models as proficient time series machines.

In the realm of 10% few-shot learning, our methodology realizes a **5%** MSE reduction in comparison to GPT4TS, without necessitating any fine-tuning on the LLM. In relation to recent SOTA models

Table 3: Few-shot learning on 10% training data. We use the same protocol in Tab. 1. All results are averaged from four different forecasting horizons: $H \in \{96, 192, 336, 720\}$. Our full results are in Appendix E.

| Methods | TIME-LLM (**Ours**) | | GPT4TS (2023a) | | DLinear (2023) | | PatchTST (2023) | | TimesNet (2023) | | FEDformer (2022) | | Autoformer (2021) | | Stationary (2022) | | ETSformer (2022) | | LightTS (2022a) | | Informer (2021) | | Reformer (2020) | |
|---|---|---|---|---|---|---|---|---|---|---|---|---|---|---|---|---|---|---|---|---|---|---|---|---|
| Metric | MSE | MAE | MSE | MAE | MSE | MAE | MSE | MAE | MSE | MAE | MSE | MAE | MSE | MAE | MSE | MAE | MSE | MAE | MSE | MAE | MSE | MAE | MSE | MAE |
| $ETTh1$ | **0.556** | **0.522** | 0.590 | 0.525 | 0.691 | 0.600 | 0.633 | 0.542 | 0.869 | 0.628 | 0.639 | 0.561 | 0.702 | 0.596 | 0.915 | 0.639 | 1.180 | 0.834 | 1.375 | 0.877 | 1.199 | 0.809 | 1.249 | 0.833 |
| $ETTh2$ | **0.370** | **0.394** | 0.397 | 0.421 | 0.605 | 0.538 | 0.415 | 0.431 | 0.479 | 0.465 | 0.466 | 0.475 | 0.488 | 0.499 | 0.462 | 0.455 | 0.894 | 0.713 | 2.655 | 1.160 | 3.872 | 1.513 | 3.485 | 1.486 |
| $ETTm1$ | **0.404** | **0.427** | 0.464 | 0.441 | 0.411 | 0.429 | 0.501 | 0.466 | 0.677 | 0.537 | 0.722 | 0.605 | 0.802 | 0.628 | 0.797 | 0.578 | 0.980 | 0.714 | 0.971 | 0.705 | 1.192 | 0.821 | 1.426 | 0.856 |
| $ETTm2$ | **0.277** | **0.323** | 0.293 | 0.335 | 0.316 | 0.368 | 0.296 | 0.343 | 0.320 | 0.353 | 0.463 | 0.488 | 1.342 | 0.930 | 0.332 | 0.366 | 0.447 | 0.487 | 0.987 | 0.756 | 3.370 | 1.440 | 3.978 | 1.587 |
| $Weather$ | **0.234** | **0.273** | 0.238 | 0.275 | 0.241 | 0.283 | 0.242 | 0.279 | 0.279 | 0.301 | 0.284 | 0.324 | 0.300 | 0.342 | 0.318 | 0.323 | 0.318 | 0.360 | 0.289 | 0.322 | 0.597 | 0.495 | 0.546 | 0.469 |
| $ECL$ | **0.175** | 0.270 | 0.176 | **0.269** | 0.180 | 0.280 | 0.180 | 0.273 | 0.323 | 0.392 | 0.346 | 0.427 | 0.431 | 0.478 | 0.444 | 0.480 | 0.660 | 0.617 | 0.441 | 0.489 | 1.195 | 0.891 | 0.965 | 0.768 |
| $Traffic$ | **0.429** | 0.306 | 0.440 | 0.310 | 0.447 | 0.313 | 0.430 | **0.305** | 0.951 | 0.535 | 0.663 | 0.425 | 0.749 | 0.446 | 1.453 | 0.815 | 1.914 | 0.936 | 1.248 | 0.684 | 1.534 | 0.811 | 1.551 | 0.821 |
| $1^{st}$Count | **7** | | 1 | | 0 | | 1 | | 0 | | 0 | | 0 | | 0 | | 0 | | 0 | | 0 | | 0 | |

Table 4: Few-shot learning on 5% training data. We use the same protocol in Tab. 1. All results are averaged from four different forecasting horizons: $H \in \{96, 192, 336, 720\}$. Our full results are in Appendix E.

| Methods | TIME-LLM (Ours) | | GPT4TS (2023a) | | DLinear (2023) | | PatchTST (2023) | | TimesNet (2023) | | FEDformer (2022) | | Autoformer (2021) | | Stationary (2022) | | ETSformer (2022) | | LightTS (2022a) | | Informer (2021) | | Reformer (2020) | |
|---|---|---|---|---|---|---|---|---|---|---|---|---|---|---|---|---|---|---|---|---|---|---|---|---|
| Metric | MSE | MAE | MSE | MAE | MSE | MAE | MSE | MAE | MSE | MAE | MSE | MAE | MSE | MAE | MSE | MAE | MSE | MAE | MSE | MAE | MSE | MAE | MSE | MAE |
| $ETTh1$ | **0.627** | **0.543** | 0.681 | 0.560 | 0.750 | 0.611 | 0.694 | 0.569 | 0.925 | 0.647 | 0.658 | 0.562 | 0.722 | 0.598 | 0.943 | 0.646 | 1.189 | 0.839 | 1.451 | 0.903 | 1.225 | 0.817 | 1.241 | 0.835 |
| $ETTh2$ | **0.382** | **0.418** | 0.400 | 0.433 | 0.694 | 0.577 | 0.827 | 0.615 | 0.439 | 0.448 | 0.463 | 0.454 | 0.441 | 0.457 | 0.470 | 0.489 | 0.809 | 0.681 | 3.206 | 1.268 | 3.922 | 1.653 | 3.527 | 1.472 |
| $ETTm1$ | 0.425 | 0.434 | 0.472 | 0.450 | **0.400** | **0.417** | 0.526 | 0.476 | 0.717 | 0.561 | 0.730 | 0.592 | 0.796 | 0.620 | 0.857 | 0.598 | 1.125 | 0.782 | 1.123 | 0.765 | 1.163 | 0.791 | 1.264 | 0.826 |
| $ETTm2$ | **0.274** | **0.323** | 0.308 | 0.346 | 0.399 | 0.426 | 0.314 | 0.352 | 0.344 | 0.372 | 0.381 | 0.404 | 0.388 | 0.433 | 0.341 | 0.372 | 0.534 | 0.547 | 1.415 | 0.871 | 3.658 | 1.489 | 3.581 | 1.487 |
| $Weather$ | **0.260** | 0.309 | 0.263 | **0.301** | 0.263 | 0.308 | 0.269 | 0.303 | 0.298 | 0.318 | 0.309 | 0.353 | 0.310 | 0.353 | 0.327 | 0.328 | 0.333 | 0.371 | 0.305 | 0.345 | 0.584 | 0.527 | 0.447 | 0.453 |
| $ECL$ | 0.179 | **0.268** | 0.178 | 0.273 | 0.176 | 0.275 | 0.181 | 0.277 | 0.402 | 0.453 | 0.266 | 0.353 | 0.346 | 0.404 | 0.627 | 0.603 | 0.800 | 0.685 | 0.878 | 0.725 | 1.281 | 0.929 | 1.289 | 0.904 |
| $Traffic$ | 0.423 | 0.298 | 0.434 | 0.305 | 0.450 | 0.317 | **0.418** | **0.296** | 0.867 | 0.493 | 0.676 | 0.423 | 0.833 | 0.502 | 1.526 | 0.839 | 1.859 | 0.927 | 1.557 | 0.795 | 1.591 | 0.832 | 1.618 | 0.851 |
| 1$^{st}$Count | **5** | | 2 | | 1 | | 1 | | 0 | | 0 | | 0 | | 0 | | 0 | | 0 | | 0 | | 0 | |

such as PatchTST, DLinear, and TimesNet, our average enhancements surpass **8%**, **12%**, and **33%** w.r.t. MSE. Analogous trends are discernible in the 5% few-shot learning scenarios, where our average advancement over GPT4TS exceeds **5%**. When compared with PatchTST, DLinear, and TimesNet, TIME-LLM manifests a striking average improvement of over **20%**.

## 4.4 ZERO-SHOT FORECASTING

**Setups.** Beyond few-shot learning, LLMs hold potential as effective zero-shot reasoners (Kojima et al., 2022). In this section, we evaluate the zero-shot learning capabilities of the reprogrammed LLM within the framework of cross-domain adaptation. Specifically, we examine how well a model performs on a dataset ♣ when it is optimized on another dataset ♠, where the model has not encountered any data samples from the dataset ♣.

Table 5: Zero-shot learning results. **Red**: the best, Blue: the second best. Appendix E shows our detailed results.

| Methods | TIME-LLM (Ours) | | GPT4TS (2023a) | | LLMTime (2023) | | DLinear (2023) | | PatchTST (2023) | | TimesNet (2023) | |
|---|---|---|---|---|---|---|---|---|---|---|---|---|
| Metric | MSE | MAE | MSE | MAE | MSE | MAE | MSE | MAE | MSE | MAE | MSE | MAE |
| $ETTh1 \to ETTh2$ | **0.353** | **0.387** | 0.406 | 0.422 | 0.992 | 0.708 | 0.493 | 0.488 | 0.380 | 0.405 | 0.421 | 0.431 |
| $ETTh1 \to ETTm2$ | **0.273** | **0.340** | 0.325 | 0.363 | 1.867 | 0.869 | 0.415 | 0.452 | 0.314 | 0.360 | 0.327 | 0.361 |
| $ETTh2 \to ETTh1$ | **0.479** | **0.474** | 0.757 | 0.578 | 1.961 | 0.981 | 0.703 | 0.574 | 0.565 | 0.513 | 0.865 | 0.621 |
| $ETTh2 \to ETTm2$ | **0.272** | **0.341** | 0.335 | 0.370 | 1.867 | 0.869 | 0.328 | 0.386 | 0.325 | 0.365 | 0.342 | 0.376 |
| $ETTm1 \to ETTh2$ | **0.381** | **0.412** | 0.433 | 0.439 | 0.992 | 0.708 | 0.464 | 0.475 | 0.439 | 0.438 | 0.457 | 0.454 |
| $ETTm1 \to ETTm2$ | **0.268** | **0.320** | 0.313 | 0.348 | 1.867 | 0.869 | 0.335 | 0.389 | 0.296 | 0.334 | 0.322 | 0.354 |
| $ETTm2 \to ETTh2$ | **0.354** | **0.400** | 0.435 | 0.443 | 0.992 | 0.708 | 0.455 | 0.471 | 0.409 | 0.425 | 0.435 | 0.443 |
| $ETTm2 \to ETTm1$ | **0.414** | **0.438** | 0.769 | 0.567 | 1.933 | 0.984 | 0.649 | 0.537 | 0.568 | 0.492 | 0.769 | 0.567 |

encountered any data samples from the dataset ♣. Similar to few-shot learning, we use long-term forecasting protocol and evaluate on various cross-domain scenarios utilizing the ETT datasets.

**Results.** Our brief results are in Tab. 5. TIME-LLM consistently outperforms the most competitive baselines by a large margin, over **14.2%** w.r.t. the second-best in MSE reduction. Considering the few-shot results, we observe that reprogramming an LLM tends to yield significantly better results in data scarcity scenarios. For example, our overall error reductions w.r.t. GPT4TS in 10% few-shot forecasting, 5% few-shot forecasting, and zero-shot forecasting are increasing gradually: **7.7%**, **8.4%**, and **22%**. Even when benchmarked against LLMTime, the most recent approach in this field, with the backbone LLM of comparable size (7B), TIME-LLM shows a substantial improvement exceeding **75%**. We attribute this to our approach being better at activating the LLM's knowledge transfer and reasoning capabilities in a resource-efficient manner when performing time series tasks.

## 4.5 MODEL ANALYSIS

**Language Model Variants.** We compare two representative backbones with varying capacities (**A.1-4** in Tab. 6). Our results indicate that the scaling law retain after the LLM reprogramming. We adopt Llama-7B by default in its full capacity, which manifestly outperforms its 1/4 capacity variant (**A.2**; inclusive of the first 8 Transformer layers) by **14.5%**. An average MSE reduction of **14.7%** is observed over GPT-2 (**A.3**), which slightly outperforms its variant GPT-2 (6) (**A.4**) by 2.7%.

**Cross-modality Alignment.** Our results in Tab. 6 indicate that ablating either patch reprogramming or Prompt-as-Prefix hurts knowledge transfer in reprogramming the LLM for effective time series forecasting. In the absence of representation alignment (**B.1**), we observe a notable average performance degradation of **9.2%**, which becomes more pronounced (exceeding **17%**) in few-shot tasks. In TIME-LLM, the act of prompting stands as a pivotal element in harnessing the LLM's capacity for understanding the inputs and tasks. Ablation of this component (**B.2**) results in over **8%** and **19%** degradation in standard and few-shot forecasting tasks, respectively. We find that removing the input statistics (**C.1**) hurts the most, resulting in an average increase of **10.2%** MSE. This is an-

Table 6: Ablations on ETTh1 and ETTm1 in predicting 96 and 192 steps ahead (MSE reported). **Red**: the best.

| Variant | Long-term Forecasting | | | | Few-shot Forecasting | | | |
|---|---|---|---|---|---|---|---|---|
| | ETTh1-96 | ETTh1-192 | ETTm1-96 | ETThm1-192 | ETTh1-96 | ETTh1-192 | ETTm1-96 | ETThm1-192 |
| **A.1** Llama (**Default**; 32) | **0.362** | **0.398** | **0.272** | **0.310** | **0.448** | **0.484** | **0.346** | **0.373** |
| **A.2** Llama (8) | 0.389 | 0.412 | 0.297 | 0.329 | 0.567 | 0.632 | 0.451 | 0.490 |
| **A.3** GPT-2 (12) | 0.385 | 0.419 | 0.306 | 0.332 | 0.548 | 0.617 | 0.447 | 0.509 |
| **A.4** GPT-2 (6) | 0.394 | 0.427 | 0.311 | 0.342 | 0.571 | 0.640 | 0.468 | 0.512 |
| **B.1** w/o Patch Reprogramming | 0.410 | 0.412 | 0.310 | 0.342 | 0.498 | 0.570 | 0.445 | 0.487 |
| **B.2** w/o Prompt-as-Prefix | 0.398 | 0.423 | 0.298 | 0.339 | 0.521 | 0.617 | 0.432 | 0.481 |
| **C.1** w/o Dataset Context | 0.402 | 0.417 | 0.298 | 0.331 | 0.491 | 0.538 | 0.392 | 0.447 |
| **C.2** w/o Task Instruction | 0.388 | 0.420 | 0.285 | 0.327 | 0.476 | 0.529 | 0.387 | 0.439 |
| **C.3** w/o Statistical Context | 0.391 | 0.419 | 0.279 | 0.347 | 0.483 | 0.547 | 0.421 | 0.461 |

Table 7: Efficiency analysis of TIME-LLM on ETTh1 in forecasting different steps ahead.

| Length | ETTh1-96 | | | ETTh1-192 | | | ETTh1-336 | | | ETTh1-512 | | |
|---|---|---|---|---|---|---|---|---|---|---|---|---|
| Metric | Param. (M) | Mem. (MiB) | Speed(s/iter) | Param. (M) | Mem. (MiB) | Speed(s/iter) | Param. (M) | Mem. (MiB) | Speed(s/iter) | Param. (M) | Mem.(MiB) | Speed(s/iter) |
| **D.1** LLama (32) | 3404.53 | 32136 | 0.517 | 3404.57 | 33762 | 0.582 | 3404.62 | 37988 | 0.632 | 3404.69 | 39004 | 0.697 |
| **D.2** LLama (8) | 975.83 | 11370 | 0.184 | 975.87 | 12392 | 0.192 | 975.92 | 13188 | 0.203 | 976.11 | 13616 | 0.217 |
| **D.3** w/o LLM | 6.39 | 3678 | 0.046 | 6.42 | 3812 | 0.087 | 6.48 | 3960 | 0.093 | 6.55 | 4176 | 0.129 |

ticipated as external knowledge can be naturally incorporated via prompting to facilitate the learning and inference. Additionally, providing the LLM with clear task instructions and input context (e.g., dataset captioning) is also beneficial (i.e., **C.2** and **C.1**; eliciting over **7.7%** and **9.6%**, respectively).

**Reprogramming Interpretation.** We provide a case study on ETTh1 of reprogramming 48 time series patches with 100 text prototypes in Fig. 5. The top 4 subplots visualize the optimization of reprogramming space from randomly-initialized (**a**) to well-optimized (**d**). We find only a small set of prototypes (columns) participated in reprogramming the input patches (rows) in subplot (**e**). Also, patches undergo different representations through varying combinations of prototypes. This indicates: (1) text prototypes learn to summarize language cues, and a select few are highly relevant for representing information in local time series patches, which we visualize by

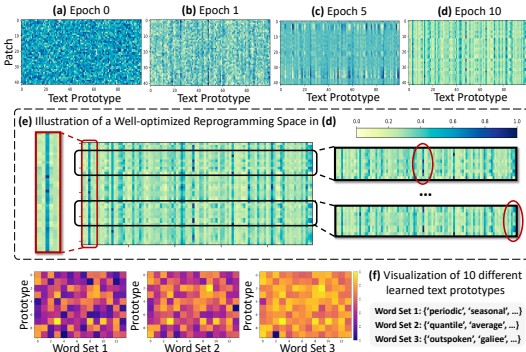

Figure 5: A showcase of patch reprogramming.

randomly selecting 10 in subplot (**f**). Our results suggest a high relevance to the words that describe time series properties (i.e., word sets 1 and 2); (2) patches usually have different underlying semantics, necessitating different prototypes to represent.

**Reprogramming Efficiency.** Tab. 7 provides an overall efficiency analysis of TIME-LLM with and without the backbone LLM. Our proposed reprogramming network itself (**D.3**) is lightweight in activating the LLM's ability for time series forecasting (i.e., fewer than 6.6 million *trainable* parameters; only around **0.2%** of the total parameters in Llama-7B), and the overall efficiency of TIME-LLM is actually capped by the leveraged backbones (e.g., **D.1** and **D.2**). This is favorable even compared to the parameter-efficient fine-tuning methods (e.g., QLoRA (Dettmers et al., 2023)) in balancing task performance and efficiency.

## 5 CONCLUSION AND FUTURE WORK

TIME-LLM shows promise in adapting frozen large language models for time series forecasting by reprogramming time series data into text prototypes more natural for LLMs and providing natural language guidance via Prompt-as-Prefix to augment reasoning. Evaluations demonstrate the adapted LLMs can outperform specialized expert models, indicating their potential as effective time series machines. Our results also provide a novel insight that time series forecasting can be cast as yet another "language" task that can be tackled by an off-the-shelf LLM to achieve state-of-the-art performance through our Time-LLM framework. Further research should explore optimal reprogramming representations, enrich LLMs with explicit time series knowledge through continued pre-training, and build towards multimodal models with joint reasoning across time series, natural language, and other modalities. Furthermore, applying the reprogramming framework to equip LLMs with broader time series analytical abilities or other new capabilities should also be considered.

ACKNOWLEDGMENTS AND DISCLOSURE OF FUNDING

The authors extend their deep gratitude to the Intelligent Engine Technology Division of Ant Group for their support in completing this research. We also express special thanks to the Language and Machine Intelligence Department and the Optimization Intelligence Department for their support. Our sincere appreciation is directed to Jun Zhou, Vice President of the Intelligent Engine Technology Division, and Xingyu Lu, Senior Staff Engineer, for their expert guidance.

This material is based on research partially sponsored by the DARPA Assured Neuro Symbolic Learning and Reasoning (ANSR) program under award number FA8750-23-2-1016 and the DARPA Knowledge Management at Scale and Speed (KMASS) program under award number HR00112220047. S. Pan was supported in part by the Australian Research Council (ARC) under grants FT210100097 and DP240101547, and the CSIRO – National Science Foundation (US) AI Research Collaboration Program. Y. Liang was supported by the Guangzhou-HKUST(GZ) Joint Funding Program (No. 2024A03J0620).

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

## A   MORE RELATED WORK

**Task-specific Learning.**  We furnish an extension of the related work on task-specific learning, focusing particularly on the most related models to which we made comparisons.  Recent works improve Transformer (Vaswani et al., 2017) for time series forecasting by incorporating signal processing principles like patching, exponential smoothing, decomposition, and frequency analysis. For example, PatchTST (Nie et al., 2023) segments time series into patches as input tokens to Transformer.  This retains local semantics, reduces computation/memory for attention, and allows longer history.  It improves long-term forecast accuracy over other Transformer models.  It also achieves excellent performance on self-supervised pretraining and transfer learning.  ETSformer (Woo et al., 2022) incorporates exponential smoothing principles into Transformer attention to improve accuracy and efficiency.  It uses exponential smoothing attention and frequency attention to replace standard self-attention.  FEDformer (Zhou et al., 2022) combines Transformer with seasonal-trend decomposition.  The decomposition captures the global profile while Transformer captures detailed structures. It also uses frequency enhancement for long-term prediction.  This provides better performance and efficiency than the standard Transformer.  Autoformer (Wu et al., 2021) uses a decomposition architecture with auto-correlation to enable progressive decomposition capacities for complex series. Auto-correlation is designed based on series periodicity to conduct dependency discovery and representation aggregation.  It outperforms self-attention in efficiency and accuracy.

Although these methods enhance efficiency and accuracy compared to vanilla Transformer, they are mostly designed and optimized for narrow prediction tasks within specific domains.  These models are typically trained end-to-end on small, domain-specific datasets.  While achieving strong performance on their target tasks, such specialized models sacrifice versatility and generalizability across the diverse range of time series data encountered in the real world.  The narrow focus limits their applicability to new datasets and tasks.  To advance time series forecasting, there is a need for more flexible, widely applicable models that can adapt to new data distributions and tasks without extensive retraining.  Ideal models would learn robust time series representations that transfer knowledge across domains.  Developing such broadly capable forecasting models remains an open challenge. According to our discussions of related previous work, recent studies have begun to explore model versatility through pre-training and architectural innovations.  However, further efforts are needed to realize the truly general-purpose forecasting systems that we are advancing in this research.

**Cross-modality Adaptation.** We provide an extended overview of related work in cross-modality adaptation, with a particular focus on recent advancements in model reprogramming for time series and other data modalities.  Model reprogramming is a resource-efficient cross-domain learning approach that involves adapting a well-developed, pre-trained model from one domain (source) to address tasks in a different domain (target) without the need for model fine-tuning, even when these domains are significantly distinct, as noted by Chen (2022).  In the context of time series data, Voice2Series (Yang et al., 2021) adapts an acoustic model from speech recognition for time series classification by transforming the time series to fit the model and remapping outputs to new labels. Similarly, LLMTime (Gruver et al., 2023) adapts LLMs for zero-shot time series forecasting, focusing on the effective tokenization of input time series for the backbone LLM, which then generates forecasts autoregressively.  Diverging from these methods, TIME-LLM does not edit the input time series directly.  Instead, it proposes reprogramming time series with the source data modality along with prompting to unleash the full potential of LLMs as versatile forecasters in standard, few-shot, and zero-shot scenarios.  Other notable works in this field, mostly in biology, include R2DL (Vinod et al., 2020) and ReproBert (Melnyk et al., 2023), which reprogram amino acids using word embeddings.  A key distinction with our patch reprogramming approach is that, unlike the complete set of amino acids, time series patches do not form a complete set.  Thus, we propose optimizing a small set of text prototypes and their mapping to time series patches, rather than directly optimizing a large transformation matrix between two complete sets, such as vocabulary and amino acids.

## B   EXPERIMENTAL DETAILS

### B.1   IMPLEMENTATION

We mainly follow the experimental configurations in (Wu et al., 2023) across all baselines within a unified evaluation pipeline in `https://github.com/thuml/Time-Series-Library` for

fair comparisons. We use Llama-7B (Touvron et al., 2023) as the default backbone model unless stated otherwise. All our experiments are repeated three times and we report the averaged results. Our model implementation is on PyTorch (Paszke et al., 2019) with all experiments conducted on NVIDIA A100-80G GPUs. Our detailed model configurations are in Appendix B.4, and our code is made available at `https://github.com/KimMeen/Time-LLM`.

**Technical Details.** We provide additional technical details of TIME-LLM in three aspects: (1) the learning of text prototypes, (2) the calculation of trends and lags in time series for use in prompts, and (3) the implementation of the output projection. To identify a small set of text prototypes $\mathbf{E}' \in \mathbb{R}^{V' \times D}$ from $\mathbf{E} \in \mathbb{R}^{V \times D}$, we learn a matrix $\mathbf{W} \in \mathbb{R}^{V' \times V}$ as the intermediary. To describe the overall time series trend in natural language, we calculate the sum of differences between consecutive time steps. A sum greater than 0 indicates an upward trend, while a lesser sum denotes a downward trend. In addition, we calculate the top-5 lags of the time series, identified by computing the autocorrelation using fast Fourier transformation and selecting the five lags with the highest correlation values. After we pack and feedforward the prompt and patch embeddings $\mathbf{O}^{(i)} \in \mathbb{R}^{P \times D}$ through the frozen LLM, we discard the prefixal part and obtain the output representations, denoted as $\tilde{\mathbf{O}}^i \in \mathbb{R}^{P \times D}$. Subsequently, we follow PatchTST (Nie et al., 2023) and flatten $\tilde{\mathbf{O}}^i$ into a 1D tensor with the length $P \times D$, which is then linear projected as $\hat{\mathbf{Y}}^i \in \mathbb{R}^H$.

## B.2 DATASET DETAILS

Dataset statistics are summarized in Tab. 8. We evaluate the long-term forecasting performance on the well-established eight different benchmarks, including four ETT datasets (Zhou et al., 2021) (i.e., ETTh1, ETTh2, ETTm1, and ETTm2), Weather, Electricity, Traffic, and ILI from (Wu et al., 2023). Furthermore, we evaluate the performance of short-term forecasting on the M4 benchmark (Makridakis et al., 2018) and the quarterly dataset in the M3 benchmark (Makridakis & Hibon, 2000).

Table 8: Dataset statistics are from (Wu et al., 2023). The dimension indicates the number of time series (i.e., channels), and the dataset size is organized in (training, validation, testing).

| Tasks | Dataset | Dim. | Series Length | Dataset Size | Frequency | Domain |
|---|---|---|---|---|---|---|
| Long-term Forecasting | ETTm1 | 7 | {96, 192, 336, 720} | (34465, 11521, 11521) | 15 min | Temperature |
| | ETTm2 | 7 | {96, 192, 336, 720} | (34465, 11521, 11521) | 15 min | Temperature |
| | ETTh1 | 7 | {96, 192, 336, 720} | (8545, 2881, 2881) | 1 hour | Temperature |
| | ETTh2 | 7 | {96, 192, 336, 720} | (8545, 2881, 2881) | 1 hour | Temperature |
| | Electricity | 321 | {96, 192, 336, 720} | (18317, 2633, 5261) | 1 hour | Electricity |
| | Traffic | 862 | {96, 192, 336, 720} | (12185, 1757, 3509) | 1 hour | Transportation |
| | Weather | 21 | {96, 192, 336, 720} | (36792, 5271, 10540) | 10 min | Weather |
| | ILI | 7 | {24, 36, 48, 60} | (617, 74, 170) | 1 week | Illness |
| Short-term Forecasting | M3-Quarterly | 1 | 8 | (756, 0, 756) | Quarterly | Multiple |
| | M4-Yearly | 1 | 6 | (23000, 0, 23000) | Yearly | Demographic |
| | M4-Quarterly | 1 | 8 | (24000, 0, 24000) | Quarterly | Finance |
| | M4-Monthly | 1 | 18 | (48000, 0, 48000) | Monthly | Industry |
| | M4-Weakly | 1 | 13 | (359, 0, 359) | Weakly | Macro |
| | M4-Daily | 1 | 14 | (4227, 0, 4227) | Daily | Micro |
| | M4-Hourly | 1 | 48 | (414, 0, 414) | Hourly | Other |

The Electricity Transformer Temperature (ETT; An indicator reflective of long-term electric power deployment) benchmark is comprised of two years of data, sourced from two counties in China, and is subdivided into four distinct datasets, each with varying sampling rates: ETTh1 and ETTh2, which are sampled at a 1-hour level, and ETTm1 and ETTm2, which are sampled at a 15-minute level. Each entry within the ETT datasets includes six power load features and a target variable, termed "oil temperature". The Electricity dataset comprises records of electricity consumption from 321 customers, measured at a 1-hour sampling rate. The Weather dataset includes one-year records

from 21 meteorological stations located in Germany, with a sampling rate of 10 minutes. The Traffic dataset includes data on the occupancy rates of the freeway system, recorded from 862 sensors across the State of California, with a sampling rate of 1 hour. The influenza-like illness (ILI) dataset contains records of patients experiencing severe influenza with complications.

The M4 benchmark comprises 100K time series, amassed from various domains commonly present in business, financial, and economic forecasting. These time series have been partitioned into six distinctive datasets, each with varying sampling frequencies that range from yearly to hourly. The M3-Quarterly dataset comprises 756 quarterly sampled time series in the M3 benchmark. These series are categorized into five different domains: demographic, micro, macro, industry, and finance.

## B.3 EVALUATION METRICS

For evaluation metrics, we utilize the mean square error (MSE) and mean absolute error (MAE) for long-term forecasting. In terms of the short-term forecasting on M4 benchmark, we adopt the symmetric mean absolute percentage error (SMAPE), mean absolute scaled error (MASE), and overall weighted average (OWA) as in N-BEATS (Oreshkin et al., 2020). Note that OWA is a specific metric utilized in the M4 competition. The calculations of these metrics are as follows:

$$\text{MSE} = \frac{1}{H} \sum_{h=1}^{T} (\mathbf{Y}_h - \hat{\mathbf{Y}}_h)^2, \qquad \text{MAE} = \frac{1}{H} \sum_{h=1}^{H} |\mathbf{Y}_h - \hat{\mathbf{Y}}_h|,$$

$$\text{SMAPE} = \frac{200}{H} \sum_{h=1}^{H} \frac{|\mathbf{Y}_h - \hat{\mathbf{Y}}_h|}{|\mathbf{Y}_h| + |\hat{\mathbf{Y}}_h|}, \qquad \text{MAPE} = \frac{100}{H} \sum_{h=1}^{H} \frac{|\mathbf{Y}_h - \hat{\mathbf{Y}}_h|}{|\mathbf{Y}_h|},$$

$$\text{MASE} = \frac{1}{H} \sum_{h=1}^{H} \frac{|\mathbf{Y}_h - \hat{\mathbf{Y}}_h|}{\frac{1}{H-s} \sum_{j=s+1}^{H} |\mathbf{Y}_j - \mathbf{Y}_{j-s}|}, \qquad \text{OWA} = \frac{1}{2} \left[ \frac{\text{SMAPE}}{\text{SMAPE}_{\text{Naïve2}}} + \frac{\text{MASE}}{\text{MASE}_{\text{Naïve2}}} \right],$$

where $s$ is the periodicity of the time series data. $H$ denotes the number of data points (i.e., prediction horizon in our cases). $\mathbf{Y}_h$ and $\hat{\mathbf{Y}}_h$ are the $h$-th ground truth and prediction where $h \in \{1, \cdots, H\}$.

## B.4 MODEL CONFIGURATIONS

The configurations of our models, relative to varied tasks and datasets, are consolidated in Tab. 9. By default, the Adam optimizer (Kingma & Ba, 2015) is employed throughout all experiments. Specifically, the quantity of text prototypes $V'$ is held constant at 100 and 1000 for short-term and long-term forecasting tasks, respectively. We utilize the Llama-7B model at full capacity, maintaining the backbone model layers at 32 across all tasks as a standard. The term input length $T$ signifies the number of time steps present in the original input time series data. Patch dimensions $d_m$ represent the hidden dimensions of the embedded time series patches prior to reprogramming. Lastly, heads $K$ correlate to the multi-head cross-attention utilized for patch reprogramming. In the four rightmost columns of Tab. 9, we detail the configurations related to model training.

Table 9: An overview of the experimental configurations for TIME-LLM. "LTF" and "STF" denote long-term and short-term forecasting, respectively.

| Task-Dataset / Configuration | Model Hyperparameter | | | | | Training Process | | | |
|---|---|---|---|---|---|---|---|---|---|
| | Text Prototype $V'$ | Backbone Layers | Input Length $T$ | Patch Dim. $d_m$ | Heads $K$ | LR* | Loss | Batch Size | Epochs |
| LTF - ETTh1 | 1000 | 32 | 512 | 16 | 8 | $10^{-3}$ | MSE | 16 | 50 |
| LTF - ETTh2 | 1000 | 32 | 512 | 16 | 8 | $10^{-3}$ | MSE | 16 | 50 |
| LTF - ETTm1 | 1000 | 32 | 512 | 16 | 8 | $10^{-3}$ | MSE | 16 | 100 |
| LTF - ETTm2 | 1000 | 32 | 512 | 16 | 8 | $10^{-3}$ | MSE | 16 | 100 |
| LTF - Weather | 1000 | 32 | 512 | 16 | 8 | $10^{-2}$ | MSE | 8 | 100 |
| LTF - Electricity | 1000 | 32 | 512 | 16 | 8 | $10^{-2}$ | MSE | 8 | 100 |
| LTF - Traffic | 1000 | 32 | 512 | 16 | 8 | $10^{-2}$ | MSE | 8 | 100 |
| LTF - ILI | 100 | 32 | 96 | 16 | 8 | $10^{-2}$ | MSE | 16 | 50 |
| STF - M3-Quarterly | 100 | 32 | $2 \times H$ † | 32 | 8 | $10^{-4}$ | SMAPE | 32 | 50 |
| STF - M4 | 100 | 32 | $2 \times H$ † | 32 | 8 | $10^{-4}$ | SMAPE | 32 | 50 |

† $H$ represents the forecasting horizon of the M4 and M3 datasets.
∗ LR means the initial learning rate.

## C  HYPERPARAMETER SENSITIVITY

We conduct a hyperparameter sensitivity analysis focusing on the four important hyperparameters within TIME-LLM: namely, the number of backbone model layers, the number of text prototypes $V'$, the time series input length $T$, and the number of patch reprogramming cross-attention heads $K$. The correlated results can be found in Fig. 6. From our analysis, we derive the following observations: (1) There is a positive correlation between the number of Transformer layers in the backbone LLM and the performance of TIME-LLM, affirming that the scaling law is preserved post-LLM reprogramming.; (2) Generally, acquiring more text prototypes enhances performance. We hypothesize that a limited number of prototypes $V'$ might induce noise when aggregating language cues, consequently obstructing the efficient learning of highly representative prototypes essential for characterizing the input time series patches; (3) The input time length $T$ exhibits a direct relation with forecasting accuracy, particularly evident when predicting extended horizons. This observation is logical and is in congruence with conventional time series models; (4) Increasing the number of attention heads during the reprogramming of input patches proves to be advantageous.

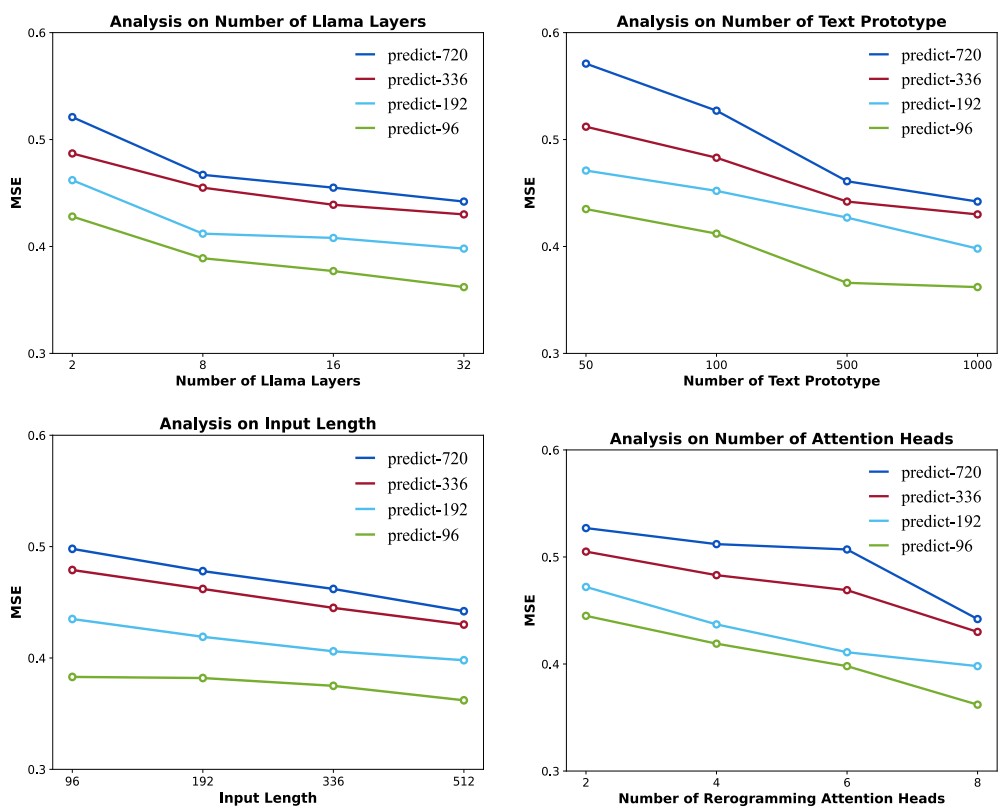

Figure 6: Analysis of hyperparameter sensitivity on ETTh1 dataset.

## D  LONG-TERM AND SHORT-TERM FORECASTING

### D.1  LONG-TERM FORECASTING

By solely reprogramming the smallest Llama model while keeping it intact, TIME-LLM attains SOTA performance in **36** out of 40 instances across eight time series benchmarks. This underscores the considerable potential of LLMs as robust and reliable time series forecasters. Furthermore, we benchmark the proposed method against other well-established baselines in Tab. 11. This comparison includes three notable statistical methods (AutoARIMA, AutoTheta, and AutoETS) (Herzen et al., 2022) and two recent time series models, N-HiTS (Challu et al., 2023b) and N-BEATS (Oreshkin et al., 2020). Remarkably, TIME-LLM secures SOTA performance across all cases, surpassing the second-best results by significant margins of over **22%** and **16%** in terms of MSE and MAE.

Table 10: Full long-term forecasting results. We set the forecasting horizons $H \in \{24, 36, 48, 60\}$ for ILI and $\{96, 192, 336, 720\}$ for the others. A lower value indicates better performance. **Red**: the best, Blue: the second best.

| Methods | | TIME-LLM | | GPT4TS | | DLinear | | PatchTST | | TimesNet | | FEDformer | | Autoformer | | Stationary | | ETSformer | | LightTS | | Informer | | Reformer | |
|---|---|---|---|---|---|---|---|---|---|---|---|---|---|---|---|---|---|---|---|---|---|---|---|---|---|
| Metric | | MSE | MAE | MSE | MAE | MSE | MAE | MSE | MAE | MSE | MAE | MSE | MAE | MSE | MAE | MSE | MAE | MSE | MAE | MSE | MAE | MSE | MAE | MSE | MAE |
| ETTh1 | 96 | 0.362 | 0.392 | 0.376 | 0.397 | 0.375 | 0.399 | 0.370 | 0.399 | 0.384 | 0.402 | 0.376 | 0.419 | 0.449 | 0.459 | 0.513 | 0.491 | 0.494 | 0.479 | 0.424 | 0.432 | 0.865 | 0.713 | 0.837 | 0.728 |
| | 192 | 0.398 | 0.418 | 0.416 | 0.418 | 0.405 | 0.416 | 0.413 | 0.421 | 0.436 | 0.429 | 0.420 | 0.448 | 0.500 | 0.482 | 0.534 | 0.504 | 0.538 | 0.504 | 0.475 | 0.462 | 1.008 | 0.792 | 0.923 | 0.766 |
| | 336 | 0.430 | 0.427 | 0.442 | 0.433 | 0.439 | 0.443 | 0.422 | 0.436 | 0.491 | 0.469 | 0.459 | 0.465 | 0.521 | 0.496 | 0.588 | 0.535 | 0.574 | 0.521 | 0.518 | 0.488 | 1.107 | 0.809 | 1.097 | 0.835 |
| | 720 | 0.442 | 0.457 | 0.477 | 0.456 | 0.472 | 0.490 | 0.447 | 0.466 | 0.521 | 0.500 | 0.506 | 0.507 | 0.514 | 0.512 | 0.643 | 0.616 | 0.562 | 0.535 | 0.547 | 0.533 | 1.181 | 0.865 | 1.257 | 0.889 |
| | Avg | 0.408 | 0.423 | 0.465 | 0.455 | 0.422 | 0.437 | 0.413 | 0.430 | 0.458 | 0.450 | 0.440 | 0.460 | 0.496 | 0.487 | 0.570 | 0.537 | 0.542 | 0.510 | 0.491 | 0.479 | 1.040 | 0.795 | 1.029 | 0.805 |
| ETTh2 | 96 | 0.268 | 0.328 | 0.285 | 0.342 | 0.289 | 0.353 | 0.274 | 0.336 | 0.340 | 0.374 | 0.358 | 0.397 | 0.346 | 0.388 | 0.476 | 0.458 | 0.340 | 0.391 | 0.397 | 0.437 | 3.755 | 1.525 | 2.626 | 1.317 |
| | 192 | 0.329 | 0.375 | 0.354 | 0.389 | 0.383 | 0.418 | 0.339 | 0.379 | 0.402 | 0.414 | 0.429 | 0.439 | 0.456 | 0.452 | 0.512 | 0.493 | 0.430 | 0.439 | 0.520 | 0.504 | 5.602 | 1.931 | 11.12 | 2.979 |
| | 336 | 0.368 | 0.409 | 0.373 | 0.407 | 0.448 | 0.465 | 0.329 | 0.380 | 0.452 | 0.452 | 0.496 | 0.487 | 0.482 | 0.486 | 0.552 | 0.551 | 0.485 | 0.479 | 0.626 | 0.559 | 4.721 | 1.835 | 9.323 | 2.769 |
| | 720 | 0.372 | 0.420 | 0.406 | 0.441 | 0.605 | 0.551 | 0.379 | 0.422 | 0.462 | 0.468 | 0.463 | 0.474 | 0.515 | 0.511 | 0.562 | 0.560 | 0.500 | 0.497 | 0.863 | 0.672 | 3.647 | 1.625 | 3.874 | 1.697 |
| | Avg | 0.334 | 0.383 | 0.381 | 0.412 | 0.431 | 0.446 | 0.330 | 0.379 | 0.414 | 0.427 | 0.437 | 0.449 | 0.450 | 0.459 | 0.526 | 0.516 | 0.439 | 0.452 | 0.602 | 0.543 | 4.431 | 1.729 | 6.736 | 2.191 |
| ETTm1 | 96 | 0.272 | 0.334 | 0.292 | 0.346 | 0.299 | 0.343 | 0.290 | 0.342 | 0.338 | 0.375 | 0.379 | 0.419 | 0.505 | 0.475 | 0.386 | 0.398 | 0.375 | 0.398 | 0.374 | 0.400 | 0.672 | 0.571 | 0.538 | 0.528 |
| | 192 | 0.310 | 0.358 | 0.332 | 0.372 | 0.335 | 0.365 | 0.332 | 0.369 | 0.374 | 0.387 | 0.426 | 0.441 | 0.553 | 0.496 | 0.459 | 0.444 | 0.408 | 0.410 | 0.400 | 0.407 | 0.795 | 0.669 | 0.658 | 0.592 |
| | 336 | 0.352 | 0.384 | 0.366 | 0.394 | 0.369 | 0.386 | 0.366 | 0.392 | 0.410 | 0.411 | 0.445 | 0.459 | 0.621 | 0.537 | 0.495 | 0.464 | 0.435 | 0.428 | 0.438 | 0.438 | 1.212 | 0.871 | 0.898 | 0.721 |
| | 720 | 0.383 | 0.411 | 0.417 | 0.421 | 0.425 | 0.421 | 0.416 | 0.420 | 0.478 | 0.450 | 0.543 | 0.490 | 0.671 | 0.561 | 0.585 | 0.516 | 0.499 | 0.462 | 0.527 | 0.502 | 1.166 | 0.823 | 1.102 | 0.841 |
| | Avg | 0.329 | 0.372 | 0.388 | 0.403 | 0.357 | 0.378 | 0.351 | 0.380 | 0.400 | 0.406 | 0.448 | 0.452 | 0.588 | 0.517 | 0.481 | 0.456 | 0.429 | 0.425 | 0.435 | 0.437 | 0.961 | 0.734 | 0.799 | 0.671 |
| ETTm2 | 96 | 0.161 | 0.253 | 0.173 | 0.262 | 0.167 | 0.269 | 0.165 | 0.255 | 0.187 | 0.267 | 0.203 | 0.287 | 0.255 | 0.339 | 0.192 | 0.274 | 0.189 | 0.280 | 0.209 | 0.308 | 0.365 | 0.453 | 0.658 | 0.619 |
| | 192 | 0.219 | 0.293 | 0.229 | 0.301 | 0.224 | 0.303 | 0.220 | 0.292 | 0.249 | 0.309 | 0.269 | 0.328 | 0.281 | 0.340 | 0.280 | 0.339 | 0.253 | 0.319 | 0.311 | 0.382 | 0.533 | 0.563 | 1.078 | 0.827 |
| | 336 | 0.271 | 0.329 | 0.286 | 0.341 | 0.281 | 0.342 | 0.274 | 0.329 | 0.321 | 0.351 | 0.325 | 0.366 | 0.339 | 0.372 | 0.334 | 0.361 | 0.314 | 0.357 | 0.442 | 0.466 | 1.363 | 0.887 | 1.549 | 0.972 |
| | 720 | 0.352 | 0.379 | 0.378 | 0.401 | 0.397 | 0.421 | 0.362 | 0.385 | 0.408 | 0.403 | 0.421 | 0.415 | 0.433 | 0.432 | 0.417 | 0.413 | 0.414 | 0.413 | 0.675 | 0.587 | 3.379 | 1.338 | 2.631 | 1.242 |
| | Avg | 0.251 | 0.313 | 0.284 | 0.339 | 0.267 | 0.333 | 0.255 | 0.315 | 0.291 | 0.333 | 0.305 | 0.349 | 0.327 | 0.371 | 0.306 | 0.347 | 0.293 | 0.342 | 0.409 | 0.436 | 1.410 | 0.810 | 1.479 | 0.915 |
| Weather | 96 | 0.147 | 0.201 | 0.162 | 0.212 | 0.176 | 0.237 | 0.149 | 0.198 | 0.172 | 0.220 | 0.217 | 0.296 | 0.266 | 0.336 | 0.173 | 0.223 | 0.197 | 0.281 | 0.182 | 0.242 | 0.300 | 0.384 | 0.689 | 0.596 |
| | 192 | 0.189 | 0.234 | 0.204 | 0.248 | 0.220 | 0.282 | 0.194 | 0.241 | 0.219 | 0.261 | 0.276 | 0.336 | 0.307 | 0.367 | 0.245 | 0.285 | 0.237 | 0.312 | 0.227 | 0.287 | 0.598 | 0.544 | 0.752 | 0.638 |
| | 336 | 0.262 | 0.279 | 0.254 | 0.286 | 0.265 | 0.319 | 0.245 | 0.282 | 0.280 | 0.306 | 0.339 | 0.380 | 0.359 | 0.395 | 0.321 | 0.338 | 0.298 | 0.353 | 0.282 | 0.334 | 0.578 | 0.523 | 0.639 | 0.596 |
| | 720 | 0.304 | 0.316 | 0.326 | 0.337 | 0.333 | 0.362 | 0.314 | 0.334 | 0.365 | 0.359 | 0.403 | 0.428 | 0.419 | 0.428 | 0.414 | 0.410 | 0.352 | 0.288 | 0.352 | 0.386 | 1.059 | 0.741 | 1.130 | 0.792 |
| | Avg | 0.225 | 0.257 | 0.237 | 0.270 | 0.248 | 0.300 | 0.225 | 0.264 | 0.259 | 0.287 | 0.309 | 0.360 | 0.338 | 0.382 | 0.288 | 0.314 | 0.271 | 0.334 | 0.261 | 0.312 | 0.634 | 0.548 | 0.803 | 0.656 |
| Electricity | 96 | 0.131 | 0.224 | 0.139 | 0.238 | 0.140 | 0.237 | 0.129 | 0.222 | 0.168 | 0.272 | 0.193 | 0.308 | 0.201 | 0.317 | 0.169 | 0.273 | 0.187 | 0.304 | 0.207 | 0.307 | 0.274 | 0.368 | 0.312 | 0.402 |
| | 192 | 0.152 | 0.241 | 0.153 | 0.251 | 0.153 | 0.249 | 0.157 | 0.240 | 0.184 | 0.289 | 0.201 | 0.315 | 0.222 | 0.334 | 0.182 | 0.286 | 0.199 | 0.315 | 0.213 | 0.316 | 0.296 | 0.386 | 0.348 | 0.433 |
| | 336 | 0.160 | 0.248 | 0.169 | 0.266 | 0.169 | 0.267 | 0.163 | 0.259 | 0.198 | 0.300 | 0.214 | 0.329 | 0.231 | 0.338 | 0.200 | 0.304 | 0.212 | 0.329 | 0.230 | 0.333 | 0.300 | 0.394 | 0.350 | 0.433 |
| | 720 | 0.192 | 0.298 | 0.206 | 0.297 | 0.203 | 0.301 | 0.197 | 0.290 | 0.220 | 0.320 | 0.246 | 0.355 | 0.254 | 0.361 | 0.222 | 0.321 | 0.233 | 0.345 | 0.265 | 0.360 | 0.373 | 0.439 | 0.340 | 0.420 |
| | Avg | 0.158 | 0.252 | 0.167 | 0.263 | 0.166 | 0.263 | 0.161 | 0.252 | 0.192 | 0.295 | 0.214 | 0.327 | 0.227 | 0.338 | 0.193 | 0.296 | 0.208 | 0.323 | 0.229 | 0.329 | 0.311 | 0.397 | 0.338 | 0.422 |
| Traffic | 96 | 0.362 | 0.248 | 0.388 | 0.282 | 0.410 | 0.282 | 0.360 | 0.249 | 0.593 | 0.321 | 0.587 | 0.366 | 0.613 | 0.388 | 0.612 | 0.338 | 0.607 | 0.392 | 0.615 | 0.391 | 0.719 | 0.391 | 0.732 | 0.423 |
| | 192 | 0.374 | 0.247 | 0.407 | 0.290 | 0.423 | 0.287 | 0.379 | 0.256 | 0.617 | 0.336 | 0.604 | 0.373 | 0.616 | 0.382 | 0.613 | 0.340 | 0.621 | 0.399 | 0.601 | 0.382 | 0.696 | 0.379 | 0.733 | 0.420 |
| | 336 | 0.385 | 0.271 | 0.412 | 0.294 | 0.436 | 0.296 | 0.392 | 0.264 | 0.629 | 0.336 | 0.621 | 0.383 | 0.622 | 0.337 | 0.618 | 0.328 | 0.622 | 0.396 | 0.613 | 0.386 | 0.777 | 0.420 | 0.742 | 0.420 |
| | 720 | 0.430 | 0.288 | 0.450 | 0.312 | 0.466 | 0.315 | 0.432 | 0.286 | 0.640 | 0.350 | 0.626 | 0.382 | 0.660 | 0.408 | 0.653 | 0.355 | 0.632 | 0.396 | 0.658 | 0.407 | 0.864 | 0.472 | 0.755 | 0.423 |
| | Avg | 0.388 | 0.264 | 0.414 | 0.294 | 0.433 | 0.295 | 0.390 | 0.263 | 0.620 | 0.336 | 0.610 | 0.376 | 0.628 | 0.379 | 0.624 | 0.340 | 0.621 | 0.396 | 0.622 | 0.392 | 0.764 | 0.416 | 0.741 | 0.422 |
| ILI | 24 | 1.285 | 0.727 | 2.063 | 0.881 | 2.215 | 1.081 | 1.319 | 0.754 | 2.317 | 0.934 | 3.228 | 1.260 | 3.483 | 1.287 | 2.294 | 0.945 | 2.527 | 1.020 | 8.313 | 2.144 | 5.764 | 1.677 | 4.400 | 1.382 |
| | 36 | 1.404 | 0.814 | 1.868 | 0.892 | 1.963 | 0.963 | 1.430 | 0.834 | 1.972 | 0.920 | 2.679 | 1.080 | 3.103 | 1.148 | 1.825 | 0.848 | 2.615 | 1.007 | 6.631 | 1.902 | 4.755 | 1.467 | 4.783 | 1.448 |
| | 48 | 1.523 | 0.807 | 1.790 | 0.884 | 2.130 | 1.024 | 1.553 | 0.815 | 2.238 | 0.940 | 2.622 | 1.078 | 2.669 | 1.085 | 2.010 | 0.900 | 2.359 | 0.972 | 7.299 | 1.982 | 4.763 | 1.469 | 4.832 | 1.465 |
| | 60 | 1.531 | 0.854 | 1.979 | 0.957 | 2.368 | 1.096 | 1.470 | 0.788 | 2.027 | 0.928 | 2.857 | 1.157 | 2.770 | 1.125 | 2.178 | 0.963 | 2.487 | 1.016 | 7.283 | 1.985 | 5.264 | 1.564 | 4.882 | 1.483 |
| | Avg | 1.435 | 0.801 | 1.925 | 0.903 | 2.169 | 1.041 | 1.443 | 0.797 | 2.139 | 0.931 | 2.847 | 1.144 | 3.006 | 1.161 | 2.077 | 0.914 | 2.497 | 1.004 | 7.382 | 2.003 | 5.137 | 1.544 | 4.724 | 1.445 |
| 1st Count | | 36 | | 0 | | 1 | | 17 | | 0 | | 0 | | 0 | | 0 | | 0 | | 0 | | 0 | | 0 | |

## D.2 SHORT-TERM FORECASTING

Our complete results on short-term forecasting are presented in Tab. 12. TIME-LLM consistently outperforms the majority of baseline models in most cases. Notably, we surpass GPT4TS by a large margin (e.g., **8.7%** overall, **13.4%** on M4-Yearly, and an average of **21.5%** on M4-Hourly, M4-Daily, and M4-Weekly), as well as TimesNet (e.g., **10%** overall, **14.1%** on M4-Yearly, and an average of **30.1%** on M4-Hourly, M4-Daily, and M4-Weekly). Compared to the recent state-of-the-art forecasting models, N-HiTS and PatchTST, TIME-LLM exhibits comparable or superior performances without any parameter updates on the backbone LLM.

In addition, we conduct a comparative analysis between TIME-LLM and the top-performing models on the M3-Quarterly dataset, with the findings presented in Tab. 13. We provide additional metrics, namely MRAE and MAPE, alongside the default SMAPE used in the M3 competition. On this dataset, TIME-LLM attains on-par performance compared to TimesNet and PatchTST, outperforming GPT4TS by substantial margins, achieving reductions of over **23%**, **35%**, and **26%** in SMAPE, MRAE, and MAPE, respectively.

## E FEW-SHOT AND ZERO-SHOT FORECASTING

## E.1 FEW-SHOT FORECASTING

Our full results in few-shot forecasting tasks are detailed in Tab. 14 and Tab. 15. Within the scope of 10% few-shot learning, TIME-LLM secures SOTA performance in **32** out of 35 cases, spanning seven different time series benchmarks. Our approach's advantage becomes even more pronounced in the context of 5% few-shot scenarios, achieving SOTA results in **21** out of 32 cases. We attribute this to the successful knowledge activation in our reprogrammed LLM.

Table 11: Additional comparison with other baselines in long-term forecasting tasks. We set the forecasting horizons $H \in \{24, 36, 48, 60\}$ for ILI and $\{96, 192, 336, 720\}$ for the others. A lower value indicates better performance. **Red**: the best, Blue: the second best.

| Methods | | TIME-LLM MSE | TIME-LLM MAE | N-BEATS MSE | N-BEATS MAE | N-HiTS MSE | N-HiTS MAE | AutoARIMA MSE | AutoARIMA MAE | AutoTheta MSE | AutoTheta MAE | AutoETS MSE | AutoETS MAE |
|---|---|---|---|---|---|---|---|---|---|---|---|---|---|
| ETTh1 | 96 | **0.362** | **0.392** | 0.496 | 0.475 | 0.392 | 0.407 | 0.933 | 0.635 | 1.266 | 0.758 | 1.264 | 0.756 |
| | 192 | **0.398** | **0.418** | 0.544 | 0.504 | 0.442 | 0.438 | 0.868 | 0.621 | 1.188 | 0.749 | 1.181 | 0.745 |
| | 336 | **0.430** | **0.427** | 0.592 | 0.533 | 0.497 | 0.471 | 0.964 | 0.663 | 1.310 | 0.799 | 1.292 | 0.792 |
| | 720 | **0.442** | **0.457** | 0.639 | 0.588 | 0.559 | 0.533 | 1.043 | 0.705 | 1.510 | 0.882 | 1.405 | 0.842 |
| | Avg | **0.408** | **0.423** | 0.568 | 0.525 | 0.473 | 0.462 | 0.952 | 0.656 | 1.319 | 0.797 | 1.286 | 0.784 |
| ETTh2 | 96 | **0.268** | **0.328** | 0.384 | 0.431 | 0.321 | 0.368 | 0.390 | 0.417 | 0.461 | 0.430 | 0.444 | 0.403 |
| | 192 | **0.329** | **0.375** | 0.496 | 0.493 | 0.398 | 0.421 | 0.545 | 0.492 | 0.754 | 0.537 | 0.771 | 0.461 |
| | 336 | **0.368** | **0.409** | 0.585 | 0.542 | 0.453 | 0.459 | 0.697 | 0.562 | 1.355 | 0.683 | 1.526 | 0.522 |
| | 720 | **0.372** | **0.420** | 0.792 | 0.651 | 0.775 | 0.609 | 0.907 | 0.658 | 3.971 | 1.061 | 5.183 | 0.633 |
| | Avg | **0.334** | **0.383** | 0.564 | 0.529 | 0.487 | 0.464 | 0.635 | 0.532 | 1.635 | 0.678 | 1.981 | 0.505 |
| ETTm1 | 96 | **0.272** | **0.334** | 0.393 | 0.412 | 0.327 | 0.368 | 1.091 | 0.661 | 1.211 | 0.704 | 1.519 | 0.768 |
| | 192 | **0.310** | **0.358** | 0.425 | 0.427 | 0.376 | 0.400 | 1.119 | 0.682 | 1.237 | 0.724 | 1.535 | 0.784 |
| | 336 | **0.352** | **0.384** | 0.464 | 0.454 | 0.407 | 0.423 | 1.125 | 0.698 | 1.231 | 0.735 | 1.472 | 0.782 |
| | 720 | **0.383** | **0.411** | 0.521 | 0.488 | 0.471 | 0.456 | 1.243 | 0.745 | 1.394 | 0.801 | 1.591 | 0.825 |
| | Avg | **0.329** | **0.372** | 0.451 | 0.445 | 0.395 | 0.412 | 1.145 | 0.697 | 1.268 | 0.741 | 1.529 | 0.790 |
| ETTm2 | 96 | **0.161** | **0.253** | 0.204 | 0.302 | 0.188 | 0.273 | 0.435 | 0.375 | 0.245 | 0.316 | 0.359 | 0.333 |
| | 192 | **0.219** | **0.293** | 0.282 | 0.358 | 0.274 | 0.338 | 0.995 | 0.494 | 0.413 | 0.401 | 0.756 | 0.396 |
| | 336 | **0.271** | **0.329** | 0.384 | 0.425 | 0.378 | 0.406 | 2.324 | 0.648 | 0.790 | 0.528 | 1.747 | 0.467 |
| | 720 | **0.352** | **0.379** | 0.555 | 0.523 | 0.501 | 0.488 | 9.064 | 1.020 | 2.451 | 0.847 | 6.856 | 0.639 |
| | Avg | **0.251** | **0.313** | 0.355 | 0.402 | 0.337 | 0.376 | 3.205 | 0.634 | 0.975 | 0.523 | 2.430 | 0.459 |
| Weather | 96 | **0.147** | **0.201** | 0.185 | 0.244 | 0.160 | 0.222 | 0.255 | 0.273 | 0.279 | 0.266 | 0.331 | 0.277 |
| | 192 | **0.189** | **0.234** | 0.225 | 0.282 | 0.202 | 0.265 | 0.390 | 0.353 | 0.337 | 0.316 | 0.498 | 0.345 |
| | 336 | 0.262 | 0.279 | 0.274 | 0.323 | **0.253** | **0.303** | 0.775 | 0.457 | 0.472 | 0.385 | 0.898 | 0.423 |
| | 720 | **0.304** | **0.316** | 0.340 | 0.373 | 0.323 | 0.354 | 2.898 | 0.707 | 0.818 | 0.526 | 2.820 | 0.580 |
| | Avg | **0.225** | **0.257** | 0.256 | 0.306 | 0.235 | 0.286 | 1.080 | 0.448 | 0.477 | 0.373 | 1.137 | 0.406 |
| Electricity | 96 | **0.131** | **0.224** | 0.233 | 0.327 | 0.184 | 0.275 | 0.520 | 0.466 | 0.653 | 0.532 | 0.650 | 0.526 |
| | 192 | **0.152** | **0.241** | 0.246 | 0.340 | 0.190 | 0.282 | 0.581 | 0.499 | 0.713 | 0.561 | 0.704 | 0.549 |
| | 336 | **0.160** | **0.248** | 0.262 | 0.355 | 0.205 | 0.298 | 0.602 | 0.515 | 0.797 | 0.603 | 0.766 | 0.577 |
| | 720 | **0.192** | **0.298** | 0.296 | 0.383 | 0.239 | 0.330 | 0.685 | 0.558 | 1.023 | 0.688 | 0.901 | 0.628 |
| | Avg | **0.158** | **0.252** | 0.259 | 0.351 | 0.205 | 0.296 | 0.597 | 0.510 | 0.797 | 0.596 | 0.755 | 0.570 |
| Traffic | 96 | **0.362** | **0.248** | 0.608 | 0.447 | 0.410 | 0.329 | 1.068 | 0.694 | 3.207 | 1.219 | 3.254 | 1.221 |
| | 192 | **0.374** | **0.247** | 0.605 | 0.448 | 0.414 | 0.330 | 1.380 | 0.775 | 3.407 | 1.262 | 3.569 | 1.264 |
| | 336 | **0.385** | **0.271** | 0.618 | 0.454 | 0.428 | 0.337 | 1.448 | 0.790 | 3.473 | 1.274 | 3.971 | 1.275 |
| | 720 | **0.430** | **0.288** | 0.650 | 0.467 | 0.456 | 0.354 | 1.481 | 0.799 | 3.952 | 1.382 | 6.784 | 1.379 |
| | Avg | **0.388** | **0.264** | 0.620 | 0.454 | 0.427 | 0.338 | 1.344 | 0.765 | 3.510 | 1.284 | 4.395 | 1.285 |
| ILI | 24 | **1.285** | **0.727** | 6.809 | 1.870 | 2.675 | 1.080 | 4.909 | 1.329 | 5.991 | 1.510 | 4.869 | 1.315 |
| | 36 | **1.404** | **0.814** | 6.850 | 1.890 | 3.081 | 1.194 | 5.079 | 1.440 | 5.922 | 1.539 | 4.917 | 1.422 |
| | 48 | **1.523** | **0.807** | 6.788 | 1.876 | 2.973 | 1.176 | 4.276 | 1.339 | 4.637 | 1.329 | 3.966 | 1.301 |
| | 60 | **1.531** | **0.854** | 6.908 | 1.893 | 3.259 | 1.232 | 3.855 | 1.276 | 4.378 | 1.345 | 3.540 | 1.229 |
| | Avg | **1.435** | **0.801** | 6.839 | 1.882 | 2.997 | 1.171 | 4.530 | 1.346 | 5.232 | 1.431 | 4.323 | 1.317 |
| 1st Count | | **40** | | 0 | | 1 | | 0 | | 0 | | 0 | |

Table 12: Full short-term time series forecasting results. The forecasting horizons are in [6, 48] and the last three rows are weighted averaged from all datasets under different sampling intervals. A lower value indicates better performance. **Red**: the best, Blue: the second best.

| | Methods | TIME-LLM | GPT4TS | TimesNet | PatchTST | N-HiTS | N-BEATS | ETSformer | LightTS | DLinear | FEDformer | Stationary | Autoformer | Informer | Reformer |
|---|---|---|---|---|---|---|---|---|---|---|---|---|---|---|---|
| Yearly | SMAPE | **13.419** | 15.11 | 15.378 | 13.477 | 13.422 | 13.487 | 18.009 | 14.247 | 16.965 | 14.021 | 13.717 | 13.974 | 14.727 | 16.169 |
| | MASE | **3.005** | 3.565 | 3.554 | 3.019 | 3.056 | 3.036 | 4.487 | 3.109 | 4.283 | 3.036 | 3.078 | 3.134 | 3.418 | 3.800 |
| | OWA | **0.789** | 0.911 | 0.918 | 0.792 | 0.795 | 0.795 | 1.115 | 0.827 | 1.058 | 0.811 | 0.807 | 0.822 | 0.881 | 0.973 |
| Quarterly | SMAPE | **10.110** | 10.597 | 10.465 | 10.38 | 10.185 | 10.564 | 13.376 | 11.364 | 12.145 | 11.1 | 10.958 | 11.338 | 11.360 | 13.313 |
| | MASE | **1.178** | 1.253 | 1.227 | 1.233 | 1.18 | 1.252 | 1.906 | 1.328 | 1.520 | 1.35 | 1.325 | 1.365 | 1.401 | 1.775 |
| | OWA | **0.889** | 0.938 | 0.923 | 0.921 | 0.893 | 0.936 | 1.302 | 1.000 | 1.106 | 0.996 | 0.981 | 1.012 | 1.027 | 1.252 |
| Monthly | SMAPE | 12.980 | 13.258 | 13.513 | **12.959** | 13.059 | 13.089 | 14.588 | 14.014 | 13.514 | 14.403 | 13.917 | 13.958 | 14.062 | 20.128 |
| | MASE | **0.963** | 1.003 | 1.039 | 0.97 | 1.013 | 0.996 | 1.368 | 1.053 | 1.037 | 1.147 | 1.097 | 1.103 | 1.141 | 2.614 |
| | OWA | **0.903** | 0.931 | 0.957 | 0.905 | 0.929 | 0.922 | 1.149 | 0.981 | 0.956 | 1.021 | 0.998 | 1.002 | 1.024 | 1.927 |
| Others | SMAPE | 4.795 | 6.124 | 6.913 | 4.952 | **4.711** | 6.599 | 7.267 | 15.880 | 6.709 | 7.148 | 6.302 | 5.485 | 24.460 | 32.491 |
| | MASE | 3.178 | 4.116 | 4.507 | 3.347 | **3.054** | 4.43 | 5.240 | 11.434 | 4.953 | 4.041 | 4.064 | 3.865 | 20.960 | 33.355 |
| | OWA | 1.006 | 1.259 | 1.438 | 1.049 | **0.977** | 1.393 | 1.591 | 3.474 | 1.487 | 1.389 | 1.304 | 1.187 | 5.879 | 8.679 |
| Average | SMAPE | **11.983** | 12.69 | 12.88 | 12.059 | 12.035 | 12.25 | 14.718 | 13.525 | 13.639 | 13.16 | 12.780 | 12.909 | 14.086 | 18.200 |
| | MASE | **1.595** | 1.808 | 1.836 | 1.623 | 1.625 | 1.698 | 2.408 | 2.111 | 2.095 | 1.775 | 1.756 | 1.771 | 2.718 | 4.223 |
| | OWA | **0.859** | 0.94 | 0.955 | 0.869 | 0.869 | 0.896 | 1.172 | 1.051 | 1.051 | 0.949 | 0.930 | 0.939 | 1.230 | 1.775 |

## E.2 ZERO-SHOT FORECASTING

The full results of zero-shot forecasting are summarized in Tab. 16. TIME-LLM remarkably surpasses the six most competitive time series models in zero-shot adaptation. Overall, we observe over **23.5%** and **12.4%** MSE and MAE reductions across all baselines on average. Our improvements are consistently significant on those typical cross-domain scenarios (e.g., ETTh2 → ETTh1 and ETTm2 → ETTm1), over **20.8%** and **11.3%** on average w.r.t. MSE and MAE. Significantly, TIME-LLM exhibits superior performance gains in comparison to LLMTime (Gruver et al., 2023), which employs a similarly sized backbone LLM (7B) and is the latest effort in leveraging LLMs for

Table 13: Additional short-term time series forecasting results on M3 (Quarterly). The forecasting horizon is 8. A lower value indicates better performance. **Red**: the best, Blue: the second best.

| Methods | TIME-LLM | GPT4TS | TimesNet | PatchTST | N-HiTS | N-BEATS | DLinear | FEDformer |
|---|---|---|---|---|---|---|---|---|
| SMAPE | 11.171 | 14.453 | **10.410** | 12.380 | 12.616 | 18.640 | 15.028 | 12.927 |
| MRAE | 3.282 | 5.035 | 3.310 | **2.401** | 4.271 | 4.612 | 2.793 | 3.653 |
| MAPE | 0.151 | 0.203 | **0.140** | 0.154 | 0.168 | 0.247 | 0.196 | 0.174 |

zero-shot time series forecasting. We attribute this success to our reprogramming framework being better at activating the LLM's knowledge transfer and reasoning capabilities in a resource-efficient manner when performing time series tasks.

## F ABLATION STUDY

The full ablation results are in Tab. 17. We additionally compare the model performance under reprogramming and fine-tuning (with QLoRA Dettmers et al. (2023)) protocols. Our results indicate a clear performance gain of our approach compared to the QLoRA variant (**A.5**) by **19%** in average.

## G EFFICIENCY COMPARISON WITH MODEL FINE-TUNING

**Setups.** We compare the efficiency of model fine-tuning (with QLoRA Dettmers et al. (2023)) and our proposed model reprogramming in this section with two different backbones, that is, Llama in 1/4 capacity (first 8 Transformer layers) and full capacity. Here, we adhere to the long-term forecasting protocol on ETTh1 to forecast two different steps (that is, 96 and 336 in this case) ahead. For the evaluation metrics, we report the total number of trainable parameters (in million), GPU memory (in mebibyte), and running time (seconds per iteration).

**Results.** Our results are given in Tab. 18. We see that model reprogramming remarkably results in better efficiency compared to parameter-efficient fine-tuning (PEFT) with QLoRA on long-range

Table 14: Full few-shot learning results on 10% training data. We use the same protocol as in Tab. 1.

| Methods | | TIME-LLM | | GPT4TS | | DLinear | | PatchTST | | TimesNet | | FEDformer | | Autoformer | | Stationary | | ETSformer | | LightTS | | Informer | | Reformer | |
|---|---|---|---|---|---|---|---|---|---|---|---|---|---|---|---|---|---|---|---|---|---|---|---|---|---|
| Metric | | MSE | MAE | MSE | MAE | MSE | MAE | MSE | MAE | MSE | MAE | MSE | MAE | MSE | MAE | MSE | MAE | MSE | MAE | MSE | MAE | MSE | MAE | MSE | MAE |
| ETTh1 | 96 | 0.448 | 0.460 | 0.458 | 0.456 | 0.492 | 0.495 | 0.516 | 0.485 | 0.861 | 0.628 | 0.512 | 0.499 | 0.613 | 0.552 | 0.918 | 0.639 | 1.112 | 0.806 | 1.298 | 0.838 | 1.179 | 0.792 | 1.184 | 0.790 |
| | 192 | 0.484 | 0.483 | 0.570 | 0.516 | 0.565 | 0.538 | 0.598 | 0.524 | 0.797 | 0.593 | 0.624 | 0.555 | 0.722 | 0.598 | 0.915 | 0.629 | 1.155 | 0.823 | 1.322 | 0.854 | 1.199 | 0.806 | 1.295 | 0.850 |
| | 336 | 0.589 | 0.540 | 0.608 | 0.535 | 0.721 | 0.622 | 0.657 | 0.550 | 0.941 | 0.648 | 0.691 | 0.574 | 0.750 | 0.619 | 0.939 | 0.644 | 1.179 | 0.832 | 1.347 | 0.870 | 1.202 | 0.811 | 1.294 | 0.854 |
| | 720 | 0.700 | 0.604 | 0.725 | 0.591 | 0.986 | 0.743 | 0.762 | 0.610 | 0.877 | 0.641 | 0.728 | 0.614 | 0.721 | 0.616 | 0.887 | 0.645 | 1.273 | 0.874 | 1.534 | 0.947 | 1.217 | 0.825 | 1.223 | 0.838 |
| | Avg | 0.556 | 0.522 | 0.590 | 0.525 | 0.691 | 0.600 | 0.633 | 0.542 | 0.869 | 0.628 | 0.639 | 0.561 | 0.702 | 0.596 | 0.915 | 0.639 | 1.180 | 0.834 | 1.375 | 0.877 | 1.199 | 0.809 | 1.249 | 0.833 |
| ETTh2 | 96 | 0.275 | 0.326 | 0.331 | 0.374 | 0.357 | 0.411 | 0.353 | 0.389 | 0.378 | 0.409 | 0.382 | 0.416 | 0.413 | 0.451 | 0.389 | 0.411 | 0.678 | 0.619 | 2.022 | 1.006 | 3.837 | 1.508 | 3.788 | 1.533 |
| | 192 | 0.374 | 0.373 | 0.402 | 0.411 | 0.569 | 0.519 | 0.403 | 0.414 | 0.490 | 0.467 | 0.478 | 0.474 | 0.474 | 0.477 | 0.473 | 0.455 | 0.785 | 0.666 | 2.329 | 1.104 | 3.856 | 1.513 | 3.552 | 1.483 |
| | 336 | 0.406 | 0.429 | 0.406 | 0.433 | 0.671 | 0.572 | 0.426 | 0.441 | 0.537 | 0.494 | 0.504 | 0.501 | 0.547 | 0.543 | 0.507 | 0.480 | 0.839 | 0.694 | 2.453 | 1.122 | 3.952 | 1.526 | 3.395 | 1.526 |
| | 720 | 0.427 | 0.449 | 0.449 | 0.464 | 0.824 | 0.648 | 0.477 | 0.480 | 0.510 | 0.491 | 0.499 | 0.509 | 0.516 | 0.523 | 0.477 | 0.472 | 1.273 | 0.874 | 2.655 | 1.160 | 3.842 | 1.503 | 3.205 | 1.401 |
| | Avg | 0.370 | 0.394 | 0.397 | 0.421 | 0.605 | 0.538 | 0.415 | 0.431 | 0.479 | 0.465 | 0.466 | 0.475 | 0.488 | 0.499 | 0.462 | 0.455 | 0.894 | 0.713 | 2.655 | 1.160 | 3.872 | 1.513 | 3.485 | 1.486 |
| ETTm1 | 96 | 0.346 | 0.388 | 0.390 | 0.404 | 0.352 | 0.392 | 0.410 | 0.419 | 0.583 | 0.501 | 0.578 | 0.518 | 0.774 | 0.614 | 0.761 | 0.568 | 0.911 | 0.688 | 0.921 | 0.682 | 1.162 | 0.785 | 1.442 | 0.847 |
| | 192 | 0.373 | 0.416 | 0.429 | 0.423 | 0.382 | 0.412 | 0.437 | 0.434 | 0.630 | 0.528 | 0.617 | 0.546 | 0.754 | 0.592 | 0.781 | 0.574 | 0.955 | 0.703 | 0.957 | 0.701 | 1.172 | 0.793 | 1.444 | 0.862 |
| | 336 | 0.413 | 0.426 | 0.469 | 0.439 | 0.419 | 0.434 | 0.476 | 0.454 | 0.725 | 0.568 | 0.998 | 0.775 | 0.869 | 0.677 | 0.803 | 0.587 | 0.991 | 0.719 | 0.998 | 0.716 | 1.227 | 0.908 | 1.450 | 0.866 |
| | 720 | 0.485 | 0.476 | 0.569 | 0.498 | 0.490 | 0.477 | 0.681 | 0.556 | 0.769 | 0.549 | 0.693 | 0.579 | 0.810 | 0.630 | 0.844 | 0.581 | 1.062 | 0.747 | 1.007 | 0.719 | 1.207 | 0.797 | 1.366 | 0.850 |
| | Avg | 0.404 | 0.427 | 0.464 | 0.441 | 0.411 | 0.429 | 0.501 | 0.466 | 0.677 | 0.537 | 0.722 | 0.605 | 0.802 | 0.628 | 0.797 | 0.578 | 0.980 | 0.714 | 0.971 | 0.705 | 1.192 | 0.821 | 1.426 | 0.856 |
| ETTm2 | 96 | 0.177 | 0.261 | 0.188 | 0.269 | 0.213 | 0.303 | 0.191 | 0.274 | 0.212 | 0.285 | 0.291 | 0.399 | 0.352 | 0.454 | 0.229 | 0.308 | 0.331 | 0.430 | 0.813 | 0.688 | 3.203 | 1.407 | 4.195 | 1.628 |
| | 192 | 0.241 | 0.314 | 0.251 | 0.309 | 0.278 | 0.345 | 0.252 | 0.317 | 0.270 | 0.323 | 0.307 | 0.379 | 0.694 | 0.691 | 0.291 | 0.343 | 0.400 | 0.464 | 1.008 | 0.768 | 3.112 | 1.387 | 4.042 | 1.601 |
| | 336 | 0.274 | 0.327 | 0.307 | 0.346 | 0.338 | 0.385 | 0.306 | 0.353 | 0.323 | 0.353 | 0.543 | 0.559 | 2.408 | 1.407 | 0.348 | 0.376 | 0.469 | 0.498 | 1.031 | 0.775 | 3.255 | 1.421 | 3.963 | 1.585 |
| | 720 | 0.417 | 0.390 | 0.426 | 0.417 | 0.436 | 0.440 | 0.433 | 0.427 | 0.474 | 0.449 | 0.712 | 0.614 | 1.913 | 1.166 | 0.461 | 0.438 | 0.589 | 0.557 | 1.096 | 0.791 | 3.909 | 1.543 | 3.711 | 1.532 |
| | Avg | 0.277 | 0.323 | 0.293 | 0.335 | 0.316 | 0.368 | 0.296 | 0.343 | 0.320 | 0.353 | 0.463 | 0.488 | 1.342 | 0.930 | 0.332 | 0.366 | 0.447 | 0.487 | 0.987 | 0.756 | 3.370 | 1.440 | 3.978 | 1.587 |
| Weather | 96 | 0.161 | 0.210 | 0.163 | 0.215 | 0.171 | 0.224 | 0.165 | 0.215 | 0.184 | 0.230 | 0.188 | 0.253 | 0.221 | 0.297 | 0.192 | 0.234 | 0.199 | 0.272 | 0.217 | 0.269 | 0.374 | 0.401 | 0.335 | 0.380 |
| | 192 | 0.204 | 0.248 | 0.210 | 0.254 | 0.215 | 0.263 | 0.210 | 0.257 | 0.245 | 0.283 | 0.250 | 0.304 | 0.270 | 0.322 | 0.269 | 0.295 | 0.279 | 0.332 | 0.259 | 0.304 | 0.552 | 0.478 | 0.522 | 0.462 |
| | 336 | 0.261 | 0.302 | 0.256 | 0.292 | 0.258 | 0.299 | 0.259 | 0.297 | 0.305 | 0.321 | 0.312 | 0.346 | 0.320 | 0.351 | 0.370 | 0.357 | 0.356 | 0.386 | 0.303 | 0.334 | 724 | 0.541 | 0.715 | 0.535 |
| | 720 | 0.309 | 0.332 | 0.321 | 0.339 | 0.320 | 0.346 | 0.332 | 0.346 | 0.381 | 0.371 | 0.387 | 0.393 | 0.390 | 0.396 | 0.441 | 0.405 | 0.437 | 0.448 | 0.377 | 0.382 | 0.739 | 0.558 | 0.611 | 0.500 |
| | Avg | 0.234 | 0.273 | 0.238 | 0.275 | 0.241 | 0.283 | 0.242 | 0.279 | 0.279 | 0.301 | 0.284 | 0.324 | 0.300 | 0.342 | 0.318 | 0.323 | 0.318 | 0.360 | 0.289 | 0.322 | 0.597 | 0.495 | 0.546 | 0.469 |
| Electricity | 96 | 0.139 | 0.241 | 0.139 | 0.237 | 0.150 | 0.253 | 0.140 | 0.238 | 0.299 | 0.373 | 0.231 | 0.323 | 0.261 | 0.348 | 0.420 | 0.466 | 0.599 | 0.587 | 0.350 | 0.425 | 1.259 | 0.919 | 0.993 | 0.784 |
| | 192 | 0.151 | 0.248 | 0.156 | 0.252 | 0.164 | 0.264 | 0.160 | 0.255 | 0.305 | 0.379 | 0.261 | 0.356 | 0.338 | 0.406 | 0.411 | 0.459 | 0.620 | 0.598 | 0.376 | 0.448 | 1.160 | 0.873 | 0.938 | 0.753 |
| | 336 | 0.169 | 0.270 | 0.175 | 0.270 | 0.181 | 0.282 | 0.180 | 0.276 | 0.319 | 0.391 | 0.360 | 0.445 | 0.410 | 0.474 | 0.434 | 0.473 | 0.662 | 0.619 | 0.428 | 0.485 | 1.157 | 0.872 | 0.925 | 0.745 |
| | 720 | 0.240 | 0.322 | 0.233 | 0.317 | 0.223 | 0.321 | 0.241 | 0.323 | 0.369 | 0.426 | 0.530 | 0.585 | 0.715 | 0.685 | 0.510 | 0.521 | 0.757 | 0.664 | 0.611 | 0.597 | 1.203 | 0.898 | 1.004 | 0.790 |
| | Avg | 0.175 | 0.270 | 0.176 | 0.269 | 0.180 | 0.280 | 0.180 | 0.273 | 0.323 | 0.392 | 0.346 | 0.427 | 0.431 | 0.478 | 0.444 | 0.480 | 0.660 | 0.617 | 0.441 | 0.489 | 1.195 | 0.891 | 0.965 | 0.768 |
| Traffic | 96 | 0.418 | 0.291 | 0.414 | 0.297 | 0.419 | 0.298 | 0.403 | 0.289 | 0.719 | 0.416 | 0.639 | 0.400 | 0.672 | 0.405 | 1.412 | 0.802 | 1.643 | 0.855 | 1.157 | 0.636 | 1.557 | 0.821 | 1.527 | 0.815 |
| | 192 | 0.414 | 0.296 | 0.426 | 0.301 | 0.434 | 0.305 | 0.415 | 0.296 | 0.748 | 0.428 | 0.637 | 0.416 | 0.727 | 0.424 | 1.419 | 0.806 | 1.641 | 0.854 | 1.207 | 0.661 | 1.454 | 0.765 | 1.538 | 0.817 |
| | 336 | 0.421 | 0.311 | 0.434 | 0.303 | 0.449 | 0.313 | 0.426 | 0.304 | 0.853 | 0.471 | 0.655 | 0.427 | 0.749 | 0.454 | 1.443 | 0.815 | 1.711 | 0.878 | 1.334 | 0.713 | 1.521 | 0.812 | 1.550 | 0.819 |
| | 720 | 0.462 | 0.327 | 0.487 | 0.337 | 0.484 | 0.336 | 0.474 | 0.331 | 1.485 | 0.825 | 0.722 | 0.456 | 0.847 | 0.499 | 1.539 | 0.837 | 2.660 | 1.157 | 1.292 | 0.726 | 1.605 | 0.846 | 1.588 | 0.833 |
| | Avg | 0.429 | 0.306 | 0.440 | 0.310 | 0.447 | 0.313 | 0.430 | 0.305 | 0.951 | 0.535 | 0.663 | 0.425 | 0.749 | 0.446 | 1.453 | 0.815 | 1.914 | 0.936 | 1.248 | 0.684 | 1.534 | 0.811 | 1.551 | 0.821 |
| 1st Count | | 32 | | 9 | | 3 | | 3 | | 0 | | 0 | | 0 | | 0 | | 0 | | 0 | | 0 | | 0 | | 0 | |

Table 15: Full few-shot learning results on 5% training data. We use the same protocol as in Tab. 1. '-' means that 5% time series is not sufficient to constitute a training set.

| | Methods | TIME-LLM | | GPT4TS | | DLinear | | PatchTST | | TimesNet | | FEDformer | | Autoformer | | Stationary | | ETSformer | | LightTS | | Informer | | Reformer | |
|---|---|---|---|---|---|---|---|---|---|---|---|---|---|---|---|---|---|---|---|---|---|---|---|---|---|
| | Metric | MSE | MAE | MSE | MAE | MSE | MAE | MSE | MAE | MSE | MAE | MSE | MAE | MSE | MAE | MSE | MAE | MSE | MAE | MSE | MAE | MSE | MAE | MSE | MAE |
| ETTh1 | 96 | **0.483** | **0.464** | 0.543 | 0.506 | 0.547 | 0.503 | 0.557 | 0.519 | 0.892 | 0.625 | 0.593 | 0.529 | 0.681 | 0.570 | 0.952 | 0.650 | 1.169 | 0.832 | 1.483 | 0.91 | 1.225 | 0.812 | 1.198 | 0.795 |
| | 192 | **0.629** | **0.540** | 0.748 | 0.580 | 0.720 | 0.604 | 0.711 | 0.570 | 0.940 | 0.665 | 0.652 | 0.563 | 0.725 | 0.602 | 0.943 | 0.645 | 1.221 | 0.853 | 1.525 | 0.93 | 1.249 | 0.828 | 1.273 | 0.853 |
| | 336 | 0.768 | 0.626 | 0.754 | 0.595 | 0.984 | 0.727 | 0.816 | 0.619 | 0.945 | 0.653 | **0.731** | **0.594** | 0.761 | 0.624 | 0.935 | 0.644 | 1.179 | 0.832 | 1.347 | 0.87 | 1.202 | 0.811 | 1.254 | 0.857 |
| | 720 | - | - | - | - | - | - | - | - | - | - | - | - | - | - | - | - | - | - | - | - | - | - | - | - |
| | Avg | **0.627** | **0.543** | 0.681 | 0.560 | 0.750 | 0.611 | 0.694 | 0.569 | 0.925 | 0.647 | 0.658 | 0.562 | 0.722 | 0.598 | 0.943 | 0.646 | 1.189 | 0.839 | 1.451 | 0.903 | 1.225 | 0.817 | 1.241 | 0.835 |
| ETTh2 | 96 | **0.336** | **0.397** | 0.376 | 0.421 | 0.442 | 0.456 | 0.401 | 0.421 | 0.409 | 0.420 | 0.390 | 0.424 | 0.428 | 0.468 | 0.408 | 0.423 | 0.678 | 0.619 | 2.022 | 1.006 | 3.837 | 1.508 | 3.753 | 1.518 |
| | 192 | **0.406** | **0.425** | 0.418 | 0.441 | 0.617 | 0.542 | 0.452 | 0.455 | 0.483 | 0.464 | 0.457 | 0.465 | 0.496 | 0.504 | 0.497 | 0.468 | 0.845 | 0.697 | 3.534 | 1.348 | 3.975 | 1.933 | 3.516 | 1.473 |
| | 336 | **0.405** | **0.432** | 0.408 | 0.439 | 1.424 | 0.849 | 0.464 | 0.469 | 0.499 | 0.479 | 0.477 | 0.483 | 0.486 | 0.496 | 0.507 | 0.481 | 0.905 | 0.727 | 4.063 | 1.451 | 3.956 | 1.520 | 3.312 | 1.427 |
| | 720 | - | - | - | - | - | - | - | - | - | - | - | - | - | - | - | - | - | - | - | - | - | - | - | - |
| | Avg | **0.382** | **0.418** | 0.400 | 0.433 | 0.694 | 0.577 | 0.827 | 0.615 | 0.439 | 0.448 | 0.463 | 0.454 | 0.441 | 0.457 | 0.470 | 0.489 | 0.809 | 0.681 | 3.206 | 1.268 | 3.922 | 1.653 | 3.527 | 1.472 |
| ETTm1 | 96 | **0.316** | 0.377 | 0.386 | 0.405 | 0.332 | **0.374** | 0.399 | 0.414 | 0.606 | 0.518 | 0.628 | 0.544 | 0.726 | 0.578 | 0.823 | 0.587 | 1.031 | 0.747 | 1.048 | 0.733 | 1.130 | 0.775 | 1.234 | 0.798 |
| | 192 | 0.450 | 0.464 | **0.440** | 0.438 | 0.358 | 0.390 | 0.441 | 0.436 | 0.681 | 0.539 | 0.666 | 0.566 | 0.750 | 0.591 | 0.844 | 0.591 | 1.087 | 0.766 | 1.097 | 0.756 | 1.150 | 0.788 | 1.287 | 0.839 |
| | 336 | 0.450 | 0.424 | 0.485 | 0.459 | **0.402** | **0.416** | 0.499 | 0.467 | 0.786 | 0.597 | 0.807 | 0.628 | 0.851 | 0.659 | 0.870 | 0.603 | 1.138 | 0.787 | 1.147 | 0.775 | 1.198 | 0.809 | 1.288 | 0.842 |
| | 720 | **0.483** | **0.471** | 0.577 | 0.499 | 0.511 | 0.489 | 0.767 | 0.587 | 0.796 | 0.593 | 0.822 | 0.633 | 0.857 | 0.655 | 0.893 | 0.611 | 1.245 | 0.831 | 1.200 | 0.799 | 1.175 | 0.794 | 1.247 | 0.828 |
| | Avg | 0.425 | 0.434 | 0.472 | 0.450 | **0.400** | **0.417** | 0.526 | 0.476 | 0.717 | 0.561 | 0.730 | 0.592 | 0.796 | 0.620 | 0.857 | 0.598 | 1.125 | 0.782 | 1.123 | 0.765 | 1.163 | 0.791 | 1.264 | 0.826 |
| ETTm2 | 96 | **0.174** | **0.261** | 0.199 | 0.280 | 0.236 | 0.326 | 0.206 | 0.288 | 0.220 | 0.299 | 0.229 | 0.320 | 0.232 | 0.322 | 0.238 | 0.316 | 0.404 | 0.485 | 1.108 | 0.772 | 3.599 | 1.478 | 3.883 | 1.545 |
| | 192 | **0.215** | **0.287** | 0.256 | 0.316 | 0.306 | 0.373 | 0.264 | 0.324 | 0.311 | 0.361 | 0.394 | 0.361 | 0.291 | 0.357 | 0.298 | 0.349 | 0.479 | 0.521 | 1.317 | 0.850 | 3.578 | 1.475 | 3.553 | 1.484 |
| | 336 | **0.273** | **0.330** | 0.318 | 0.353 | 0.380 | 0.423 | 0.334 | 0.367 | 0.338 | 0.366 | 0.378 | 0.427 | 0.478 | 0.517 | 0.353 | 0.380 | 0.552 | 0.555 | 1.415 | 0.879 | 3.561 | 1.473 | 3.446 | 1.460 |
| | 720 | **0.433** | **0.412** | 0.460 | 0.436 | 0.674 | 0.583 | 0.454 | 0.432 | 0.509 | 0.465 | 0.523 | 0.510 | 0.553 | 0.538 | 0.475 | 0.445 | 0.701 | 0.627 | 1.822 | 0.984 | 3.896 | 1.533 | 3.445 | 1.460 |
| | Avg | **0.274** | **0.323** | 0.308 | 0.346 | 0.399 | 0.426 | 0.314 | 0.352 | 0.344 | 0.372 | 0.381 | 0.404 | 0.388 | 0.433 | 0.341 | 0.372 | 0.534 | 0.547 | 1.415 | 0.871 | 3.658 | 1.489 | 3.581 | 1.487 |
| Weather | 96 | 0.172 | 0.263 | 0.175 | 0.230 | 0.184 | 0.242 | **0.171** | **0.224** | 0.207 | 0.253 | 0.229 | 0.309 | 0.227 | 0.299 | 0.215 | 0.252 | 0.218 | 0.295 | 0.230 | 0.285 | 0.497 | 0.497 | 0.406 | 0.435 |
| | 192 | **0.224** | **0.271** | 0.227 | 0.276 | 0.228 | 0.283 | 0.230 | 0.277 | 0.272 | 0.307 | 0.265 | 0.317 | 0.278 | 0.333 | 0.290 | 0.307 | 0.294 | 0.331 | 0.274 | 0.323 | 0.620 | 0.545 | 0.446 | 0.450 |
| | 336 | **0.282** | **0.321** | 0.286 | 0.322 | 0.279 | 0.322 | 0.294 | 0.326 | 0.313 | 0.328 | 0.353 | 0.392 | 0.351 | 0.393 | 0.353 | 0.348 | 0.359 | 0.398 | 0.318 | 0.355 | 0.649 | 0.547 | 0.465 | 0.459 |
| | 720 | **0.366** | 0.381 | 0.366 | **0.379** | 0.364 | 0.388 | 0.384 | 0.387 | 0.400 | 0.385 | 0.391 | 0.394 | 0.387 | 0.389 | 0.452 | 0.407 | 0.461 | 0.461 | 0.401 | 0.418 | 0.570 | 0.522 | 0.471 | 0.468 |
| | Avg | **0.260** | 0.309 | 0.263 | **0.301** | 0.263 | 0.308 | 0.269 | 0.303 | 0.298 | 0.318 | 0.309 | 0.353 | 0.310 | 0.353 | 0.327 | 0.328 | 0.333 | 0.371 | 0.305 | 0.345 | 0.584 | 0.527 | 0.447 | 0.453 |
| Electricity | 96 | 0.147 | **0.242** | **0.143** | **0.241** | 0.150 | 0.251 | 0.145 | 0.244 | 0.315 | 0.389 | 0.235 | 0.322 | 0.297 | 0.367 | 0.484 | 0.518 | 0.697 | 0.638 | 0.639 | 0.609 | 1.265 | 0.919 | 1.414 | 0.855 |
| | 192 | **0.158** | **0.241** | 0.159 | 0.255 | 0.163 | 0.263 | 0.163 | 0.260 | 0.318 | 0.396 | 0.247 | 0.341 | 0.308 | 0.375 | 0.501 | 0.531 | 0.718 | 0.648 | 0.772 | 0.678 | 1.298 | 0.939 | 1.240 | 0.919 |
| | 336 | **0.178** | 0.277 | 0.179 | **0.274** | 0.175 | 0.278 | 0.183 | 0.281 | 0.340 | 0.415 | 0.267 | 0.356 | 0.354 | 0.411 | 0.574 | 0.578 | 0.758 | 0.667 | 0.901 | 0.745 | 1.302 | 0.942 | 1.253 | 0.921 |
| | 720 | 0.224 | 0.312 | 0.233 | 0.323 | 0.219 | 0.311 | 0.233 | 0.323 | 0.635 | 0.613 | 0.318 | 0.394 | 0.426 | 0.466 | 0.952 | 0.786 | 1.028 | 0.788 | 1.200 | 0.871 | 1.259 | 0.919 | 1.249 | 0.921 |
| | Avg | 0.179 | 0.268 | 0.178 | 0.273 | 0.176 | 0.275 | 0.181 | 0.277 | 0.402 | 0.453 | 0.266 | 0.353 | 0.346 | 0.404 | 0.627 | 0.603 | 0.800 | 0.685 | 0.878 | 0.725 | 1.281 | 0.929 | 1.289 | 0.904 |
| Traffic | 96 | 0.414 | 0.291 | 0.419 | 0.298 | 0.427 | 0.304 | **0.404** | **0.286** | 0.854 | 0.492 | 0.670 | 0.421 | 0.795 | 0.481 | 1.468 | 0.821 | 1.643 | 0.855 | 1.157 | 0.636 | 1.557 | 0.821 | 1.586 | 0.841 |
| | 192 | 0.419 | **0.291** | 0.434 | 0.305 | 0.447 | 0.315 | **0.412** | 0.294 | 0.894 | 0.517 | 0.653 | 0.405 | 0.837 | 0.503 | 1.509 | 0.838 | 1.856 | 0.928 | 1.688 | 0.848 | 1.596 | 0.834 | 1.602 | 0.844 |
| | 336 | **0.437** | 0.314 | 0.449 | 0.313 | 0.478 | 0.333 | 0.439 | **0.310** | 0.853 | 0.471 | 0.707 | 0.445 | 0.867 | 0.523 | 1.602 | 0.860 | 2.080 | 0.999 | 1.826 | 0.903 | 1.621 | 0.841 | 1.668 | 0.868 |
| | 720 | - | - | - | - | - | - | - | - | - | - | - | - | - | - | - | - | - | - | - | - | - | - | - | - |
| | Avg | 0.423 | 0.298 | 0.434 | 0.305 | 0.450 | 0.317 | **0.418** | **0.296** | 0.867 | 0.493 | 0.676 | 0.423 | 0.833 | 0.502 | 1.526 | 0.839 | 1.859 | 0.927 | 1.557 | 0.795 | 1.591 | 0.832 | 1.618 | 0.851 |
| 1st Count | | **21** | | 6 | | 7 | | 6 | | 0 | | 1 | | 0 | | 0 | | 0 | | 0 | | 0 | | 0 | |

forecasting tasks in terms of the total number of trainable parameters, GPU memory overhead, and training speed. Quantitatively, there is an **71.2%** trainable parameter reduction on average over four scenarios, leading to **23.1%** smaller memory consumption and **25.3%** faster training speed.

## H  ERROR BARS

All experiments have been conducted three times, and we present the standard deviations of our model and the runner-up model here. The comparisons between our method and the second-best method, PatchTST (Nie et al., 2023), on long-term forecasting tasks, are delineated in Tab. 19. In this table, the average MSE and MAE have been reported across four ETT datasets, complete with standard deviations. Furthermore, Tab. 20 contrasts the effectiveness of our method with that of the second-best method, N-HiTS (Challu et al., 2023a), employing varying M4 datasets for the comparison.

## I  VISUALIZATION

In this part, we visualize the forecasting results of TIME-LLM compared with the state-of-the-art and representative methods (e.g., GPT4TS (Zhou et al., 2023a), PatchTST (Nie et al., 2023), and Autoformer (Wu et al., 2021)) in various scenarios to demonstrate the superior performance of TIME-LLM.

In Fig. 7 and Fig. 8, the long-term (input-96-predict-96) and short-term (input-36-predict-36) forecasts of various approaches are compared with the ground truth. Here, TIME-LLM showcases forecasting accuracy that is notably superior compared to GPT4TS, PatchTST, and a classical Transformer-based method, Autoformer.

We also offer visual comparisons of the forecasting results in both few-shot and zero-shot scenarios, as depicted in Fig. 9 and Fig. 10. We adhere to the long-term (input-96-predict-96) forecasting setup

Table 16: Full zero-shot learning results on ETT datasets. A lower value indicates better performance. **Red**: the best, Blue: the second best.

| Methods | | TIME-LLM | | LLMTime | | GPT4TS | | DLinear | | PatchTST | | TimesNet | | Autoformer | |
|---|---|---|---|---|---|---|---|---|---|---|---|---|---|---|---|
| Metric | | MSE | MAE | MSE | MAE | MSE | MAE | MSE | MAE | MSE | MAE | MSE | MAE | MSE | MAE |
| $ETTh1 \rightarrow ETTh2$ | 96 | **0.279** | **0.337** | 0.510 | 0.576 | 0.335 | 0.374 | 0.347 | 0.400 | 0.304 | 0.350 | 0.358 | 0.387 | 0.469 | 0.486 |
| | 192 | **0.351** | **0.374** | 0.523 | 0.586 | 0.412 | 0.417 | 0.447 | 0.460 | 0.386 | 0.400 | 0.427 | 0.429 | 0.634 | 0.567 |
| | 336 | **0.388** | **0.415** | 0.640 | 0.637 | 0.441 | 0.444 | 0.515 | 0.505 | 0.414 | 0.428 | 0.449 | 0.451 | 0.655 | 0.588 |
| | 720 | **0.391** | **0.420** | 2.296 | 1.034 | 0.438 | 0.452 | 0.665 | 0.589 | 0.419 | 0.443 | 0.448 | 0.458 | 0.570 | 0.549 |
| | Avg | **0.353** | **0.387** | 0.992 | 0.708 | 0.406 | 0.422 | 0.493 | 0.488 | 0.380 | 0.405 | 0.421 | 0.431 | 0.582 | 0.548 |
| $ETTh1 \rightarrow ETTm2$ | 96 | **0.189** | **0.293** | 0.646 | 0.563 | 0.236 | 0.315 | 0.255 | 0.357 | 0.215 | 0.304 | 0.239 | 0.313 | 0.352 | 0.432 |
| | 192 | **0.237** | **0.312** | 0.934 | 0.654 | 0.287 | 0.342 | 0.338 | 0.413 | 0.275 | 0.339 | 0.291 | 0.342 | 0.413 | 0.460 |
| | 336 | **0.291** | **0.365** | 1.157 | 0.728 | 0.341 | 0.374 | 0.425 | 0.465 | 0.334 | 0.373 | 0.342 | 0.371 | 0.465 | 0.489 |
| | 720 | **0.372** | **0.390** | 4.730 | 1.531 | 0.435 | 0.422 | 0.640 | 0.573 | 0.431 | 0.424 | 0.434 | 0.419 | 0.599 | 0.551 |
| | Avg | **0.273** | **0.340** | 1.867 | 0.869 | 0.325 | 0.363 | 0.415 | 0.452 | 0.314 | 0.360 | 0.327 | 0.361 | 0.457 | 0.483 |
| $ETTh2 \rightarrow ETTh1$ | 96 | **0.450** | **0.452** | 1.130 | 0.777 | 0.732 | 0.577 | 0.689 | 0.555 | 0.485 | 0.465 | 0.848 | 0.601 | 0.693 | 0.569 |
| | 192 | **0.465** | **0.461** | 1.242 | 0.820 | 0.758 | 0.559 | 0.707 | 0.568 | 0.565 | 0.509 | 0.860 | 0.610 | 0.760 | 0.601 |
| | 336 | **0.501** | **0.482** | 1.328 | 0.864 | 0.759 | 0.578 | 0.710 | 0.577 | 0.581 | 0.515 | 0.867 | 0.626 | 0.781 | 0.619 |
| | 720 | **0.501** | **0.502** | 4.145 | 1.461 | 0.781 | 0.597 | 0.704 | 0.596 | 0.628 | 0.561 | 0.887 | 0.648 | 0.796 | 0.644 |
| | Avg | **0.479** | **0.474** | 1.961 | 0.981 | 0.757 | 0.578 | 0.703 | 0.574 | 0.565 | 0.513 | 0.865 | 0.621 | 0.757 | 0.608 |
| $ETTh2 \rightarrow ETTm2$ | 96 | **0.174** | **0.276** | 0.646 | 0.563 | 0.253 | 0.329 | 0.240 | 0.336 | 0.226 | 0.309 | 0.248 | 0.324 | 0.263 | 0.352 |
| | 192 | **0.233** | **0.315** | 0.934 | 0.654 | 0.293 | 0.346 | 0.295 | 0.369 | 0.289 | 0.345 | 0.296 | 0.352 | 0.326 | 0.389 |
| | 336 | **0.291** | **0.337** | 1.157 | 0.728 | 0.347 | 0.376 | 0.345 | 0.397 | 0.348 | 0.379 | 0.353 | 0.383 | 0.387 | 0.426 |
| | 720 | **0.392** | **0.417** | 4.730 | 1.531 | 0.446 | 0.429 | 0.432 | 0.442 | 0.439 | 0.427 | 0.471 | 0.446 | 0.487 | 0.478 |
| | Avg | **0.272** | **0.341** | 1.867 | 0.869 | 0.335 | 0.370 | 0.328 | 0.386 | 0.325 | 0.365 | 0.342 | 0.376 | 0.366 | 0.411 |
| $ETTm1 \rightarrow ETTh2$ | 96 | **0.321** | **0.369** | 0.510 | 0.576 | 0.353 | 0.392 | 0.365 | 0.415 | 0.354 | 0.385 | 0.377 | 0.407 | 0.435 | 0.470 |
| | 192 | **0.389** | **0.410** | 0.523 | 0.586 | 0.443 | 0.437 | 0.454 | 0.462 | 0.447 | 0.434 | 0.471 | 0.453 | 0.495 | 0.489 |
| | 336 | **0.408** | **0.433** | 0.640 | 0.637 | 0.469 | 0.461 | 0.496 | 0.494 | 0.481 | 0.463 | 0.472 | 0.484 | 0.470 | 0.472 |
| | 720 | **0.406** | **0.436** | 2.296 | 1.034 | 0.466 | 0.468 | 0.541 | 0.529 | 0.474 | 0.471 | 0.495 | 0.482 | 0.480 | 0.485 |
| | Avg | **0.381** | **0.412** | 0.992 | 0.708 | 0.433 | 0.439 | 0.464 | 0.475 | 0.439 | 0.438 | 0.457 | 0.454 | 0.470 | 0.479 |
| $ETTm1 \rightarrow ETTm2$ | 96 | **0.169** | **0.257** | 0.646 | 0.563 | 0.217 | 0.294 | 0.221 | 0.314 | 0.195 | 0.271 | 0.222 | 0.295 | 0.385 | 0.457 |
| | 192 | **0.227** | **0.318** | 0.934 | 0.654 | 0.277 | 0.327 | 0.286 | 0.359 | 0.258 | 0.311 | 0.288 | 0.337 | 0.433 | 0.469 |
| | 336 | **0.290** | **0.338** | 1.157 | 0.728 | 0.331 | 0.360 | 0.357 | 0.406 | 0.317 | 0.348 | 0.341 | 0.367 | 0.476 | 0.477 |
| | 720 | **0.375** | **0.367** | 4.730 | 1.531 | 0.429 | 0.413 | 0.476 | 0.476 | 0.416 | 0.404 | 0.436 | 0.418 | 0.582 | 0.535 |
| | Avg | **0.268** | **0.320** | 1.867 | 0.869 | 0.313 | 0.348 | 0.335 | 0.389 | 0.296 | 0.334 | 0.322 | 0.354 | 0.469 | 0.484 |
| $ETTm2 \rightarrow ETTh2$ | 96 | **0.298** | **0.356** | 0.510 | 0.576 | 0.360 | 0.401 | 0.333 | 0.391 | 0.327 | 0.367 | 0.360 | 0.401 | 0.353 | 0.393 |
| | 192 | **0.359** | **0.397** | 0.523 | 0.586 | 0.434 | 0.437 | 0.441 | 0.456 | 0.411 | 0.418 | 0.434 | 0.437 | 0.432 | 0.437 |
| | 336 | **0.367** | **0.412** | 0.640 | 0.637 | 0.460 | 0.459 | 0.505 | 0.503 | 0.439 | 0.447 | 0.460 | 0.459 | 0.452 | 0.459 |
| | 720 | **0.393** | **0.434** | 2.296 | 1.034 | 0.485 | 0.477 | 0.543 | 0.534 | 0.459 | 0.470 | 0.485 | 0.477 | 0.453 | 0.467 |
| | Avg | **0.354** | **0.400** | 0.992 | 0.708 | 0.435 | 0.443 | 0.455 | 0.471 | 0.409 | 0.425 | 0.435 | 0.443 | 0.423 | 0.439 |
| $ETTm2 \rightarrow ETTm1$ | 96 | **0.359** | **0.397** | 1.179 | 0.781 | 0.747 | 0.558 | 0.570 | 0.490 | 0.491 | 0.437 | 0.747 | 0.558 | 0.735 | 0.576 |
| | 192 | **0.390** | **0.420** | 1.327 | 0.846 | 0.781 | 0.560 | 0.590 | 0.506 | 0.530 | 0.470 | 0.781 | 0.560 | 0.753 | 0.586 |
| | 336 | **0.421** | **0.445** | 1.478 | 0.902 | 0.778 | 0.578 | 0.706 | 0.567 | 0.565 | 0.497 | 0.778 | 0.578 | 0.750 | 0.593 |
| | 720 | **0.487** | **0.488** | 3.749 | 1.408 | 0.769 | 0.573 | 0.731 | 0.584 | 0.686 | 0.565 | 0.769 | 0.573 | 0.782 | 0.609 |
| | Avg | **0.414** | **0.438** | 1.933 | 0.984 | 0.769 | 0.567 | 0.649 | 0.537 | 0.568 | 0.492 | 0.769 | 0.567 | 0.755 | 0.591 |

Table 17: Full ablations on ETTh1 and ETTm1 in predicting 96 and 192 steps ahead (MSE reported).

| Variant | Long-term Forecasting | | | | Few-shot Forecasting | | | |
|---|---|---|---|---|---|---|---|---|
| | ETTh1-96 | ETTh1-192 | ETTm1-96 | ETThm1-192 | ETTh1-96 | ETTh1-192 | ETTm1-96 | ETThm1-192 |
| **A.1** Llama (**Default**; 32) | 0.362 | 0.398 | 0.272 | 0.310 | 0.448 | 0.484 | 0.346 | 0.373 |
| **A.2** Llama (8) | 0.389 | 0.412 | 0.297 | 0.329 | 0.567 | 0.632 | 0.451 | 0.490 |
| **A.3** GPT-2 (12) | 0.385 | 0.419 | 0.306 | 0.332 | 0.548 | 0.617 | 0.447 | 0.509 |
| **A.4** GPT-2 (6) | 0.394 | 0.427 | 0.311 | 0.342 | 0.571 | 0.640 | 0.468 | 0.512 |
| **A.5** Llama (QLoRA; 32) | 0.391 | 0.420 | 0.310 | 0.338 | 0.543 | 0.611 | 0.578 | 0.618 |
| **B.1** w/o Patch Reprogramming | 0.410 | 0.412 | 0.310 | 0.342 | 0.498 | 0.570 | 0.445 | 0.487 |
| **B.2** w/o Prompt-as-Prefix | 0.398 | 0.423 | 0.298 | 0.339 | 0.521 | 0.617 | 0.432 | 0.481 |
| **C.1** w/o Dataset Context | 0.402 | 0.417 | 0.298 | 0.331 | 0.491 | 0.538 | 0.392 | 0.447 |
| **C.2** w/o Task Instruction | 0.388 | 0.420 | 0.285 | 0.327 | 0.476 | 0.529 | 0.387 | 0.439 |
| **C.3** w/o Statistical Context | 0.391 | 0.419 | 0.279 | 0.347 | 0.483 | 0.547 | 0.421 | 0.461 |

in both cases. TIME-LLM exhibits remarkable superiority in forecasting with limited data—a fact that becomes particularly salient when compared to GPT4TS.

Table 18: Efficiency comparison between model reprogramming and parameter-efficient fine-tuning (PEFT) with QLoRA (Dettmers et al., 2023) on ETTh1 dataset in forecasting two different steps ahead.

| Length | | ETTh1-96 | | | ETTh1-336 | | |
|---|---|---|---|---|---|---|---|
| Metric | | Trainable Param. (M) | Mem. (MiB) | Speed(s/iter) | Trainable Param. (M) | Mem. (MiB) | Speed(s/iter) |
| Llama (8) | QLoRA | 12.60 | 14767 | 0.237 | 12.69 | 15982 | 0.335 |
| | Reprogram | 5.62 | 11370 | 0.184 | 5.71 | 13188 | 0.203 |
| Llama (32) | QLoRA | 50.29 | 45226 | 0.697 | 50.37 | 49374 | 0.732 |
| | Reprogram | 6.39 | 32136 | 0.517 | 6.48 | 37988 | 0.632 |

Table 19: Standard deviations of our approach and the second-best method (PatchTST) on all time series datasets for long-term forecasting.

| Model | TIME-LLM | | PatchTST (2023) | |
|---|---|---|---|---|
| Dataset | MSE | MAE | MSE | MAE |
| ETTh1 | $0.408 \pm 0.011$ | $0.423 \pm 0.012$ | $0.413 \pm 0.001$ | $0.430 \pm 0.002$ |
| ETTh2 | $0.334 \pm 0.005$ | $0.383 \pm 0.009$ | $0.330 \pm 0.002$ | $0.379 \pm 0.007$ |
| ETTm1 | $0.329 \pm 0.006$ | $0.372 \pm 0.007$ | $0.351 \pm 0.006$ | $0.380 \pm 0.002$ |
| ETTm2 | $0.251 \pm 0.002$ | $0.313 \pm 0.003$ | $0.255 \pm 0.003$ | $0.315 \pm 0.002$ |
| Weather | $0.225 \pm 0.009$ | $0.257 \pm 0.008$ | $0.225 \pm 0.001$ | $0.264 \pm 0.001$ |
| Electricity | $0.158 \pm 0.004$ | $0.252 \pm 0.007$ | $0.161 \pm 0.001$ | $0.252 \pm 0.001$ |
| Traffic | $0.388 \pm 0.001$ | $0.264 \pm 0.006$ | $0.390 \pm 0.003$ | $0.263 \pm 0.003$ |
| ILI | $1.435 \pm 0.011$ | $0.801 \pm 0.008$ | $1.443 \pm 0.012$ | $0.797 \pm 0.002$ |

Table 20: Standard deviations of our TIME-LLM and the second-best method (N-HiTS) on M4 datasets for short-term forecasting.

| Model | TIME-LLM | | | N-HiTS (2023a) | | |
|---|---|---|---|---|---|---|
| Dataset | SMAPE | MAPE | OWA | SMAPE | MAPE | OWA |
| Yearly | $13.419 \pm 0.117$ | $3.005 \pm 0.011$ | $0.789 \pm 0.003$ | $13.422 \pm 0.009$ | $3.056 \pm 0.017$ | $0.795 \pm 0.010$ |
| Quarterly | $10.110 \pm 0.107$ | $1.178 \pm 0.009$ | $0.889 \pm 0.007$ | $10.185 \pm 0.107$ | $1.180 \pm 0.007$ | $0.893 \pm 0.001$ |
| Monthly | $12.980 \pm 0.102$ | $0.963 \pm 0.005$ | $0.903 \pm 0.001$ | $13.059 \pm 0.101$ | $1.013 \pm 0.007$ | $0.929 \pm 0.005$ |
| Others | $4.795 \pm 0.117$ | $3.178 \pm 0.012$ | $1.006 \pm 0.009$ | $4.711 \pm 0.117$ | $3.054 \pm 0.011$ | $0.997 \pm 0.012$ |
| Averaged | $11.983 \pm 0.011$ | $1.595 \pm 0.021$ | $0.859 \pm 0.002$ | $12.035 \pm 0.111$ | $1.625 \pm 0.012$ | $0.869 \pm 0.005$ |

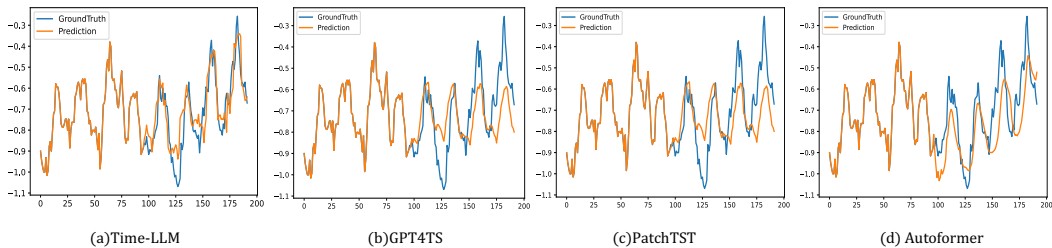

Figure 7: Long-term forecasting cases from ETTh1 by different models under the input-96-predict-96 settings. Blue lines are the ground truths and orange lines are the model predictions.

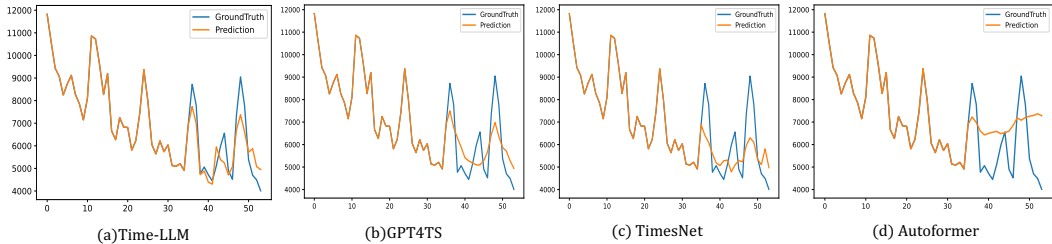

Figure 8: Short-term forecasting from the M4 dataset by different models under the input-36-predict-18 settings.

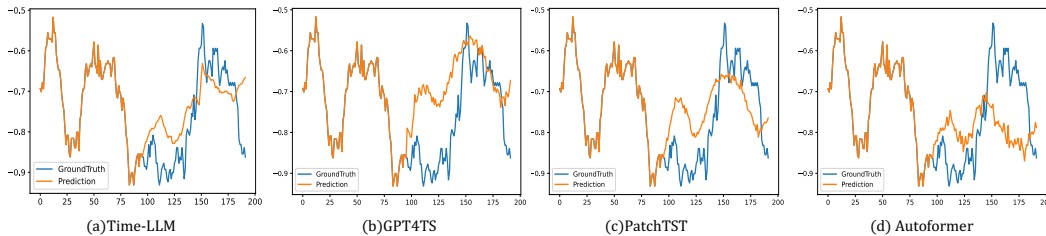

Figure 9: Few-shot forecasting cases from ETTm1 by different models under the input-96-predict-96 settings. Blue lines are the ground truths and orange lines are the model predictions.

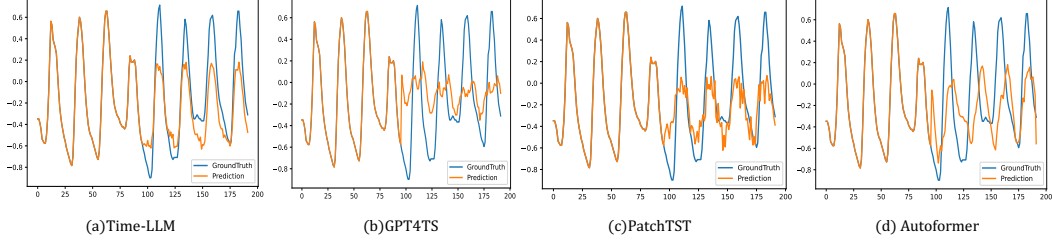

Figure 10: Zero-shot forecasting cases from ETTh1→ETTh2 by different models under the input-96-predict-96 settings. Blue lines are the ground truths and orange lines are the model predictions.

