# OpenReview forum: "Time-LLM: Time Series Forecasting by Reprogramming Large Language Models"
_ICLR.cc/2024/Conference — ICLR 2024 poster_

### Official Review · Reviewer_baDP · 2023-10-31

**Soundness:** 4 excellent
**Presentation:** 4 excellent
**Contribution:** 4 excellent
**Rating:** 8
**Confidence:** 4

**Summary:**

This paper introduces a pioneering approach, Time-LLM, which harnesses the reprogramming and Prompt-as-Prefix techniques to repurpose large language models for time series forecasting while keeping the backbone LLM intact. The method innovatively bridges input time series to optimized text prototypes, making the time series inputs more digestible for language models. This integration enables large language models to seamlessly tackle time series forecasting. Furthermore, Time-LLM incorporates the Prompt-as-Prefix mechanism to facilitate the LLM's reasoning capabilities over time series. By adding natural language prompts, the approach enriches time series input and presents task directives in a comprehensible language format. In my view, this holds substantial promise for controlled time series analysis across diverse applications. The authors have conducted extensive experiments to demonstrate the effectiveness and efficiency of the proposed reprogramming framework, showing considerable potential in leveraging LLMs for time series tasks.

**Strengths:**

1.	The paper is articulately composed and well-organized, with most concepts clearly presented. It offers an in-depth exploration of the proposed concepts related to LLM reprogramming and Prompt-as-Prefix.

2.	The proposed reprogramming framework is novel and has proven to be highly effective. Instead of directly feeding original time series into LLMs, the innovative approach of converting time series into text prototype representations for language model comprehension stands out. This strategy might set the foundation for a broader method of cross-modality adaptation in LLMs.

3.	The augmentation of input context with declarative prompts, such as domain expertise and task guidelines, to steer LLM reasoning is notably promising. It offers significant potential for controlled time series analysis in a variety of applications.

4.	The architectural design of Time-LLM is logical and underpinned by clear motivations. Dividing time series into patches and reprogramming each into text prototype representations is a wise choice, which aligns well with the use of natural language prompts to guide LLM in time series reasoning.

5.	Comprehensive experimental results are provided to evaluate the proposed reprogramming framework from various aspects. Indeed, Time-LLM demonstrates promising results, especially under the few-shot and zero-shot protocols, showing considerable sample efficiency and applicability in real-word applications. There are also abundant ablation and other side experiments to study the proposed method from various aspects.

**Weaknesses:**

1.	While this paper is generally well-written, there are areas that could benefit from further refinement. For instance, Fig. 5 lacks clarity and might benefit from being displayed at a larger scale to improve visibility. In patch reprogramming (Sec. 3.1), it would be beneficial to illustrate how the linear projection aligns the hidden dimensions in Fig. 2. Within the Prompt-as-Prefix discussion, reference is made to the inclusion of statistics within the prompt. Elaborating on the specific content and the calculation methods employed would enhance clarity.

2.	The paper's foundational concept hinges on the idea of reprogramming. However, it omits references to several pivotal reprogramming studies from recent years (as listed below). Integrating these pertinent studies in the introductory or related works section could provide a more robust understanding of the concept.

3.	There are some writing issues. For instance, there should be uniformity in the references' formatting, with a preference for citing formally published works from conferences or journals over preprints.

[1] Vinod, R., Chen, P. Y., & Das, P. (2020). Reprogramming Language Models for Molecular Representation Learning. In Annual Conference on Neural Information Processing Systems.

[2] Melnyk, I., Chenthamarakshan, V., Chen, P., Das, P., Dhurandhar, A., Padhi, I., & Das, D. (2022). Reprogramming Pretrained Language Models for Antibody Sequence Infilling. International Conference on Machine Learning.

**Questions:**

1.	Could you elucidate the mechanism behind "Output Projection" (Sec. 3.1), particularly the aspects related to flattening and the linear projection? A formulaic representation would greatly aid in understanding.

2.	In the "Prompt-as-Prefix" paragraph, there is a reference to computing the trend and lag in relation to time series, with several potential implementations hinted at. Could the authors detail the methodology used to determine the trend and lag information within the prompts?

---

> ### Author Response · Authors · 2023-11-18
> **Response to Reviewer baDP**
>
> We sincerely thank the reviewer for their valuable suggestions to further improve our paper. We are appreciative of the reviewer's recognition of our contributions as innovative and significant in illuminating future research across a wide range of applications. Below, we address the primary concerns raised by the reviewer.
>
> > W1. While this paper is generally well-written, there are areas that could benefit from further refinement. For instance, Fig. 5 lacks clarity and might benefit from being displayed at a larger scale to improve visibility. In patch reprogramming (Sec. 3.1), it would be beneficial to illustrate how the linear projection aligns the hidden dimensions in Fig. 2. Within the Prompt-as-Prefix discussion, reference is made to the inclusion of statistics within the prompt. Elaborating on the specific content and the calculation methods employed would enhance clarity.
>
> + We agree with the reviewer's suggestion to highlight Fig. 5 in a larger size. However, due to space constraints in the main text, scaling up Fig. 5 poses challenges as it may compromise the presentation of other results and discussions.
> + In response to the suggestion of depicting the transformation from $\mathbf{Z}^{(i)}$ to $\mathbf{O}^{(i)}$ in Fig. 2, **we have slightly revised this figure in the part of "Patch Reprogram"**.
> + To construct the statistical prompting block shown in Fig. 4, we compute the following representative statistics: (1) the minimum, maximum, and median values of the given time series; (2) the overall trend of the time series, determined by calculating the sum of differences between consecutive time steps. A sum greater than 0 indicates an upward trend, while a lesser sum denotes a downward trend; (3) the top-5 lags of the time series, identified by computing the autocorrelation using fast Fourier transformation and selecting the five lags with the highest correlation values. Please refer to our code for more details, which has been made available to AC and all reviewers. **We have also revised the paper accordingly and included this technical detail in the appendix.**
>
> > W2. The paper's foundational concept hinges on the idea of reprogramming. However, it omits references to several pivotal reprogramming studies from recent years (as listed below). Integrating these pertinent studies in the introductory or related works section could provide a more robust understanding of the concept.
>
> Thank you for the suggestion. To enhance comprehension of model reprogramming and more related works, **we have revised and expanded the discussion on cross-modality adaptation in Appendix A**.
>
> > W3. There are some writing issues. For instance, there should be uniformity in the references' formatting, with a preference for citing formally published works from conferences or journals over preprints.
>
> Thank you for the suggestion. **We have thoroughly reviewed and updated the reference list, including both the format and venues of the references**. Additionally, considering the rapid growth in LLMs and related studies, many references in our work are recent preprints from 2023. These references are crucial in offering the necessary background knowledge for a better understanding of the proposed method.
>
> > Q1. Could you elucidate the mechanism behind "Output Projection" (Sec. 3.1), particularly the aspects related to flattening and the linear projection? A formulaic representation would greatly aid in understanding.
>
> Upon packing and feedforwarding the prompt and patch embeddings $\mathbf{O}^{i}$ through the frozen LLM, we discard the prefixal part and obtain the output representations, denoted as $\tilde{\mathbf{O}}^{i}$. Subsequently, we follow PatchTST [1] and flatten $\tilde{\mathbf{O}}^{i}$ into a 1D tensor with the length $P \times D$, which is then linear projected as $\hat{\mathbf{Y}}^{i} \in \mathbb{R}^H$. Due to space constraints in the main text, **we have revised the paper accordingly and included this technical detail in Appendix B.1**.
>
> > Q2. In the "Prompt-as-Prefix" paragraph, there is a reference to computing the trend and lag in relation to time series, with several potential implementations hinted at. Could the authors detail the methodology used to determine the trend and lag information within the prompts?
>
> See our response above to W1.
>
> We hope our responses provided above can adequately address the reviewer's concerns and questions.

---

> > ### Comment · Reviewer_baDP · 2023-11-21
> >
> > Thanks to the authors for their great effort in clarifying my questions. The substantial addition of new experimental results further strengthens the paper. I will maintain my rating, and I believe a score of 8 is well-deserved.

---

> > > ### Author Response · Authors · 2023-11-23
> > > **Sincere Gratitude from Authors**
> > >
> > > We are thrilled that our responses have effectively addressed all your questions and comments. We would like to express our sincerest gratitude for taking the time to review our paper and provide us with such detailed and invaluable comments.

---

### Official Review · Reviewer_cwaB · 2023-11-01

**Soundness:** 2 fair
**Presentation:** 3 good
**Contribution:** 3 good
**Rating:** 8
**Confidence:** 3

**Summary:**

This work casts time series modeling as yet another "language" task by addressing the problem of casting continuous-valued data into a discrete representation and leveraging prompts to the language model. Here the authors draw from previous works, including PatchTST, to patch the continuous valued time series data and project them onto word embeddings via an attention mechanism. The new discretized embeddings and a word prompt are fed into a pretrained LLM which predicts future values. This work produces state-of-the-art results on both short-term and long-term predictions on the M4 and electricity transformer temperature (ETT) datasets, respectively. The work also produces the state-of-the-art results on zero and few-shot learning evaluated on ETT.

**Strengths:**

Instead of creating a new representation of the continous-valued time series data the authors discretized the data by projecting it onto existing word embeddings via an attention mechanism. This approach is exciting in it's use of the word embeddings as the discretization medium rather than using a linear layer in the final calculation of the embeddings. In doing so, this work shows how using not only trained LLMs but also the learned embeddings can transfer to time series problems.

By building the final embeddings from word embeddings, the authors cast the time-series problem into a "language" problem. This allows the authors to further utilize language model features by employing a prompt which is shown to improve time series predictions. One could imagine expanding further on this new property by trying different prompts and seeing if such prompts can illicit adversarial results. Ultimately, this approach may be utilized for new experiments to both probe LLMs and further exploit their capabilities.

**Weaknesses:**

The authors chose a limited set of time series data to benchmark this model on. Many of the models they compare against have been benchmarked against other data sets: weather, traffic, electricity consumption, and the spread of influenza. This confuses the objective of the paper. If the objective is to show this model gives the state-of-the-art results for the two data sets then that is clear. If the authors wish to claim this model outperforms others across many domains then more evidence is needed. If the authors wish to introduce this model and benchmark it on a single data set to demonstrate its potential capabilities then they provide many examples.

**Questions:**

My primary concern is that the evidence provided, the stated intentions, and the claims of this work do not fully line up. I would like to know the primary objective and claims of this work. Depending on that answer I feel there might be more evidence required.

a) If this work is meant to show that the model is state-of-the-art across many potential time series tasks then comparisons across other datasets (mentioned above) would be much more convincing. This work only uses ETT and M4.

b) If this work is meant to benchmark this model's capabilities on a single data set then please make that clear. The state-of-the-art claims should only be made with regard to the data sets the model was benchmarked against.

---

> ### Author Response · Authors · 2023-11-17
> **Response to Reviewer cwaB (Part 1)**
>
> We extend our sincere thanks to the reviewer for their constructive feedback on our paper. We are deeply grateful for the recognition of our contributions and the significant potential of this work. Below, we have addressed the main concern raised by the reviewer.
>
> > W1. The authors chose a limited set of time series data to benchmark this model on. Many of the models they compare against have been benchmarked against other data sets: weather, traffic, electricity consumption, and the spread of influenza. a) If this work is meant to show that the model is state-of-the-art across many potential time series tasks then comparisons across other datasets (mentioned above) would be much more convincing. This work only uses ETT and M4. b) If this work is meant to benchmark this model's capabilities on a single data set then please make that clear. The state-of-the-art claims should only be made with regard to the data sets the model was benchmarked against.
>
> + Our primary objective in this work is extending the capabilities of large language models to practical time series forecasting within a model reprogramming space. We also mean to evidence that Time-LLM consistently outperforms or matches the state-of-the-art performance in mainstream forecasting tasks, especially in few-shot and zero-shot scenarios. We believe this falls into the category (a) mentioned by the reviewer.
>
> + We have performed additional experiments on four datasets: (1) Weather, (2) Electricity, (3) Traffic, and (4) Influenza-like Illness (ILI). **Our detailed results can be found in the main text (i.e., Tables 1,3, and 4) and appendices (i.e., Tables 10,11,14, and 15) of the revised paper**. For the reviewer's convenience, *a very brief summary* is provided below, highlighting the overall performance on a set of most representative methods on additionaly added four datasets.
>
>   + A concise summary of additional long-term forecasting results in average on Weather, Electricity, Traffic, and Influenza-like Illness (ILI):
>
>   | Method      | Time-LLM  |                     | GPT4TS |       | DLinear |       | PatchTST            |                     | TimesNet |       | FEDformer |       |
>   | ----------- | --------- | ------------------- | ------ | ----- | ------- | ----- | ------------------- | ------------------- | -------- | ----- | --------- | ----- |
>   | Metric      | MSE       | MAE                 | MSE    | MAE   | MSE     | MAE   | MSE                 | MAE                 | MSE      | MAE   | MSE       | MAE   |
>   | Weather     | **0.225** | **0.257**           | 0.237  | 0.270 | 0.248   | 0.300 | **0.225**           | $\underline{0.264}$ | 0.259    | 0.287 | 0.309     | 0.360 |
>   | Electricity | **0.158** | **0.252**           | 0.167  | 0.263 | 0.166   | 0.263 | $\underline{0.161}$ | **0.252**           | 0.192    | 0.295 | 0.214     | 0.327 |
>   | Traffic     | **0.388** | $\underline{0.264}$ | 0.414  | 0.294 | 0.433   | 0.295 | $\underline{0.390}$ | **0.263**           | 0.620    | 0.336 | 0.610     | 0.376 |
>   | ILI         | **1.435** | $\underline{0.801}$ | 1.925  | 0.903 | 2.169   | 1.041 | $\underline{1.443}$ | **0.797**           | 2.139    | 0.931 | 2.847     | 1.144 |

---

> > ### Author Response · Authors · 2023-11-18
> > **Response to Reviewer cwaB (Part 2)**
> >
> > + A concise summary of additional 10% few-shot learning results in average on Weather, Electricity, Traffic, and Influenza-like Illness (ILI):
> >
> >   | Method      | Time-LLM  |                     | GPT4TS              |                     | DLinear |       | PatchTST            |           | TimesNet |       | FEDformer |       |
> >   | ----------- | --------- | ------------------- | ------------------- | ------------------- | ------- | ----- | ------------------- | --------- | -------- | ----- | --------- | ----- |
> >   | Metric      | MSE       | MAE                 | MSE                 | MAE                 | MSE     | MAE   | MSE                 | MAE       | MSE      | MAE   | MSE       | MAE   |
> >   | Weather     | **0.234** | **0.273**           | $\underline{0.238}$ | $\underline{0.275}$ | 0.241   | 0.283 | 0.242               | 0.279     | 0.279    | 0.301 | 0.284     | 0.324 |
> >   | Electricity | **0.175** | $\underline{0.270}$ | $\underline{0.176}$ | **0.269**           | 0.180   | 0.280 | 0.180               | 0.273     | 0.323    | 0.392 | 0.346     | 0.427 |
> >   | Traffic     | **0.429** | $\underline{0.306}$ | 0.440               | 0.310               | 0.447   | 0.313 | $\underline{0.430}$ | **0.305** | 0.951    | 0.535 | 0.663     | 0.425 |
> >
> >   + A concise summary of additional 5% few-shot learning results in average on Weather, Electricity, Traffic, and Influenza-like Illness (ILI):
> >
> >   | Method      | Time-LLM            |                     | GPT4TS              |                     | DLinear |       | PatchTST  |           | TimesNet |       | FEDformer |       |
> >   | ----------- | ------------------- | ------------------- | ------------------- | ------------------- | ------- | ----- | --------- | --------- | -------- | ----- | --------- | ----- |
> >   | Metric      | MSE                 | MAE                 | MSE                 | MAE                 | MSE     | MAE   | MSE       | MAE       | MSE      | MAE   | MSE       | MAE   |
> >   | Weather     | **0.260**           | 0.309               | $\underline{0.263}$ | **0.301**           | 0.263   | 0.308 | 0.269     | 0.303     | 0.298    | 0.318 | 0.309     | 0.353 |
> >   | Electricity | $\underline{0.179}$ | **0.268**           | **0.178**           | $\underline{0.273}$ | 0.176   | 0.275 | 0.181     | 0.277     | 0.402    | 0.453 | 0.266     | 0.353 |
> >   | Traffic     | $\underline{0.423}$ | $\underline{0.298}$ | 0.434               | 0.305               | 0.450   | 0.317 | **0.418** | **0.296** | 0.867    | 0.493 | 0.676     | 0.423 |
> >
> > + For more experimental results, **see our revised Appendix D and E**. We have also carried out a new set of short-term forecasting experiments on the M3-Quarterly benchmark, **with the results presented in Table 13 of the revised paper**. Furthermore, comparisons with extra baseline methods were performed, and **these results are detailed in Tables 11 and 16 of the revised paper**.
> >
> > **We have made our best effort to include the experiments suggested by the reviewer and have revised the paper accordingly**. We trust that these modifications address the concern raised by the reviewer.

---

> > > ### Author Response · Authors · 2023-11-20
> > > **Request of Reviewer's attention and feedback**
> > >
> > > Dear Reviewer,
> > >
> > > Thank you for your valuable and constructive feedback, which has inspired further improvements to our paper. As a gentle reminder, it has been more than 2 days since we submitted our rebuttal. We would like to know whether our response addressed your concerns.
> > >
> > > In accordance with your comments, we have answered your concerns and made the following revisions:
> > >
> > > + We have made it clear that **our primary objective** is extending the capabilities of large language models to practical time series forecasting within a model reprogramming space. We also mean to evidence that **Time-LLM consistently outperforms or matches the state-of-the-art performance in mainstream forecasting tasks**, especially in few-shot and zero-shot scenarios.
> > >
> > > + **We have conducted additional experiments (amounting to over 1,000 new results in total) and revised the paper accordingly**, along with the relevant discussions. You may find our responses to Reviewer Z9NA helpful. In a nutshell, the following additional experiments are incorporated:
> > >   + We performed additional experiments on (1) Weather, (2) Electricity, (3) Traffic, and (4) Influenza-like Illness (ILI). Our detailed results can be found in the main text (Tabs. 1,3, and 4) and appendices (Tabs. 10,11,14, and 15) of the revised paper.
> > >   + We included AutoARIMA, AutoTheta, and AutoETS as baseline methods. The detailed results can be found in Tab. 11 of the revised paper.
> > >   + We compared to N-BEATS and N-HITS in long-term forecasting tasks. The detailed results are in Tab. 11 of the revised paper.
> > >   + We also included the comparison with LLMTime, even though this is completely optional according to the author guide this year. The results we have incorporated can be found in Tabs. 5 and 16.
> > >   + We have included additional experiments on M3-Quarterly for short-term forecasting due to rebuttal time constraints. Our results can be found in Table 13 of the revised paper.
> > >
> > > We again thank you for your insightful review. We eagerly await your feedback and are ready to respond to any further questions you may have.

---

> > > > ### Comment · Reviewer_cwaB · 2023-11-22
> > > > **Response to rebuttal**
> > > >
> > > > Thank you for thoroughly addressing my concerns. I would like for the authors to the statement above in the main text as it would have helped improve clarity: "Our primary objective in this work is extending the capabilities of large language models to practical time series forecasting within a model reprogramming space. We also mean to evidence that Time-LLM consistently outperforms or matches the state-of-the-art performance in mainstream forecasting tasks, especially in few-shot and zero-shot scenarios."
> > > >
> > > > At this time I do not have any more concerns related to my review and I am inclined to change my referral to accept. I would like to discuss with reviewer L3pe about their concerns with regard to selecting the best test score. Therefore my final referral will be contingent on that.

---

> > > > > ### Author Response · Authors · 2023-11-23
> > > > > **Sincere Gratitude from Authors**
> > > > >
> > > > > We are delighted that our responses have effectively addressed all your concerns. Thank you for taking the time to review our paper and provide us with such detailed and valuable comments. We look forward to your favorable support during the following PC/AC discussion phase if needed.

---

### Official Review · Reviewer_MAkL · 2023-11-01

**Soundness:** 3 good
**Presentation:** 3 good
**Contribution:** 3 good
**Rating:** 8
**Confidence:** 4

**Summary:**

The paper presents Time-LLM, a framework designed to harness Large Language Models (LLMs) for time series forecasting. By converting time series data into text-like prototypes and introducing the Prompt-as-Prefix (PaP) method to supplement this data with additional context, the framework effectively aligns time series data with the modalities of natural language. The empirical results indicate Time-LLM's superiority in comparison to other leading models, particularly in few-shot and zero-shot contexts.

**Strengths:**

1. The paper is articulate and systematically structured, making the motivation and methodology behind the proposed solution evident.
2. The approach of modality alignment from time series to natural language is both innovative and promising, offering a new perspective for future research.
3. The empirical evaluation is thorough, encompassing an analysis of different LLM variations, an ablation study, computational efficiency considerations, and model interpretation.

**Weaknesses:**

**Major**
1. The choice of datasets for evaluation is restrictive, as the ETT datasets involve similar metrics monitored under different conditions. Their mutual similarities might overinflate the perceived performance of Time-LLM. Inclusion of diverse datasets such as Weather, Electricity, and Traffic, commonly featured in literature, would offer a more holistic assessment. Moreover, there's an emerging consensus that long-term forecasting benchmarks have a preference for univariate models, potentially bypassing the capability of handling cross-variate correlations  ([1], [2]). As Time-LLM also processes each channel separately, this limitation should be discussed.
2. The mechanism used to produce the next H steps remains unclear, warranting a more detailed explanation.

**Minor**
1. Figure 3 Ambiguities:
    1. Fig 3(a): Initially, the patches seem to represent input patches $X_P$. To eliminate any confusion, clearly labeling associated variables like $E, Z$ would be helpful.
    2. Fig 3(b): The figure raises questions about whether the model outputs only the subsequent step or the next $H$ steps. Additionally, the function of the intermediate layer remains undefined.
2. Error in Fig 3(b): There seems to be a need for a one-step left shift in the output of Patch-as-Prefix to render it an auto-regressive model.

I would be glad to raising my score if the aforementioned weaknesses are addressed.

[1] Chen, Si-An, et al. "TSMixer: An All-MLP Architecture for Time Series Forecasting." Transactions on Machine Learning Research. 2023

[2] Das, Abhimanyu, et al. "Long-term Forecasting with TiDE: Time-series Dense Encoder." Transactions on Machine Learning Research. 2023

**Questions:**

1. Does the model operate in an autoregressive manner, predicting only the subsequent step, or does it directly forecast the next $H$ steps?
2. How is $E'$ derived from $E$?

---

> ### Author Response · Authors · 2023-11-17
> **Response to Reviewer MAkL (Part 1)**
>
> We express our sincere gratitude to the reviewer for providing valuable feedback on our paper. We are deeply appreciative of the acknowledgment of the contributions and significance of our work. To our best effort, we have addressed the concerns and questions raised by the reviewer below.
>
> > W1.1. Inclusion of diverse datasets such as Weather, Electricity, and Traffic, commonly featured in literature, would offer a more holistic assessment.
>
> We have performed additional experiments on four datasets: (1) Weather, (2) Electricity, (3) Traffic, and (4) Influenza-like Illness (ILI). **Our detailed results can be found in the main text (i.e., Tables 1,3, and 4) and appendices (i.e., Tables 10,11,14, and 15) of the revised paper**. For the reviewer's convenience, *a very brief summary* is provided below, highlighting the overall performance on a set of most representative methods on additionaly added four datasets.
>
> + A concise summary of additional long-term forecasting results in average on Weather, Electricity, Traffic, and Influenza-like Illness (ILI):
>
>   | Method      | Time-LLM  |                     | GPT4TS |       | DLinear |       | PatchTST            |                     | TimesNet |       | FEDformer |       |
>   | ----------- | --------- | ------------------- | ------ | ----- | ------- | ----- | ------------------- | ------------------- | -------- | ----- | --------- | ----- |
>   | Metric      | MSE       | MAE                 | MSE    | MAE   | MSE     | MAE   | MSE                 | MAE                 | MSE      | MAE   | MSE       | MAE   |
>   | Weather     | **0.225** | **0.257**           | 0.237  | 0.270 | 0.248   | 0.300 | **0.225**           | $\underline{0.264}$ | 0.259    | 0.287 | 0.309     | 0.360 |
>   | Electricity | **0.158** | **0.252**           | 0.167  | 0.263 | 0.166   | 0.263 | $\underline{0.161}$ | **0.252**           | 0.192    | 0.295 | 0.214     | 0.327 |
>   | Traffic     | **0.388** | $\underline{0.264}$ | 0.414  | 0.294 | 0.433   | 0.295 | $\underline{0.390}$ | **0.263**           | 0.620    | 0.336 | 0.610     | 0.376 |
>   | ILI         | **1.435** | $\underline{0.801}$ | 1.925  | 0.903 | 2.169   | 1.041 | $\underline{1.443}$ | **0.797**           | 2.139    | 0.931 | 2.847     | 1.144 |
>
> + A concise summary of additional 10% few-shot learning results in average on Weather, Electricity, Traffic, and Influenza-like Illness (ILI):
>
>   | Method      | Time-LLM  |                     | GPT4TS              |                     | DLinear |       | PatchTST            |           | TimesNet |       | FEDformer |       |
>   | ----------- | --------- | ------------------- | ------------------- | ------------------- | ------- | ----- | ------------------- | --------- | -------- | ----- | --------- | ----- |
>   | Metric      | MSE       | MAE                 | MSE                 | MAE                 | MSE     | MAE   | MSE                 | MAE       | MSE      | MAE   | MSE       | MAE   |
>   | Weather     | **0.234** | **0.273**           | $\underline{0.238}$ | $\underline{0.275}$ | 0.241   | 0.283 | 0.242               | 0.279     | 0.279    | 0.301 | 0.284     | 0.324 |
>   | Electricity | **0.175** | $\underline{0.270}$ | $\underline{0.176}$ | **0.269**           | 0.180   | 0.280 | 0.180               | 0.273     | 0.323    | 0.392 | 0.346     | 0.427 |
>   | Traffic     | **0.429** | $\underline{0.306}$ | 0.440               | 0.310               | 0.447   | 0.313 | $\underline{0.430}$ | **0.305** | 0.951    | 0.535 | 0.663     | 0.425 |
>
> + A concise summary of additional 5% few-shot learning results in average on Weather, Electricity, Traffic, and Influenza-like Illness (ILI):
>
>   | Method      | Time-LLM            |                     | GPT4TS              |                     | DLinear |       | PatchTST  |           | TimesNet |       | FEDformer |       |
>   | ----------- | ------------------- | ------------------- | ------------------- | ------------------- | ------- | ----- | --------- | --------- | -------- | ----- | --------- | ----- |
>   | Metric      | MSE                 | MAE                 | MSE                 | MAE                 | MSE     | MAE   | MSE       | MAE       | MSE      | MAE   | MSE       | MAE   |
>   | Weather     | **0.260**           | 0.309               | $\underline{0.263}$ | **0.301**           | 0.263   | 0.308 | 0.269     | 0.303     | 0.298    | 0.318 | 0.309     | 0.353 |
>   | Electricity | $\underline{0.179}$ | **0.268**           | **0.178**           | $\underline{0.273}$ | 0.176   | 0.275 | 0.181     | 0.277     | 0.402    | 0.453 | 0.266     | 0.353 |
>   | Traffic     | $\underline{0.423}$ | $\underline{0.298}$ | 0.434               | 0.305               | 0.450   | 0.317 | **0.418** | **0.296** | 0.867    | 0.493 | 0.676     | 0.423 |

---

> > ### Author Response · Authors · 2023-11-17
> > **Response to Reviewer MAkL (Part 2)**
> >
> > > W1.2. There's an emerging consensus that long-term forecasting benchmarks have a preference for univariate models, potentially bypassing the capability of handling cross-variate correlations. As Time-LLM also processes each channel separately, this limitation should be discussed.
> >
> > Thank you for sharing this perspective. However, **the use of channel independence is not a fundamental limitation of Time-LLM**. Our rationale is twofold: (1) channel independence is not the main focus in this work. Our primary objective is extending the capabilities of large language models to time series forecasting within a reprogramming space. **We have not made any claims or asserted any capabilities in handling cross-variate correlations**; (2) In Time-LLM, both channel independence and mixing are technically feasible, but **channel mixing may not always a good idea**. For instance, when reprogramming the LLM across different time series datasets, channel mixing faces two primary challenges: i) the number of channels often varies across datasets; ii) using a shared embedding layer to process channels from different datasets, which may have significantly different semantics, can be impractical.
> >
> > > W2. The mechanism used to produce the next $H$ steps remains unclear, warranting a more detailed explanation.
> >
> > + Time-LLM does not operate in an autoregressive manner.
> > + Upon packing and feedforwarding the prompt and patch embeddings $\mathbf{O}^{i}$ through the frozen LLM, we discard the prefixal part and obtain the output representations, denoted as $\tilde{\mathbf{O}}^{i}$. Subsequently, we follow PatchTST [1] and flatten $\tilde{\mathbf{O}}^{i}$ into a 1D tensor with the length $P \times D$, which is then linear projected as $\hat{\mathbf{Y}}^{i} \in \mathbb{R}^{H}$. **We have revised the paper accordingly and included this technical detail in Appendix B.1**, owing to space constraints in the main text.
> >
> > + Note that the left figure in Fig. 3(b), i.e., Patch-as-Prefix, is not in our method. This is only for a clearer illustration of our discussion related to Prompt-as-Prefix in Sec. 3.1.
> >
> > [1] Nie, Y., Nguyen, N. H., Sinthong, P., & Kalagnanam, J. (2022, September). A Time Series is Worth 64 Words: Long-term Forecasting with Transformers. In The Eleventh International Conference on Learning Representations.
> >
> > > W3.1. Fig 3(a): Initially, the patches seem to represent input patches $X_P$. To eliminate any confusion, clearly labeling associated variables like $E$, $Z$ would be helpful.
> >
> > Thank you for the suggestion. **We have appropriately revised Fig. 3(a)** and updated it with the corresponding notations and remarks.
> >
> > > W3.2. Fig 3(b): The figure raises questions about whether the model outputs only the subsequent step or the next $H$ steps. Additionally, the function of the intermediate layer remains undefined.
> >
> > + Apologies for any confusion caused. As previously stated, the left figure in Fig. 3(b), representing Patch-as-Prefix, **is not part of our method**. It is included solely for a clearer explanation of our discussion on Prompt-as-Prefix in Sec. 3.1.
> > + We believe that our discussion under the part of "Prompt-as-Prefix" in Sec. 3.1 effectively illustrates: (1) the connections and differences between the two paradigms; (2) the main challenge of the Patch-as-Prefix approach in time series forecasting; and (3) the advantages of our proposed Prompt-as-Prefix method.
> > + For the definition of the projection head depicted in the right side of Fig. 3(b), please refer to our response above addressing W2.
> >
> > > W3.3. Error in Fig 3(b): There seems to be a need for a one-step left shift in the output of Patch-as-Prefix to render it an auto-regressive model.
> >
> > This is correct, but it is important to note that: (1) the left figure in Fig. 3(b) illustrates the training process, and (2) this figure serves only for high-level illustrative purposes and is not the primary focus of our study as mentioned above. Here we adopted a widely recognized style used in [1] (as cited in our paper) to facilitate a clearer understanding of the differences between the two paradigms.
> >
> > [1] Tsimpoukelli, M., Menick, J. L., Cabi, S., Eslami, S. M., Vinyals, O., & Hill, F. (2021). Multimodal few-shot learning with frozen language models. Advances in Neural Information Processing Systems, 34, 200-212.

---

> > > ### Author Response · Authors · 2023-11-17
> > > **Response to Reviewer MAkL (Part 3)**
> > >
> > > > Q1. Does the model operate in an autoregressive manner, predicting only the subsequent step, or does it directly forecast the next $H$ steps?
> > >
> > > See our reponse to W2 and W3.2.
> > >
> > > > Q2. How is $E'$ derived from $E$?
> > >
> > > This is accomplished through a simple linear projection as mentioned in the paper and Fig. 2. Specifcally, given $\mathbf{E} \in \mathbb{R}^{V \times D}$, we learn a weight matrix $\mathbf{W} \in \mathbb{R}^{V' \times V}$ to identify a small set of text prototypes in $\mathbf{E}'$. **We have revised the paper accordingly and included this technical detail in Appendix B.1** due to space constraints in the main text.
> > >
> > > We hope our responses provided above can adequately address the reviewer's concerns and questions.

---

> > > > ### Author Response · Authors · 2023-11-20
> > > > **Request of Reviewer's attention and feedback**
> > > >
> > > > Dear Reviewer,
> > > >
> > > > Thank you for your valuable and constructive feedback, which has inspired further improvements to our paper. As a gentle reminder, it has been more than 2 days since we submitted our rebuttal. We would like to know whether our response addressed your concerns.
> > > >
> > > > Following your comments and suggestions, we have answered your concerns and made the following revisions:
> > > >
> > > > + We have performed **additional experiments** on four datasets: (1) **Weather**, (2) **Electricity**, (3) **Traffic**, and (4) **Influenza-like Illness (ILI)**. Our detailed results can be found in the main text (Tabs 1,3, and 4) and appendices (Tabs. 10,11,14, and 15) of the revised paper.
> > > >
> > > > + We have provided clear justifications on (1) **why channel independence does not constitute a fundamental limitation of Time-LLM**, and (2) **why channel mixing might not always be beneficial in models like ours**.
> > > > + **We have made necessary updates to Fig. 3(a)**, enriching it with relevant notations and explanations.
> > > > + Addressing your questions about forecast generation, we have clarified that (1) **Time-LLM does not function in an autoregressive manner**; (2) **we explain the mechanism for producing the next H steps and have accordingly revised Appendix B.1** to include more technical details.
> > > > + Concerning queries related to Fig. 2(b), we provide further clarifications: (1) **the left figure in Fig. 3(b), depicting Patch-as-Prefix, is not a component of our method**. It is included mainly for an enhanced understanding of our discussion on Prompt-as-Prefix in Sec. 3.1; (2) the left figure in Fig. 3(b) demonstrates the training process, and **this representation is for high-level illustrative purposes, widely recognized in the field**, and not the central focus of our research.
> > > > + **We have responded to the question regarding the learning of text prototypes and have updated Appendix B.1** with additional technical insights.
> > > >
> > > > We again thank you for your insightful review. We eagerly await your feedback and are ready to respond to any further questions you may have.

---

> > > > > ### Comment · Reviewer_MAkL · 2023-11-20
> > > > >
> > > > > Thank you for addressing my concerns in your rebuttal. The additional experiments and detailed responses have significantly improved the clarity and persuasiveness of the paper. I have accordingly increased my score to reflect this positive development.
> > > > >
> > > > > Regarding the handling of cross-variate features, I agree that the utilization of LLMs for forecasting presents a valuable contribution. As you mentioned, the added complexity of incorporating cross-variate features into the model may not always translate into improved performance on academic benchmarks. However, in real-world and industrial settings, time series data often exhibits sparsity and intermittency while also encompassing multiple variables. The ability to effectively handle cross-variate features is crucial for addressing these challenges. Therefore, I still encourage the authors to discuss on this limitation in the paper or consider elaborate it as a potential direction for future work, advancing the community awareness about this issue.

---

> > > > > > ### Author Response · Authors · 2023-11-20
> > > > > > **Sincere Gratitude from Authors**
> > > > > >
> > > > > > We are truly delighted that our responses have effectively addressed your concerns. We would like to express our sincerest gratitude once again for taking the time to review our paper and provide us with such detailed and invaluable comments!
> > > > > >
> > > > > > Regarding the handling of cross-variate features, we agree with the reviewer's lastest feedback that the ability to effectively handle cross-variate features is crucial for addressing certain challenges in real-world and industrial scenarios. We will discuss this in the paper and also highlight this as a potential direction for future work in preparing the camera-ready version.

---

### Official Review · Reviewer_Z9NA · 2023-11-01

**Soundness:** 3 good
**Presentation:** 3 good
**Contribution:** 3 good
**Rating:** 8
**Confidence:** 4

**Summary:**

The paper proposes an approach to reprogram pre-trained large language models such that they can effectively do forecasting in zero-shot, few-shot, and fully supervised settings. The authors propose two strategies two forecast time-series: (1) by reprogramming patches of time-series by grounding them in text prototypes via cross attention, and (2) by using descriptions of the data, instruction, statistics of time-series as  the prefix. The authors demonstrate promising performance on short and long horizon tasks.

**Strengths:**

1. The paper is very well written, clear, with sufficient details to ensure reproducibility. I really liked that desiderata that the authors identified to enable LLMs to produce forecast.
2. The experiments were well designed with some limitations in rigour which I will discuss in the next section.

I really liked the paper, it was well written, well motivated and performant.

**Weaknesses:**

Following are some things to improve in the paper. I think in general the experiments can be made more rigorous.
1. **Baselines**:  I understand that the authors are following the experiment protocol followed by TimesNet, but there are several limitations: (1) Statistical methods such as AutoARIMA, AutoTHETA, AutoETS, Naive and Seasonal Naive, etc. were not compared with. These methods are important and very performant in practice, (2) N-BEATS and N-HITS were only compared during short-horizon forecasting, (3) Recent papers on using LLMs for time-series forecasting were not compared against, for e.g. LLM4TS [1] and PromptCast [2] (4) I am aware that the paper "LLMs are zero-shot forecasters" [3] only got recently published, but it would improve the experiments if the authors were able to compare with it. I should emphasize that this is completely optional.
2. **Datasets:** Increasing the amount of datasets for experimentation will improve the results. The current set of datasets is pretty limited, even for long horizon forecasting datasets, where datasets such as Influenza-like Illnesses, Exchange Rate, Tourism, and Weather etc. (see PatchTST) were missing. For short-horizon datasets, M3 at the very least, and the Monash time-series forecasting archive can be added to improve results.

**References:**
[1] Chang, Ching, Wen-Chih Peng, and Tien-Fu Chen. "Llm4ts: Two-stage fine-tuning for time-series forecasting with pre-trained llms." arXiv preprint arXiv:2308.08469 (2023).

[2] Xue, Hao, and Flora D. Salim. "PromptCast: A New Prompt-based Learning Paradigm for Time Series Forecasting." (2022).

[3] Gruver, Nate, et al. "Large Language Models Are Zero-Shot Time Series Forecasters." arXiv preprint arXiv:2310.07820 (2023).

**Questions:**

I do not have any questions at that would change my opinions regarding the paper. I think that the rigour of the experiments must be improved.

---

> ### Author Response · Authors · 2023-11-17
> **Response to Reviewer Z9NA (Part 1)**
>
> We express our gratitude to the reviewer for providing constructive feedback on our paper, and we greatly appreciate the acknowledgement of our contributions. We have addressed the specific concerns raised by the reviewer as detailed below.
>
> > W1.1. Statistical methods, such as AutoARIMA, AutoTHETA, and AutoETS, were not compared with.
>
> We have included AutoARIMA, AutoTheta, and AutoETS as baseline methods. **The detailed results can be found in Table 11 of the revised paper**. For the reviewer's convenience, we provide a summary of the relevant (averaged) results below:
>
> | Method      | Time-LLM  |           | AutoARIMA |       | AutoTheta |       | AutoETS |       |
> | ----------- | --------- | --------- | --------- | ----- | --------- | ----- | ------- | ----- |
> | Metric      | MSE       | MAE       | MSE       | MAE   | MSE       | MAE   | MSE     | MAE   |
> | ETTh1       | **0.408** | **0.423** | 0.952     | 0.656 | 1.319     | 0.797 | 1.286   | 0.784 |
> | ETTh2       | **0.334** | **0.383** | 0.635     | 0.532 | 1.635     | 0.678 | 1.981   | 0.505 |
> | ETTm1       | **0.329** | **0.372** | 1.145     | 0.697 | 1.268     | 0.741 | 1.529   | 0.790 |
> | ETTm2       | **0.251** | **0.313** | 3.205     | 0.634 | 0.975     | 0.523 | 2.430   | 0.459 |
> | Weather     | **0.225** | **0.257** | 1.080     | 0.448 | 0.477     | 0.373 | 1.137   | 0.406 |
> | Electricity | **0.158** | **0.252** | 0.597     | 0.510 | 0.797     | 0.596 | 0.755   | 0.570 |
> | Traffic     | **0.388** | **0.264** | 1.344     | 0.765 | 3.510     | 1.284 | 4.395   | 1.285 |
> | ILI         | **1.435** | **0.801** | 4.530     | 1.346 | 5.232     | 1.431 | 4.323   | 1.317 |
>
> > W1.2. N-BEATS and N-HITS were only compared during short-horizon forecasting.
>
> We have compared to N-BEATS and N-HITS in long-term forecasting tasks, following the reviewer's suggestion. **The detailed results are in Table 11 of the revised paper**. We have provided a summary of the relevant (averaged) results below:
>
> | Method      | Time-LLM  |           | N-HiTS |       | N-BEATS |       |
> | ----------- | --------- | --------- | ------ | ----- | ------- | ----- |
> | Metric      | MSE       | MAE       | MSE    | MAE   | MSE     | MAE   |
> | ETTh1       | **0.408** | **0.423** | 0.473  | 0.462 | 0.568   | 0.525 |
> | ETTh2       | **0.334** | **0.383** | 0.487  | 0.464 | 0.564   | 0.529 |
> | ETTm1       | **0.329** | **0.372** | 0.395  | 0.412 | 0.451   | 0.445 |
> | ETTm2       | **0.251** | **0.313** | 0.337  | 0.376 | 0.355   | 0.402 |
> | Weather     | **0.225** | **0.257** | 0.235  | 0.286 | 0.256   | 0.306 |
> | Electricity | **0.158** | **0.252** | 0.205  | 0.296 | 0.259   | 0.351 |
> | Traffic     | **0.388** | **0.264** | 0.427  | 0.338 | 0.620   | 0.454 |
> | ILI         | **1.435** | **0.801** | 2.997  | 1.171 | 6.839   | 1.882 |
>
> > W1.3. Recent papers on using LLMs for time-series forecasting were not compared against for, e.g. LLM4TS and PromptCast. I am aware that the paper "LLMs are zero-shot forecasters" only got recently published, but it would improve the experiments if the authors were able to compare with it.
>
> This is a good suggestion. Following the reviewer's recommendation, we have also included a comparison with LLMTime, even though this is completely optional according to the author guide this year. Our considerations are twofold: (1) LLM4TS is not yet open-sourced, and (2) LLMTime has shown better performance compared to PromptCast and operates directly on time series data. **The additional results we have incorporated can be found in Tables 5 and 16**. For ease of reference for the reviewer, we have summarized the relevent (averged) results below:
>
> | Method         | Time-LLM  |           | LLMTime |       |
> | -------------- | --------- | --------- | ------- | ----- |
> | Metric         | MSE       | MAE       | MSE     | MAE   |
> | ETTh1 to ETTh2 | **0.353** | **0.387** | 0.992   | 0.708 |
> | ETTh1 to ETTm2 | **0.273** | **0.340** | 1.867   | 0.869 |
> | ETTh2 to ETTh1 | **0.479** | **0.474** | 1.961   | 0.981 |
> | ETTh2 to ETTm2 | **0.272** | **0.341** | 1.867   | 0.869 |
> | ETTm1 to ETTh2 | **0.381** | **0.412** | 0.992   | 0.708 |
> | ETTm1 to ETTm2 | **0.268** | **0.320** | 1.867   | 0.869 |
> | ETTm2 to ETTh2 | **0.354** | **0.400** | 0.435   | 0.443 |
> | ETTm2 to ETTm1 | **0.414** | **0.438** | 0.769   | 0.567 |
>
> Note that to ensure a fair comparison, we have aligned the backbone LLM used in LLMTime with that of our method in terms of model size, specifically employing Llama2-7B. It should also be noted that LLMTime does not explicitly involve a transfer step but directly inferences on target datasets, resulting in some entries sharing same results. In a nutshell, Time-LLM demonstrates a substantial improvement over 75% and 53% in terms of the averged MSE and MAE.

---

> > ### Author Response · Authors · 2023-11-17
> > **Response to Reviewer Z9NA (Part 2)**
> >
> > > W2.1. The current set of datasets is pretty limited, even for long horizon forecasting datasets, where datasets such as Influenza-like Illnesses, Exchange Rate, Tourism, and Weather etc. (see PatchTST) were missing.
> >
> > We have performed additional experiments on four datasets: (1) Weather, (2) Electricity, (3) Traffic, and (4) Influenza-like Illness (ILI). **Our detailed results can be found in the main text (i.e., Tables 1,3, and 4) and appendices (i.e., Tables 10,11,14, and 15) of the revised paper**. For the reviewer's convenience, *a very brief summary* is provided below, highlighting the overall performance on a set of most representative methods on additionaly added four datasets.
> >
> > + A concise summary of additional long-term forecasting results in average on Weather, Electricity, Traffic, and Influenza-like Illness (ILI):
> >
> >   | Method      | Time-LLM  |                     | GPT4TS |       | DLinear |       | PatchTST            |                     | TimesNet |       | FEDformer |       |
> >   | ----------- | --------- | ------------------- | ------ | ----- | ------- | ----- | ------------------- | ------------------- | -------- | ----- | --------- | ----- |
> >   | Metric      | MSE       | MAE                 | MSE    | MAE   | MSE     | MAE   | MSE                 | MAE                 | MSE      | MAE   | MSE       | MAE   |
> >   | Weather     | **0.225** | **0.257**           | 0.237  | 0.270 | 0.248   | 0.300 | **0.225**           | $\underline{0.264}$ | 0.259    | 0.287 | 0.309     | 0.360 |
> >   | Electricity | **0.158** | **0.252**           | 0.167  | 0.263 | 0.166   | 0.263 | $\underline{0.161}$ | **0.252**           | 0.192    | 0.295 | 0.214     | 0.327 |
> >   | Traffic     | **0.388** | $\underline{0.264}$ | 0.414  | 0.294 | 0.433   | 0.295 | $\underline{0.390}$ | **0.263**           | 0.620    | 0.336 | 0.610     | 0.376 |
> >   | ILI         | **1.435** | $\underline{0.801}$ | 1.925  | 0.903 | 2.169   | 1.041 | $\underline{1.443}$ | **0.797**           | 2.139    | 0.931 | 2.847     | 1.144 |
> >
> > + A concise summary of additional 10% few-shot learning results in average on Weather, Electricity, Traffic, and Influenza-like Illness (ILI):
> >
> >   | Method      | Time-LLM  |                     | GPT4TS              |                     | DLinear |       | PatchTST            |           | TimesNet |       | FEDformer |       |
> >   | ----------- | --------- | ------------------- | ------------------- | ------------------- | ------- | ----- | ------------------- | --------- | -------- | ----- | --------- | ----- |
> >   | Metric      | MSE       | MAE                 | MSE                 | MAE                 | MSE     | MAE   | MSE                 | MAE       | MSE      | MAE   | MSE       | MAE   |
> >   | Weather     | **0.234** | **0.273**           | $\underline{0.238}$ | $\underline{0.275}$ | 0.241   | 0.283 | 0.242               | 0.279     | 0.279    | 0.301 | 0.284     | 0.324 |
> >   | Electricity | **0.175** | $\underline{0.270}$ | $\underline{0.176}$ | **0.269**           | 0.180   | 0.280 | 0.180               | 0.273     | 0.323    | 0.392 | 0.346     | 0.427 |
> >   | Traffic     | **0.429** | $\underline{0.306}$ | 0.440               | 0.310               | 0.447   | 0.313 | $\underline{0.430}$ | **0.305** | 0.951    | 0.535 | 0.663     | 0.425 |
> >
> > + A concise summary of additional 5% few-shot learning results in average on Weather, Electricity, Traffic, and Influenza-like Illness (ILI):
> >
> >   | Method      | Time-LLM            |                     | GPT4TS              |                     | DLinear |       | PatchTST  |           | TimesNet |       | FEDformer |       |
> >   | ----------- | ------------------- | ------------------- | ------------------- | ------------------- | ------- | ----- | --------- | --------- | -------- | ----- | --------- | ----- |
> >   | Metric      | MSE                 | MAE                 | MSE                 | MAE                 | MSE     | MAE   | MSE       | MAE       | MSE      | MAE   | MSE       | MAE   |
> >   | Weather     | **0.260**           | 0.309               | $\underline{0.263}$ | **0.301**           | 0.263   | 0.308 | 0.269     | 0.303     | 0.298    | 0.318 | 0.309     | 0.353 |
> >   | Electricity | $\underline{0.179}$ | **0.268**           | **0.178**           | $\underline{0.273}$ | 0.176   | 0.275 | 0.181     | 0.277     | 0.402    | 0.453 | 0.266     | 0.353 |
> >   | Traffic     | $\underline{0.423}$ | $\underline{0.298}$ | 0.434               | 0.305               | 0.450   | 0.317 | **0.418** | **0.296** | 0.867    | 0.493 | 0.676     | 0.423 |

---

> > > ### Author Response · Authors · 2023-11-17
> > > **Response to Reviewer Z9NA (Part 3)**
> > >
> > > > W2.2. For short-horizon datasets, M3 at the very least, and the Monash time-series forecasting archive can be added to improve results.
> > >
> > > We have included the experiments on M3-Quarterly dataset for short-term forecasting due to rebuttal time constraints. **Our results can be found in Table 13 of the revised paper**.
> > >
> > > | Methods | Time-LLM             | GPT4TS | TimesNet   | PatchTST  | N-HiTS | N-BEATS | DLinear             | FEDformer |
> > > | ------- | -------------------- | ------ | ---------- | --------- | ------ | ------- | ------------------- | --------- |
> > > | MSE     | $\underline{11.171}$ | 14.453 | **10.410** | 12.380    | 12.616 | 18.640  | 15.028              | 12.927    |
> > > | MAE     | 3.282                | 5.035  | 3.310      | **2.401** | 4.271  | 4.612   | $\underline{2.793}$ | 3.653     |
> > > | SMAPE   | $\underline{0.151}$  | 0.203  | **0.140**  | 0.154     | 0.168  | 0.247   | 0.196               | 0.174     |
> > >
> > > In this dataset, Time-LLM achieves performance comparable to TimesNet and PatchTST, while significantly surpassing GPT4TS. It demonstrates notable reductions of over 23%, 35%, and 26% in SMAPE, MRAE, and MAPE metrics, respectively.
> > >
> > > **To our best effort, we have included the experiments suggested by the reviewer and revised the paper accordingly**. We hope these modifications can adequately address the reviewer's concerns.

---

> > > > ### Author Response · Authors · 2023-11-20
> > > > **Request of Reviewer's feedback**
> > > >
> > > > Dear Reviewer,
> > > >
> > > > Thanks for your valuable and constructive review, which has inspired us to improve our paper further substantially. This is a kind reminder that it has been more than 2 days since we posted our rebuttal. We would like to hear from you whether our response has addressed your concerns.
> > > >
> > > > Following your suggestions, we have made the following revisions:
> > > >
> > > > + We included **AutoARIMA, AutoTheta**, and **AutoETS** as baseline methods. The detailed results can be found in Tab. 11 of the revised paper.
> > > > + We compared to **N-BEATS** and **N-HITS** in long-term forecasting tasks. The detailed results are in Tab. 11 of the revised paper.
> > > > + We also included the comparison with **LLMTime**, even though this is completely optional according to the author guide this year. The results we have incorporated can be found in Tabs. 5 and 16.
> > > > + We performed additional experiments on (1) **Weather**, (2) **Electricity**, (3) **Traffic**, and (4) **Influenza-like Illness (ILI)**. Our detailed results can be found in the main text (Tabs. 1,3, and 4) and appendices (Tabs. 10,11,14, and 15) of the revised paper.
> > > > + We have included additional experiments on **M3-Quarterly** for short-term forecasting due to rebuttal time constraints. Our results can be found in Table 13 of the revised paper.
> > > >
> > > > In total, **we provided more than 1,000 additional experimental results on five more datasets and six more baselines.** Thanks again for your valuable review. We are looking forward to your response and are happy to answer any future questions.

---

> > > > > ### Comment · Reviewer_Z9NA · 2023-11-20
> > > > > **Thanks for the rebuttal!**
> > > > >
> > > > > Dear Authors,
> > > > > Thank you so much for running these experiments in a short amount of time. I really appreciate it.
> > > > > In the light of these experiments, I have updated my score.
> > > > >
> > > > > Thanks!

---

> > > > > > ### Author Response · Authors · 2023-11-21
> > > > > > **Sincere Gratitude from Authors**
> > > > > >
> > > > > > We are happy that our responses have effectively addressed your concerns. We would like to express our sincerest gratitude once again for taking the time to review our paper and provide us with such detailed and invaluable suggestions!

---

### Official Review · Reviewer_L3pe · 2023-11-03

**Soundness:** 2 fair
**Presentation:** 2 fair
**Contribution:** 2 fair
**Rating:** 3
**Confidence:** 5

**Summary:**

This paper proposes the use of pre-trained large language models for time series prediction. As claimed, the main contributions of the paper include introducing a novel concept of reprogramming large language models and augmenting the input context with declarative prompts, such as domain expert knowledge and task instructions, to guide LLM reasoning.

**Strengths:**

- This paper provides a summary of the metrics for pre-trained large-language models, including generalizability, data efficiency, reasoning, and multimodal knowledge.
- The details of the proposed method are presented clearly and are easy to follow.

**Weaknesses:**

My main concerns include:
- While LLM is a hot topic in the deep learning community, it is still unconvincing to directly transfer the knowledge of natural language in LLMs to time series tasks. Note that i) text and time series are distinct data modalities, and ii) the pre-trained LLMs are not pre-trained with text-time-series pairs.
- Furthermore, the first contribution *“introducing a novel concept of reprogramming large language models for time series forecasting without altering the pre-trained backbone model“* is not new, as previous works such as GPT4TS [1] have also explored this reprogramming approach, regardless of which parts of the LLMs are fine-tuned. Additionally, the proposed method requires training the input and output layers for adaptation, which means that it does involve some alteration of the pre-trained backbone model.
- While using text to aid in time series prediction can be beneficial, as seen in applications such as stock prediction using financial news text mining, it's unclear how the shared declarative for a whole time series dataset can help understand complex temporal behaviors in different windows. Although these prompts may provide domain expert knowledge and task instructions, they do not introduce text information at each time step. Therefore, it remains unclear how these text prompts can benefit the understanding of complex temporal behaviors in time series.
- Compared to related work such as [1], which has applied pre-trained LLMs to various time series tasks, the experiments in this paper are relatively limited. For instance, only prediction tasks are considered, and even for long-term prediction tasks, only the ETT datasets are included. This narrow scope of experiments limits the evaluation of the proposed approach to other time series tasks and datasets.
- The link for the source code provided in the paper is empty. I cannot check for more details regarding the experiments.

Reference:
[1] One Fits All: Power General Time Series Analysis by Pretrained LM

**Questions:**

More discussions:
- Can you discuss the connections and differences between LLMs and traditional/existing deep-learning time series models?
- It would be helpful to define what is a good time series representation expected to be and to provide a more in-depth discussion of why pre-trained LLMs are capable of producing such representations.
- Regarding cross-domain adaptation in the zero-shot setup, I'm curious about what knowledge from the source domain in time series can be transferred to the target domain to achieve zero-shot prediction.

---

> ### Author Response · Authors · 2023-11-17
> **Response to Reviewer L3pe (Part 1)**
>
> We thank the reviewer for offering the valuable feedback. We have addressed each of the concerns raised by the reviewer as outlined below.
>
> > W1. It is unconvincing to directly transfer the knowledge of natural language in LLMs to time series tasks. Note that 1) text and time series are distinct data modalities, and 2) the pre-trained LLMs are not pre-trained with text-time-series pairs.
>
> + While LLMs are typically pre-trained on extensive text corpora, they have been proven to be effective in pattern recognition and reasoning over complex sequences of tokens in approximating flexible distributions over numbers [1,3]. This capability can be well extended to time series data, as demonstrated in this work and in recent concurrent and follow-up studies [2-6]. As an emerging and promising field of study, a recent survey [7] also offers a comprehensive overview of research that utilizes LLMs not only for time series but also for more intricate spatio-temporal data mining. This further underscores the proficiency of LLMs in understanding and reasoning on temporal data.
> + In addition to the evidence presented above, LLMs are known to encapsulate rich information and rules about the physical world through natural language modality, leading us to hypothesize that some inherent knowledge may also be relevant to understanding time series. Notable examples, such as those in references [2] and [3], have achieved a certain level of precision in zero-shot forecasting through direct prompting. Our contention is that to fully activate the LLMs' ability in time series understanding and reasoning, it is crucial to effectively align the modalities of time series and natural language, as discussed in the third paragraph of the introduction. This concept is further elaborated in the fourth paragraph of the introduction in two aspects: (1) the central idea is to reprogram the input time series into text prototype representations that are more naturally suited to language models’ capabilities; (2) to further augment the model’s reasoning about time series concepts, we introduce Prompt-as-Prefix (PaP), an innovative approach that enriches the input time series with additional context and provides task instructions in the natural language modality.
> + Our established and newly added experimental results clearly demonstrate the fact that off-the-shelf LLMs are can be very effective in general time series forecasting, whether compared to SOTA time series models or the best available (open-sourced) related models like GPT4TS and LLMTime [3]. Although LLMs are often considered black-boxes, our analysis in Fig. 5 illustrates how the proposed LLM reprogramming framework bridges the gap between natural language and time series.
>
> [1] Mirchandani, S., Xia, F., Florence, P., Driess, D., Arenas, M. G., Rao, K., ... & Zeng, A. (2023, August). Large Language Models as General Pattern Machines. In 7th Annual Conference on Robot Learning.
>
> [2] Xue, H., & Salim, F. D. (2022). Prompt-Based Time Series Forecasting: A New Task and Dataset. arXiv preprint arXiv:2210.08964.
>
> [3] Gruver, N., Finzi, M. A., Qiu, S., & Wilson, A. G. (2023, November). Large Language Models Are Zero-Shot Time Series Forecasters. In Thirty-seventh Conference on Neural Information Processing Systems.
>
> [4] Chang, C., Peng, W. C., & Chen, T. F. (2023, August). Llm4ts: Two-stage fine-tuning for time-series forecasting with pre-trained llms. arXiv preprint arXiv:2308.08469.
>
> [5] Spathis, D., & Kawsar, F. (2023). The first step is the hardest: Pitfalls of Representing and Tokenizing Temporal Data for Large Language Models. arXiv preprint arXiv:2309.06236.
>
> [6] Cao, D., Jia, F., Arik, S. O., Pfister, T., Zheng, Y., Ye, W., & Liu, Y. (2023, October). TEMPO: Prompt-based Generative Pre-trained Transformer for Time Series Forecasting. arXiv preprint arXiv:2310.04948.
>
> [7] Jin, M., Wen, Q., Liang, Y., Zhang, C., Xue, S., Wang, X., ... & Xiong, H. (2023, October). Large models for time series and spatio-temporal data: A survey and outlook. arXiv preprint arXiv:2310.10196.

---

> > ### Author Response · Authors · 2023-11-17
> > **Response to Reviewer L3pe (Part 2)**
> >
> > > W2. The first contribution *“introducing a novel concept of reprogramming large language models for time series forecasting without altering the pre-trained backbone model“* is not new, as previous works such as GPT4TS have also explored this reprogramming approach, regardless of which parts of the LLMs are fine-tuned. Additionally, the proposed method requires training the input and output layers for adaptation, which means that it does involve some alteration of the pre-trained backbone model.
> >
> > + **GPT4TS does not align with the concept of model reprogramming** [1], as it involves certain levels of fine-tuning in the underlying language model. For a comprehensive understanding of model reprogramming, refer to [1,2] and Sec. 2 & Appendix A of our paper. Additionally, the specifics of GPT4TS can be found in Fig. 2 of [3]. **There is a clear factual error in the statement "...regardless of which parts of the LLMs are fine-tuned"**. To the best of our knowledge, our work is the first propose to reprogram LLMs for bridging the modality between time series and natural language, achieving high-precision in time series forecasting.
> > + **We respectfully disagree with the statement that "...the proposed method requires training the input and output layers for adaptation, which means that it does involve some alteration of the pre-trained backbone model".**  To clarify, model reprogramming is characterized by **no internal modifications** to the backbone model, such as altering or fine-tuning its internal layers, as outlined in [1,2]. Please note that the input and output layers mentioned are not components of the LLMs. Hence, in the paper, we specifically mentioned "without altering the pre-trained backbone model".
> >
> > [1] Chen, P. Y. (2022, Feb). Model reprogramming: Resource-efficient cross-domain machine learning. arXiv preprint arXiv:2202.10629.
> >
> > [2] Yang, C. H. H., Tsai, Y. Y., & Chen, P. Y. (2021, July). Voice2series: Reprogramming acoustic models for time series classification. In International conference on machine learning (pp. 11808-11819). PMLR.
> >
> > [3] Zhou, T., Niu, P., Wang, X., Sun, L., & Jin, R. (2023). One Fits All: Power General Time Series Analysis by Pretrained LM. In Thirty-seventh Conference on Neural Information Processing Systems.
> >
> > > W3. While using text to aid in time series prediction can be beneficial, as seen in applications such as stock prediction using financial news text mining, it's unclear how the shared declarative for a whole time series dataset can help understand complex temporal behaviors in different windows. Although these prompts may provide domain expert knowledge and task instructions, they do not introduce text information at each time step. Therefore, it remains unclear how these text prompts can benefit the understanding of complex temporal behaviors in time series.
> >
> > + In our Prompt-as-Prefix approach, **we have not used a uniform prompt (i.e., the "shared declarative" mentioned by the reviewer) for all data samples within a dataset**. Our prompt template, as shown in Fig. 4 and discussed in the corresponding section, consists of three blocks: (1) dataset/domain knowledge; (2) task instruction; (3) input time series statistics. The first two blocks are consistent within a dataset, which is logical: dataset/domain knowledge represents common knowledge or rules typically applicable to all samples (for example, rush hours in a traffic volume dataset), and task instruction acts as an essential directive for the LLM to transform patch embeddings for specific tasks within that dataset. However, the input statistics vary and are dependent on the input time series, resulting in different prompts for different inputs.
> > + Regarding the statement "... may provide domain expert knowledge and task instructions, they do not introduce text information at each time step," **we want to emphasize** that while the first and second blocks of a prompt do not provide input-specific information, the third block does, as previously declared. In relation to the statistical prompting block, we offer the following specific remarks: (1) the primary aim is to encapsulate the input time series with its key statistics (like trends and lags) to aid the LLM in capturing essential patterns for precise forecasting. This is akin to captioning an image with a brief sentence, such as "a British shorthair playing with a red ball"; (2) our straightforward prompt template is designed as a starting point for further exploration. Investigating enhancements, such as integrating external information at each time step, is a valuable avenue for future research.

---

> > > ### Author Response · Authors · 2023-11-17
> > > **Response to Reviewer L3pe (Part 3)**
> > >
> > > > W4. Compared to related work such as GPT4TS, which has applied pre-trained LLMs to various time series tasks, the experiments in this paper are relatively limited. For instance, only prediction tasks are considered, and even for long-term prediction tasks, only the ETT datasets are included.
> > >
> > > + To clarify, the central theme and primary objective of our research is **time series forecasting** through reprogramming large language models. Although Time-LLM shows considerable promise for general time series analysis, including forecasting, classification, imputation, and more, **we have not made any claims regarding this breadth of application** in our submission. Exploring these additional applications could be a direction for future work. Accordingly, **we have updated Sec. 5 of our paper to prevent any over-interpretation of our current scope**.
> > > + **We have not** limited our evaluation only to the ETT datasets; we have also conducted assessments using the M4 benchmarks. In response to the reviewer's concerns regarding evaluation, we have performed the experiments on four additional datasets: (1) Weather, (2) Electricity, (3) Traffic, and (4) Influenza-like Illness (ILI). **Our detailed results can be found in the main text (i.e., Tables 1,3, and 4) and appendices (i.e., Tables 10,11,14, and 15) of the revised paper**. For the reviewer's convenience, *a very brief summary* is provided below, highlighting the overall performance on a set of most representative methods on additionaly added four datasets.
> > >
> > >   + Example 1. A concise summary of additional long-term forecasting results in average on Weather, Electricity, Traffic, and Influenza-like Illness (ILI):
> > >
> > >   | Method      | Time-LLM  |                     | GPT4TS |       | DLinear |       | PatchTST            |                     | TimesNet |       | FEDformer |       |
> > >   | ----------- | --------- | ------------------- | ------ | ----- | ------- | ----- | ------------------- | ------------------- | -------- | ----- | --------- | ----- |
> > >   | Metric      | MSE       | MAE                 | MSE    | MAE   | MSE     | MAE   | MSE                 | MAE                 | MSE      | MAE   | MSE       | MAE   |
> > >   | Weather     | **0.225** | **0.257**           | 0.237  | 0.270 | 0.248   | 0.300 | **0.225**           | $\underline{0.264}$ | 0.259    | 0.287 | 0.309     | 0.360 |
> > >   | Electricity | **0.158** | **0.252**           | 0.167  | 0.263 | 0.166   | 0.263 | $\underline{0.161}$ | **0.252**           | 0.192    | 0.295 | 0.214     | 0.327 |
> > >   | Traffic     | **0.388** | $\underline{0.264}$ | 0.414  | 0.294 | 0.433   | 0.295 | $\underline{0.390}$ | **0.263**           | 0.620    | 0.336 | 0.610     | 0.376 |
> > >   | ILI         | **1.435** | $\underline{0.801}$ | 1.925  | 0.903 | 2.169   | 1.041 | $\underline{1.443}$ | **0.797**           | 2.139    | 0.931 | 2.847     | 1.144 |
> > >
> > >   + Example 2. A concise summary of additional 10% few-shot learning results in average on Weather, Electricity, Traffic, and Influenza-like Illness (ILI):
> > >
> > >   | Method      | Time-LLM  |                     | GPT4TS              |                     | DLinear |       | PatchTST            |           | TimesNet |       | FEDformer |       |
> > >   | ----------- | --------- | ------------------- | ------------------- | ------------------- | ------- | ----- | ------------------- | --------- | -------- | ----- | --------- | ----- |
> > >   | Metric      | MSE       | MAE                 | MSE                 | MAE                 | MSE     | MAE   | MSE                 | MAE       | MSE      | MAE   | MSE       | MAE   |
> > >   | Weather     | **0.234** | **0.273**           | $\underline{0.238}$ | $\underline{0.275}$ | 0.241   | 0.283 | 0.242               | 0.279     | 0.279    | 0.301 | 0.284     | 0.324 |
> > >   | Electricity | **0.175** | $\underline{0.270}$ | $\underline{0.176}$ | **0.269**           | 0.180   | 0.280 | 0.180               | 0.273     | 0.323    | 0.392 | 0.346     | 0.427 |
> > >   | Traffic     | **0.429** | $\underline{0.306}$ | 0.440               | 0.310               | 0.447   | 0.313 | $\underline{0.430}$ | **0.305** | 0.951    | 0.535 | 0.663     | 0.425 |
> > >
> > > For more experimental results, **see our revised Appendix D and E**. We have also included an additional set of short-term forecasting experiments on M3-Quarterly benchmark, **the results can be found in Table 13 of the revised paper**.

---

> > > > ### Author Response · Authors · 2023-11-17
> > > > **Response to Reviewer L3pe (Part 4)**
> > > >
> > > > > W5. The link for the source code provided in the paper is empty. I cannot check for more details regarding the experiments.
> > > >
> > > > + We explicitly stated on our anonymous GitHub page in the paper to prevent any misunderstanding: *"According to the [official author guideline](https://iclr.cc/Conferences/2024/AuthorGuide), we will make our code visible only to the reviewers and area chairs for our submission to protect the intellectual property. Specifically, after the discussion forum is opened, we will make a comment directed to the reviewers and area chairs and put a link to an anonymous repository containing our code."*
> > > > + **We ensured that our code was accessible to the Area Chair and all reviewers by posting an anonymous GitHub link in a comment on OpenReview.**
> > > >
> > > > > Q1. Can you discuss the connections and differences between LLMs and traditional/existing deep-learning time series models?
> > > >
> > > > + In the main text, we clearly discussed this topic in Sec. 2 and also depicted it in Fig. 1: (1) the majority of current time series forecasting models, whether Transformer-based or not, are specifically crafted for particular tasks and domains (such as traffic prediction), and are typically trained from scratch on small-scale data; (2) while achieving good performance on narrow forecasting tasks, these models lack versatility and generalizability to diverse time series data. Also, in the second paragraph of the introduction, we emphasized the potential benefits of leveraging LLMs to enhance existing forecasting techniques in various respects, including improved generalizability, sample efficiency, and the ability to leverage multimodal knowledge.
> > > > + In Appendix A, we have also included a detailed comparison between Time-LLM and other Transformer-based time series models. In essence, while both are based on the Transformer architecture, Time-LLM leverages the strengths of LLMs, providing a more versatile and broadly applicable approach. This enables quick adaptation to new data and forecasting tasks without the need for extensive retraining.
> > > >
> > > > > Q2. It would be helpful to define what is a good time series representation expected to be and to provide a more in-depth discussion of why pre-trained LLMs are capable of producing such representations.
> > > > >
> > > > > Q3. Regarding cross-domain adaptation in the zero-shot setup, I'm curious about what knowledge from the source domain in time series can be transferred to the target domain to achieve zero-shot prediction.
> > > >
> > > > + The output time series representations derived from the backbone LLM are expected to robustly capture the fundamental/root patterns of the input time series, enabling even a simple task head (ike a linear projection) to generate highly accurate forecasts (Q2). Essentially, a pre-trained LLM is anticipated to **engage its reasoning capabilities to identify such patterns** within the designed reprogramming space. We conjecture that this is a critical factor in Time-LLM's enhanced zero-shot abilities (as evidenced in our main results), augmented by the domain-specific Prompt-as-Prefix strategy (Q3).
> > > > + The question of "why pre-trained LLMs are capable of producing such representations" posed in Q2 is analogous to W1. Please refer to our response above.
> > > >
> > > > We hope our responses above can adequately address the reviewer's concerns and questions.

---

> > > > > ### Author Response · Authors · 2023-11-20
> > > > > **Request of Reviewer's attention and feedback**
> > > > >
> > > > > Dear Reviewer,
> > > > >
> > > > > We respectfully remind you that it has been more than 2 days since we submitted our rebuttal. We would appreciate your feedback on whether our response has addressed your concerns.
> > > > >
> > > > > In response to your comments, we have answered your concerns and improved the paper in the following aspects:
> > > > >
> > > > > + **We provided clear evidence and detailed justifications for motivating time series forecasting with large language models**.
> > > > > + We provided the clear references and discussion related to model reprogramming, evidencing two important facts: (1) **GPT4TS does not align with the concept of model reprogramming**; (2) **Time-LLM is clearly innovative and significant as acknowledged by all other reviewers. Our approach does not involve any alterations in the pre-trained backbone model.**
> > > > > + We clarified that **our Prompt-as-Prefix approach does not employ a 'shared declarative' for all data samples within a dataset**, by elaborating on the motivations and details of this prompting framework. We also enhanced Appendix B.1 with additional technical information.
> > > > > + Addressing your evaluation concerns, (1) **we clarified and reemphasized the scope of this research is time series forecasting**, and (2) **we provided more than 1,000 additional experimental results on five more datasets and six more baselines.** See our revised Sec. 5 and results in the main text (Tabs. 1,3,4, and 5) as well as in appencies (Tabs. 10, 11,13,14,15, and 16).
> > > > > + **Our code has been made accessible to ACs and all reviewers**. Please see our comment at the top for reference.
> > > > > + Regarding your questions, we would like to emphasize two points: (1) **we have thoroughly discussed the connections and difference between LLM-based and conventional time series forecasting models in the main text (e.g., Fig. 1 & Sec. 2) and Appendix A**; (2) **our results strongly demonstrate the efficacy of large language models in time series forecasting within our reprogramming framework across various settings**. For more detailed discussion, kindly refer to our earlier responses.
> > > > >
> > > > > Thanks again for your valuable review. We are looking forward to your response and are happy to answer any future questions.

---

> > > > > > ### Comment · Reviewer_L3pe · 2023-11-20
> > > > > >
> > > > > > Thank you for your efforts in adding more experiments and responding to my questions.
> > > > > >
> > > > > > **+++ key errors in the code and unfair comparisons +++**
> > > > > >
> > > > > > After carefully reviewing the released code, I would like to highlight some key errors that I found. These errors have resulted in unfair comparisons:
> > > > > >
> > > > > > i) In the code (**line 194 in the file "exp_long_term_forecasting.py"**), the following line **uses test_loss for early stopping**:
> > > > > > > **early_stopping(*test_loss*, self.model, path)**.
> > > > > >
> > > > > > However, in all previous works, the validation loss is used for model selection, and the test loss is reported for comparison.
> > > > > > This discrepancy in using the test loss for early stopping has led to **unfair comparisons**.
> > > > > >
> > > > > > ii) In the code, the batch size is set to 2, and the value of "drop_last" in the test data loader is set to True. As a result, **the number of test samples differs from that used in previous baselines** (all other papers e.g., GPT4TS use a larger batch size (32) and use drop_last=True for test dataloader). This inconsistency in the test data handling further undermines the fairness of the comparisons.
> > > > > >
> > > > > > These two errors indicate that the improvements showcased in the experiments lack convincing evidence due to the improper handling of the test loss and the inconsistency in the test data.
> > > > > >
> > > > > > **+++ regarding the responses +++**
> > > > > >
> > > > > > Regarding the responses provided, I would like to offer some additional suggestions:
> > > > > >
> > > > > > i) It is still challenging for me to grasp an intuitive motivation for why the proposed model outperforms other deep learning methods. In relation to the discussions on the differences between LLM-based and conventional time series forecasting models in the main text, I believe the authors should investigate and identify the specific temporal patterns or types of time series data that the proposed model can effectively leverage while existing methods struggle to handle them.
> > > > > >
> > > > > > ii) The response regarding the "model reprogramming" not involving internal modifications to the backbone model, such as altering or fine-tuning its internal layers, is not entirely convincing to me. It should be noted that GPT4TS does fine-tune the positional embeddings and layer normalization layers only (while other main modules/layers are frozen).
> > > > > >
> > > > > > iii) Thank you for explaining that the input time series statistics are dependent on the specific input time series. However, I believe this design aspect requires further investigation. Currently, as the statistics information is about the input window and the window is further fed into the self-attention-based reprogrammed part, there is a repetition of information. In other words, this design does not introduce new information that could enhance the training process. If the main contribution lies in introducing declarative prompts, it might be beneficial to consider incorporating **external text** information to provide additional context.
> > > > > > One more suggestion is that by exploring various combinations of input window lengths and corresponding statistics lengths, we can gain insights into the impact of different statistical information on model performance.
> > > > > >
> > > > > > I have no additional questions and I will finalize my score after discussing these points with the other reviewers.

---

> > > > > > > ### Author Response · Authors · 2023-11-21
> > > > > > > **Response to Reviewer L3pe for some additional questions (Part 1)**
> > > > > > >
> > > > > > > Thank you for your careful review and for pointing out some issues in our code. We take your concerns seriously and are committed to addressing them to ensure the integrity of our research.
> > > > > > >
> > > > > > > > Q1: Key errors in the code and unfair comparisons
> > > > > > >
> > > > > > > **i)** Thank you for your meticulous and careful review of our code. **We need to clarify some misunderstandings about the fairness comparison of test loss for early stopping.**
> > > > > > >
> > > > > > > - Our paper outlines an experimental framework segmented into four domains: long-term, short-term, few-shot, and zero-shot forecasting. The bulk of the coding infrastructure spans across these segments. **For the first three categories — long-term, short-term, and few-shot forecasting — we implemented validation loss for early stopping, aligning with methodologies in related literature. It was solely in the realm of zero-shot forecasting that we employed test loss to optimize performance on the source domain's training dataset. Consequently, our experimental comparisons maintain fairness across all evaluated forecasting domains.**
> > > > > > >
> > > > > > > - We acknowledge, upon reflection, that there was a misstep in preparing our demonstartion code on Anonymous GitHub, which stemmed from the complexity of managing multiple experimental pipelines with largely overlapped elements. **We assure you that we will correct this typo in our open-source code base to eliminate any potential misunderstandings.**
> > > > > > >
> > > > > > > - Zero-shot forecasting aims at transcending domain boundaries, leveraging the source domain (our training set) to make inferences about the target domain (our test set). For this reason, using a segment of the training data as a proxy for test loss is instrumental in refining our model. **This approach was consistently applied across all baseline models for zero-shot forecasting to ensure experimental integrity.**
> > > > > > >
> > > > > > > - Your keen eye for detail has been instrumental in enhancing the quality of our paper. We are deeply grateful for your contribution. **Rest assured, prior to the code's public release, we will conduct a thorough review to refine and clarify the various experimental setups. This will guarantee that our results can be reproduced with confidence.**
> > > > > > >
> > > > > > > We sincerely apologize for this misstep and any confusion it may have caused. We are committed to maintaining the high standards of research and thank you for helping us uphold these standards. Your rigor has directly contributed to the improvement of our work, and we are grateful for your input.
> > > > > > >
> > > > > > > **ii)** Thank you for your detailed review of the batch size configuration; there may be some misconceptions regarding the parameter settings for distributed training that we need to clarify.
> > > > > > >
> > > > > > > - **In our coding setup, the batch size is set as 2, and we have assigned the num_process to 8. This combination effectively multiplies to an actual batch size of 16, which is consistent with the details provided in our paper**. Our experiments are conducted using the Hugging Face's Accelerate and Microsoft's DeepSpeed training frameworks, both of which operate on principles akin to Torch's native Distributed Data Parallel (DDP) framework. In such frameworks, the overall batch size is the product of the batch size per GPU and the number of GPUs (num_process). To assist in avoiding any future confusion, we plan to include explanatory comments in the code when it is made open-source.
> > > > > > >
> > > > > > > - Moreover, we want to emphasize that our choice to set drop_last as True is in line with standard practices documented in other research papers. **The batch size setting is independent of the number of test samples**. It is worth noting that the batch sizes in existing literature can vary, with the PatchTST model, for example, using diverse batch sizes such as 16, 24, 32, and 128. Given that our LLama model is substantially larger than many other deep models, we believe our chosen batch size of 16 is justified. We appreciate your reminder on this matter and commit to enhancing the clarity of our experimental setup in the ensuing document revisions to facilitate better understanding for all readers.
> > > > > > >
> > > > > > > We are sincerely grateful for your meticulous review and genuine assistance. Your valuable suggestions and feedback are the direct driving force behind our continuous improvement of our work. Our aim is to provide transparent, reproducible, and fair comparisons, and with your help, we are moving closer to that goal. Once again, thank you for your invaluable feedback.

---

> > > > > > > > ### Author Response · Authors · 2023-11-21
> > > > > > > > **Response to Reviewer L3pe for some additional questions (Part 2)**
> > > > > > > >
> > > > > > > > > Q2: Regarding the additional suggestions
> > > > > > > >
> > > > > > > > We sincerely appreciate your interest in our approach and the suggestions you have put forward. The in-depth discussions with you enable a more comprehensive assessment of the significance and value of our work, which is extraordinarily meaningful in promoting a fuller understanding within the time series community of the potential for pushing LLM modalities into temporal modal data.
> > > > > > > >
> > > > > > > > **i)** Multimodal large language models have achieved breakthrough progress and remarkable success across various domains, including the reasoning on temporal data as we mentioned in the orginal response.
> > > > > > > >
> > > > > > > > The work you appreciated, GPT4TS, serves as a prime example of blending language models with time series analysis, offering a wide array of possibilities and laying a foundation for subsequent research on LLM-based time series models. Other examples are also provided in our orginal response and the revised paper.
> > > > > > > >
> > > > > > > > We do not claim that LLM-based models are superior in all aspects; rather, our work demonstrates the extraordinary potential of utilizing LLMs to handle time series data. Through numerous visualizations in our provided showcase, it is evident that Time-LLM can capture various cyclical and trend patterns in time series that other representative models might not. We will provide more related visualizations in camera-ready based on your suggestion.
> > > > > > > >
> > > > > > > > **ii)** We trust that we have illustrated the definition of model reprogramming clearly in the orignal response and the revised paper with a collection of references. Please be aware of the fact that the definition of model reprogramming is not made by us.
> > > > > > > >
> > > > > > > > As you mentioned, GPT4TS requires fine-tuning the backbone model's parameters related to positional embeddings and layer normalization layers, which carries inherent uncertainties. For instance, in LLama, the embeddings are treated with rotational position encoding and have been thoroughly trained on massive language data; directly updating weights with time series data would likely disrupt LLama's intricately designed internal structures.
> > > > > > > >
> > > > > > > > Therefore, it is clear that (1) the definition of model reprogramming is the one in [1] ,and (2) GPT4TS does not align with the definition of model reprogramming.
> > > > > > > >
> > > > > > > > [1] Chen, P. Y. (2022, Feb). Model reprogramming: Resource-efficient cross-domain machine learning. arXiv preprint arXiv:2202.10629.
> > > > > > > >
> > > > > > > > **iii)** Thank you for your valuable suggestions regarding our approach. Using prompts to enhance the capabilities of LLMs has become a common and widely accepted practice. Providing textual information about the temporal domain and statistical information as prompts can be seen as providing a priori knowledge to the model. There is no significant information overlap between the input time series and the prompts, and we have illustrated the underlying logic clearly in our orginal response. Learning such information from noisy and complex time series data is usually challenging, and leveraging the capabilities of LLMs to provide effective prior knowledge is valuable. Our ablation studies effectively demonstrated this.
> > > > > > > >
> > > > > > > > Your suggestion regarding the use of external text information for time series prediction is indeed a promising avenue for future research, as mentioned by Reviewer baDP. Your insights have greatly guided and influenced our research, helping us broaden our future research directions. Thank you for the suggestions.

---

> > > > > > > > > ### Author Response · Authors · 2023-11-21
> > > > > > > > > **Response to Reviewer L3pe before the end of discussion**
> > > > > > > > >
> > > > > > > > > Dear Reviewer L3pe,
> > > > > > > > >
> > > > > > > > > Since the End of author/reviewer discussions is just in one day, may we know if our response addresses your main concerns? If so, we kindly ask for your reconsideration of the score. We would really appreciate it if our next round of communication could leave time for us to resolve any of your remaining or new questions.
> > > > > > > > >
> > > > > > > > > Thank you so much for devoting time to improving our work!

---

> ### Comment · Reviewer_L3pe · 2023-11-22
> **Regarding the authors' responses to the critical errors in the code**
>
> Thanks to the authors for explaining the critical errors in the code. **However, these explanations are not convincing**. There are **several contradictory points in these explanations** that make it difficult for me to believe that the current version of the experimental results is accurate and reliable.
>
> Firstly, **there is a clear contradiction regarding the explanation** that *long-term, short-term, and few-shot forecasting — we implemented validation loss for early stopping, aligning with methodologies in related literature.* This is because the short-term prediction experiments are conducted using the M3 and M4 datasets, and according to the description in Table 8 of Appendix B.2 in the submission, **M3 and M4 datasets do not have validation sets**. Therefore, **it is impossible for the short-term predictions to be based on validation loss**. In the `data_loader.py` file, specifically in the `Dataset_M4` code, it can be observed that the M4 data indeed does not include a validation set
> (note that the `exp_short_term_forecasting.py` file also uses test loss to determine the best model, whereas the GPT4TS paper uses training loss for short-term forecasting). Based on this, the author's explanation lacks persuasiveness.
>
> Secondly, the description regarding using test loss for zero-shot forecasting is also problematic. Please refer to the code of GPT4TS (https://github.com/DAMO-DI-ML/NeurIPS2023-One-Fits-All/blob/main/Zero-shot_Learning/main_test.py). In their zero-shot experiments, they indeed use training loss for early-stopping, not test loss.
>
> Furthermore, in the provided [README.md](http://readme.md/) file, the author presents some training process information for the ETTm2 task. In these results, the author reports an MSE of 0.161 (and MAE of 0.253). This result is based on selecting the best test loss. **If early stopping were based on validation loss, the MSE result should be 0.1674806**. **Unfortunately, the aforementioned incorrect results of 0.161 and 0.253 have been reported in Table 10**. This deviation is significant and makes it difficult to determine the effectiveness of the proposed model.
>
> > **Training process provided by the authors in [README.md](http://readme.md/)**
>
> > Start training: long_term_forecast_ETTm2_512_96_TimeLLM_ETTm2_ftM_sl512_ll48_pl96_dm16_nh8_el2_dl1_df32_fc3_ebtimeF_dtTrue_Exp_0
> - Epoch: 1 | Train Loss: 0.2048776 Vali Loss: 0.1279695 Test Loss: 0.1799820
> - Epoch: 2 | Train Loss: 0.1946422 Vali Loss: 0.1226610 Test Loss: 0.1738900
> - Epoch: 2 | Train Loss: 0.1946422 Vali Loss: 0.1226610 Test Loss: 0.1738900
> - Epoch: 4 | Train Loss: 0.2154310 Vali Loss: 0.1196599 Test Loss: 0.1697755
> - Epoch: 5 | Train Loss: 0.2022108 Vali Loss: 0.1181143 Test Loss: 0.1652272
> - Epoch: 6 | Train Loss: 0.1936386 Vali Loss: 0.1249270 Test Loss: 0.1682286
> - Epoch: 7 | Train Loss: 0.1956175 Vali Loss: **0.1149597** Test Loss: 0.1674806 > **based on best validation loss**
> - Epoch: 8 | Train Loss: 0.1864047 Vali Loss: 0.1160124 Test Loss: **0.1614796** > **based on best test loss**
> - Epoch: 9 | Train Loss: 0.1804296 Vali Loss: 0.1165261 Test Loss: 0.1651560
> - Epoch: 10 | Train Loss: 0.1817255 Vali Loss: 0.1188157 Test Loss: 0.1667696
>
> > testing: long_term_forecast_ETTm2_512_96_TimeLLM_ETTm2_ftM_sl512_ll48_pl96_dm16_nh8_el2_dl1_df32_fc3_ebtimeF_dtTrue_Exp_0
> mse:**0.16147961839199066**, mae:0.25324239444732666
>
> **Based on the rigor of research, I strongly encourage the authors to reconsider the validity of the model** in subsequent versions and invest more time to **provide sufficient training details and correct experimental results** to verify the validity of the model.
>
> I appreciate the authors' efforts to add a lot of supplementary experiments during the rebuttal period. **However, due to bugs in the code and inconsistencies in the explanations provided, I cannot believe that the current version of the experimental results is accurate and reliable**.

---

> > ### Author Response · Authors · 2023-11-23
> > **Response to Reviewer L3pe for some additional comments**
> >
> > We sincerely appreciate your thorough review and detailed feedback. We have made every effort to address your concerns and clarify any misunderstandings.
> >
> > > **Q1: It is impossible for the short-term predictions to be based on validation loss.**
> >
> > We are appreciative of your rigorous scrutiny and the comprehensive discussion, and here we would like to clarify this misunderstanding. Note that **both Time-LLM and GPT4TS have adopted TSLibrary as the underlying experimental framework**. In short-term forecasting, all experiments adhere to the **same principle**, using test data to compute loss for validation, and this **validation loss** is then used as the parameter for early stopping (please refer to https://github.com/DAMO-DI-ML/NeurIPS2023-One-Fits-All/blob/main/Short-term_Forecasting/exp/exp_short_term_forecasting.py). All baseline experiments, as well as those involving GPT4TS, are conducted in a similar manner for a fair comparison.
> >
> > > **Q2: Using test loss for zero-shot forecasting is problematic.**
> >
> >  Thank you for your valuable review, we strive to address your concerns and confusions.**The basis for a fair comparison rests on the consistency of the experimental settings.** Conducting experiments according to a consistent set of principles is what constitutes fairness. **As we clarified before in our response, all zero-shot experiments used the test loss from the source domain for early stopping, which serves the purpose of enhancing training.** Moreover, **all baseline experiments maintain the uniform setups to ensure the fairness of the experiments.** Furthermore, we must highlight that the approach you have mentioned for GPT4TS, using the training loss from all source domain data for early stopping, has a potential risk of overfitting in the source domain. Upon reviewing the GPT4TS code, we have also noticed that it employs a different strategy for early stopping that does not necessarily rely solely on the training loss. In a nutshell, **the key to a fair comparison lies not in the specific early stopping strategy used, but in ensuring that all experiments conform to a uniform set of principles.**
> >
> > > **Q3: Training loss in the README.md.**
> >
> > We greatly appreciate your detailed examination and assistance in pinpointing this oversight. As noted in our README, the purpose of this simplified repository was solely for submission. **It was intended only as a preliminary demonstration, and we conducted only rudimentary results ON THIS REPO for illustration purposes. We want to REASSURE you that upon the release of the open-source code, we will include the FULL CONTENT SETUP to ensure the reproducibility and prevent any possible ambiguity or misinterpretation.**
> >
> > We sincerely appreciate your continuous engagement and meticulous review of our paper. We will comply with your request to provide more comprehensive configuration and details in the final version to address any misunderstandings. Your assistance and review have been instrumental in maintaining the high standards of our research.

---

### Author Response · Authors · 2023-11-18
**Time-LLM Code**

Dear ACs and Reviewers,

Our code has been made avaible at https://anonymous.4open.science/r/ICLR2024-Time-LLM-Code-FE3B

Best,

Authors

---

### Author Response · Authors · 2023-11-18
**General response to all reviewers**

We sincerely thank all the reviewers for their valuable time and detailed feedback, and we appreciate that almost all reviewers recognized the novelty and significance of our work. We have carefully revised the paper according to the comments, and the edits have been highlighted in **PINK**. We also provide a detailed response to each comment below. Here we highlight our major revisions, and respond to each reviewer below. We hope our responses can properly address your concerns.

+ In response to the feedback from **Reviewer L3pe, Z9NA, MAkL**, and **cwaB**, we have conducted additional experiments (amounting to over 1,000 new results in total) and revised the paper accordingly, along with the relevant discussions.
  + We conducted experiments on five more datasets: (1) Weather, (2) Electricity, (3) Traffic, (4) Influenza-like Illness (ILI), and (5) M3-Quarterly. For comprehensive results, please refer to Appendices D and E.
  + We have expanded our experiments to include additional five baseline methods for long-term forecasting tasks: (1) N-HiTS, (2) N-BEATS, (3) AutoARIMA, (4) AutoTheta, and (5) AutoETS. For complete results, refer to Appendix D.
  + Additionally, we conducted experiments with a very recent and relevant method, namely LLMTime, in the context of zero-shot forecasting tasks. For detailed results, please see Appendix E.
+ In response to **Reviewer L3pe**'s feedback, we have updated the discussion on future work and Appendix A to more clearly define the scope and background of this study. We have also furnished ample evidence and thorough justifications for (1) the use of LLMs in time series, (2) model reprogramming, and (3) the Prompt-as-Prefix approach. Additionally, our code has been made available to AC and the reviewers.
+ Addressing the feedback from **Reviewers MAkL** and **baDP**, we have enhanced Appendix B.1 by incorporating further technical details.
+ Following feedback from **Reviewer MAkL**, we have updated Fig. 3(a) to include additional notations and explanations. We have also provided clarification on the mechanism behind Time-LLM in producing forecasts.
+ In response to feedback from **Reviewer baDP**, we have updated Fig. 2. Additionally, we have revised Appendix A and meticulously reviewed all entries in the reference list.

---

### Public Comment · ~Jiahui_Li3 · 2023-12-03
**Inquiry for code**

Hi,
Would you mind sharing your code?

---

### Meta-Review · Area_Chair_f2B7 · 2023-12-12

**Metareview:**

This paper presents a new framework for time series forecasting using Large Language Models (LLMs), denoted Time-LLM. The presented approach introduces two innovations: (i) a methodology to transforms time series data into text-like formats; and (ii) a method, called 'Prompt-as-Prefix' (PaP), for contextual enhancement. The paper provides extensive empirical results, comparing Time-LLM's performance across various settings with related models. The authors also consider few-shot and zero-shot scenarios. Technically, the paper advances our understanding of LLM application in time series forecasting and promises to inspire further research in this field. The authors have satisfactorily addressed most concerns raised during the rebuttal phase, thereby improving the paper. A weakness that remains is the poor theoretical motivation for the proposed methodology to transforms time series data into text-like formats.

There was a heated debate between Reviewer L3pe and the authors, primarily over discrepancies in the code concerning the early stopping criteria. Following this, two other reviewers adjusted their scores to borderline, leaving only Reviewer baDP as a strong advocate of the paper. This situation made the paper borderline.

While I consider Reviewer L3pe's concerns with due seriousness, they don't, in my opinion, warrant rejection of the paper. The code issue only affects a minor portion of the results and does not seem to stem from any deliberate manipulation. The provided code is transparent and enables reproducibility. It is not clear to me, whether the right early stopping policy for these experiments is definitively established. Ideally, re-running all experiments, including baselines, under a uniform policy would strengthen the paper's findings, though this is subject to resource and time constraints.

In conclusion, the paper's novel and stimulating ideas contribute to the methodology of time-series forecasting. Considering the revised reviewer scores post-rebuttal and my assessment, I advocate for the acceptance of this paper.

**Justification For Why Not Higher Score:**

This paper could be considered for spotlight, since the presented ideas are novel and impactful.  However, the concerns by Reviewer L3pe are not fully resolved.

**Justification For Why Not Lower Score:**

This paper could be rejected, but I strongly feel that the paper's novel and stimulating ideas contribute to the methodology of time-series forecasting. The authors did an extremely good job addressing concerns and providing additional experiments during the rebuttal phase.

---

### Decision · Program_Chairs · 2024-01-16

Accept (poster)